# Statistical Guarantees for Approximate Stationary Points of Shallow Neural Networks

**Mahsa Taheri**  *mahsa.taheri@uni-hamburg.de*
*Department of Mathematics*
*University of Hamburg*

**Fang Xie**[*]  *fangxie@bnbu.edu.cn*
*Guangdong Provincial Key Laboratory of IRADS*
*Beijing Normal-Hong Kong Baptist University*

**Johannes Lederer**  *johannes.lederer@uni-hamburg.de*
*Department of Mathematics*
*University of Hamburg*

**Reviewed on OpenReview:** *https://openreview.net/forum?id=PNUMiLbLml*

## Abstract

Since statistical guarantees for neural networks are usually restricted to global optima of intricate objective functions, it is unclear whether these theories explain the performances of actual outputs of neural network pipelines. The goal of this paper is, therefore, to bring statistical theory closer to practice. We develop statistical guarantees for shallow linear neural networks that coincide up to logarithmic factors with the global optima but apply to stationary points and the points nearby. These results support the common notion that neural networks do not necessarily need to be optimized globally from a mathematical perspective. We then extend our statistical guarantees to shallow ReLU neural networks, assuming the first layer weight matrices are nearly identical for the stationary network and the target. More generally, despite being limited to shallow neural networks for now, our theories make an important step forward in describing the practical properties of neural networks in mathematical terms.

## 1 Introduction

Statistical theories for deep learning usually apply to exact, global optima of certain objective functions (Bartlett, 1998; Bauer & Kohler, 2019; Kohler & Langer, 2021; Lederer, 2022a; Schmidt-Hieber, 2020; Mohades & Lederer, 2025; Golestaneh et al., 2025). But those objective functions cannot be solved explicitly and are highly non-convex, so that in practice, exact, global optimization is—at least to date—an open research question, and we can currently expect only approximate stationary points from current (general) algorithms (see Figure 1). In other words, it is unclear whether the known theories have any meaning for the outputs of actual deep-learning pipelines.

Also other parts of machine learning face optimization problems that are challenging to optimize globally and to full precision. Accordingly, some statistical insights have already been established. For example, Bien et al. (2018; 2019) solve a non-convex problem in linear regression in a "convex" way and develop statistical theories for their solution. Some insights on the statistical theory of stationary points for (simple) non-convex objectives have already been presented: Loh & Wainwright (2015, Theorems 1,2) extract statistical guarantees for stationary points of non-convex objectives (allowing for non-convexity in both loss and penalty functions) in a regression-type settings, under a so-called "restricted-strong convexity" condition over the empirical

---
[*]Corresponding author

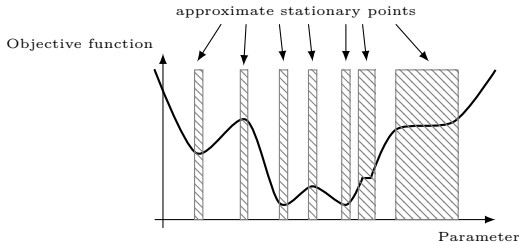

Figure 1: Since objective functions in deep learning are usually highly non-convex and cannot be solved explicitly, we can only expect approximate stationary points from practical algorithms.

loss (see their Display (4)). Loh (2017, Theorem 1) studies the behavior of stationary points of penalized robust estimators in a linear-regression setting. They prove that under a local "restricted-strong convexity" condition, stationary points within the region of restricted curvature are statistically consistent with the target. Also Elsener & van de Geer (2018, Theorem 1) derive sharp oracle inequalities for stationary points of general non-convex objectives made by a non-convex loss plus a convex penalty, under a restrictive condition called "two point marginal condition" on the theoretical loss. They exemplify their bounds for simple models like robust regression and binary classification. Their condition is kinda similar to the restricted-strong convexity but on the theoretical loss (and not on the empirical loss).

However, it remains unclear how to extend these insights to deep learning, as they either apply only to simple models like linear regression or rely on specific curvature assumptions, while it is not clear if such curvature properties hold even for simple networks.

This paper develops statistical guarantees for the stationary points of shallow neural networks and for the points in the vicinity of them. Strikingly, our statistical rates match the rates of global optimizers up to log-terms (Taheri et al., 2021; Lederer, 2022a; Golestaneh et al., 2025). Thus, our results establish a mathematical proof of the "empirical fact" that global optimization is not necessary in deep learning. This complements and contrasts studies about the existence or non-existence of spurious local minima and saddle points in both linear and non-linear networks (Zhou & Liang, 2018; Fukumizu & Amari, 2000; Safran & Shamir, 2018; Lederer, 2020; Liu, 2022).

One of the main challenges in the proofs is the complexity, intricacy, and ambiguity of the parameter space of neural networks. To address this challenge, we introduce scaling tricks (Taheri et al., 2021) and use particular arguments from empirical-process theory for regularized objectives. Moreover, in strong contrast to most theory papers, we focus on regression, which is more general and mathematically more challenging than classification. For example, unbounded losses like least-squares cannot be treated (at least not directly) with standard techniques like McDiarmid's inequality (McDiarmid, 1989, Lemma 3.3) or Rademacher complexities (Mohri et al., 2018, Chapter 3). Thus, our work also contributes considerably on the technical aspects of deep learning.

**Paper contribution**   The three main technical contribution of this paper are as follows:

1. We show that every (reasonable) stationary point of regularized shallow linear neural networks and the points nearby generalize essentially as well as the global optima (Theorem 1 and Theorem 2).

2. We extend our theories to shallow ReLU neural networks for specific stationary points (Theorem 3).

3. We determine the optimal rates for the tuning parameter across different networks and noise distributions (Theorem 4).

Of course, our theoretical framework is still far from the extremely complex pipelines of modern deep learning. But our paper makes considerable progress in closing the gap between our theoretical understanding and practical experiences. In particular, it (i) strengthens the statistical foundations of deep learning and

(ii) gives a first mathematically rigorous proof of the empirical finding that (ii.A) approximate and (ii.B) local optimization of neural networks is usually sufficient in practice.

**Paper outline**  Section 2 states the statistical guarantees for the stationary points of the shallow linear neural network (Theorem 1) and the points nearby (Theorem 2). We extend our theories to shallow ReLU networks in Section 3 (Theorem 3). We support our theories with numerical observations in Section 4. Section 5 provides an overview of related works. We represent some of our technical results in Section 6 and extend our theory for heavy-tailed noise in Sections 7. We conclude our paper in Section 8. More technical results, detailed proofs, and discussion on different assumptions are given in the Appendix.

**Notations**  We use $\text{vec}(\boldsymbol{\gamma}, \Theta)$ to generate a vector of length $\mathbb{R}^{w+w \cdot d}$ from a vector $\boldsymbol{\gamma} \in \mathbb{R}^w$ and a matrix $\Theta \in \mathbb{R}^{w \times d}$ (for generating the vector, we first push the elements of $\boldsymbol{\gamma}$ and then elements of $\Theta$ row by row). We collect first-order partial derivatives (and subdifferentials for ReLU networks) of prediction risk $\text{risk}_X[\boldsymbol{\gamma}, \Theta]$ and population risk $\text{risk}[\boldsymbol{\gamma}, \Theta]$ with respect to the $\boldsymbol{\beta} := \text{vec}(\boldsymbol{\gamma}, \Theta)$ in the gradient vectors $\nabla \text{risk}_X[\boldsymbol{\gamma}, \Theta] \in \mathbb{R}^{w+w \cdot d}$ and $\nabla \text{risk}[\boldsymbol{\gamma}, \Theta] \in \mathbb{R}^{w+w \cdot d}$, respectively. We use the notation $\| \cdot \|$ for a general vector norm and $\|\| \cdot \|\|$ for a general matrix norm. We also define $\|\boldsymbol{\gamma}\|_1 := \sum_{j=1}^{w} |\gamma_j|$ and $\|\|\Theta\|\|_1 := \sum_{j=1}^{w} \sum_{k=1}^{d} |\theta_{jk}|$. To reduce the amount of notations, we use some notation slightly differently depending on whether we treat linear or ReLU networks.

## 2  Statistical guarantees for shallow linear neural networks

Consider inputs $\boldsymbol{x}_1, \ldots, \boldsymbol{x}_n \in \mathbb{R}^d$ and corresponding outputs $y_1, \ldots, y_n \in \mathbb{R}$ that are connected via

$$y_i = f[\boldsymbol{x}_i] + u_i \tag{1}$$

for an unknown target function $f : \mathbb{R}^d \to \mathbb{R}$ and unknown stochastic noise $u_1, \ldots, u_n \in \mathbb{R}$. Deep learning is about using the available data to approximate the unknown target function $f$ by a neural network. We first focus on linear neural networks, a well-accepted toy model for more general deep learning pipelines (Saxe et al., 2013); hence, we consider

$$\boldsymbol{x} \mapsto \boldsymbol{\gamma}^\top \Theta \boldsymbol{x},$$

where

$$(\boldsymbol{\gamma}, \Theta) \in \mathcal{B} := \left\{ (\boldsymbol{\gamma}, \Theta) \in \mathbb{R}^w \times \mathbb{R}^{w \times d} \right\}.$$

We extend this setup to ReLU activation in the following section.

To avoid unnecessary digression here, we impose three mild assumptions. The assumptions are by no means necessary and relaxed in the following sections.

**Assumption 1** (Model Assumptions). *We assume that:*

1. *The target function can be approximated by such a neural network in the first place: there is a pair $(\boldsymbol{\gamma}^*, \Theta^*) \in \mathcal{B}$ such that $\|\boldsymbol{\gamma}^*\|_1, \|\|\Theta^*\|\|_1 \leq \sqrt{\log n}$ and $f[\boldsymbol{x}] = \boldsymbol{\gamma}^{*\top} \Theta^* \boldsymbol{x}$ for all $\boldsymbol{x} \in \mathbb{R}^d$.*

2. *The $\boldsymbol{x}_i$'s are independent and centered sub-Gaussian random vectors with independent coordinates.*

3. *The $u_i$'s are independent centered Gaussian random variables with standard deviation $\sigma$ and are independent of $\boldsymbol{x}_i$'s.*

The first part of Assumption 1 ensures a sharp focus on statistical guarantees rather than the approximation properties of neural networks, we assume that the target function is itself a neural network with reasonably small parameters. A detailed description of the assumption is provided in Section E of the Appendix; the assumption is relaxed in Theorem 5. Note that the parametrization of neural networks is ambiguous: there are infinitely many pairs $(\boldsymbol{\gamma}^*, \Theta^*) \in \mathcal{B}$ that satisfy those conditions—compare to Taheri et al. (2021, Proposition 1); for further reference, we define $\boldsymbol{\beta}^* := \text{vec}(\boldsymbol{\gamma}^*, \Theta^*)$ for a fixed but arbitrary such pair of parameters. The second part of the assumption on the input simplifies our theoretical analysis. Although it may not always strictly hold in practice, it is widely adopted in the literature as a convenient modeling device;

see, for example Vershynin (2018); Wainwright (2019); van de Geer (2000). Intuitively, each coordinate of $\boldsymbol{x}$ can be interpreted as a feature extracted from the raw data, and independence is then a simplifying assumption. Moreover, the mean-zero sub-Gaussian property is often assumed to capture concentration behavior of feature vectors in high-dimensional models and can be achieved through data preprocessing. The third part of the assumption, once more, simplifies the presentation here; extensions to other types of noise, including sub-Gaussian and sub-exponential noise are provided in Section 7.

We assume our regression setup ($y_i \in \mathbb{R}$) rather than a classification setup ($y_i \in \{0,1\}$ or $y_i \in \{1, \ldots, k\}$) because the unbounded outputs make regression considerably more challenging to analyze mathematically. In other words, our regression results transfer readily to classification. The usual loss function in regression is least squares. In deep-learning practice, however, least squares (and similarly logistic loss in classification) is complemented with dropout (Srivastava et al., 2014; Salehinejad & Valaee, 2019), batch normalization (Ioffe & Szegedy, 2015), low-rank approximation (Denil et al., 2013), and so forth, which yield implicit regularization, or least squares is even complemented with explicit regularization directly (Alvarez & Salzmann, 2016; Lemhadri et al., 2021; Hebiri et al., 2025). It is well understood that implicit regularization is related to explicit regularization (Lütke Schwienhorst et al., 2024). Thus, to mimic deep-learning practice, we consider least-squares complemented by (elementwise) $\ell_1$-regularization:

$$(\widehat{\boldsymbol{\gamma}}, \widehat{\Theta}) \ \in \ \operatorname*{arg\,min}_{(\boldsymbol{\gamma}, \Theta) \in \mathcal{B}} \left\{ \frac{1}{n} \sum_{i=1}^{n} \left( y_i - \boldsymbol{\gamma}^\top \Theta \boldsymbol{x}_i \right)^2 + r \|\operatorname{vec}(\boldsymbol{\gamma}, \Theta)\|_1 \right\}, \tag{2}$$

where $r \in [0, \infty)$ is a tuning parameter to be calibrated (see Sardy et al. (2020) for some theory insights). Such estimators are standard in machine learning and statistics (Lederer, 2022b; Eldar & Kutyniok, 2012). Despite $\ell_1$-norm is non-smooth, it often poses very little problems in terms of computations (see Friedman et al. (2010)). Also recently, the $\ell_1$-norm has been effectively used to promote sparsity in neural networks (Lemhadri et al., 2021).

As usual, we measure the (in-sample-)prediction risk by

$$\operatorname{risk}_X[\boldsymbol{\gamma}, \Theta] \ := \ \frac{1}{n} \sum_{i=1}^{n} \left( y_i - \boldsymbol{\gamma}^\top \Theta \boldsymbol{x}_i \right)^2$$

with $X := (\boldsymbol{x}_1, \ldots, \boldsymbol{x}_n)^\top \in \mathbb{R}^{n \times d}$ and the generalization risk by

$$\operatorname{risk}[\boldsymbol{\gamma}, \Theta] \ := \ \mathbb{E}_{(\boldsymbol{x}, y)} \left[ \left( y - \boldsymbol{\gamma}^\top \Theta \boldsymbol{x} \right)^2 \right]$$

with the expectation over a new sample $(\boldsymbol{x}, y)$ (that has the same distribution as $\boldsymbol{x}_1, \ldots, \boldsymbol{x}_n$ and $y_1, \ldots, y_n$). We call $\widetilde{\boldsymbol{\beta}} := \operatorname{vec}(\widetilde{\boldsymbol{\gamma}}, \widetilde{\Theta})$ a *stationary point* of the objective in equation 2 if it satisfies (Bertsekas, 1997, Page 194);(Elsener & van de Geer, 2018, Equation 6);(Loh & Wainwright, 2015, Equation 5)

$$\left( \nabla \operatorname{risk}_X[\widetilde{\boldsymbol{\gamma}}, \widetilde{\Theta}] \right)^\top (\boldsymbol{\beta} - \widetilde{\boldsymbol{\beta}}) + r \tilde{\boldsymbol{z}}^\top (\boldsymbol{\beta} - \widetilde{\boldsymbol{\beta}}) \ \geq \ 0 \qquad \forall\, \boldsymbol{\beta} = \operatorname{vec}(\boldsymbol{\gamma}, \Theta) \ \text{with} \ (\boldsymbol{\gamma}, \Theta) \in \mathcal{B} \tag{3}$$

for appropriate $\tilde{\boldsymbol{z}} \in \partial \|\widetilde{\boldsymbol{\beta}}\|_1$ (where $\partial \|\widetilde{\boldsymbol{\beta}}\|_1$ is the subdifferential of the regularizer at $\widetilde{\boldsymbol{\beta}}$). For an interior point $\widetilde{\boldsymbol{\beta}}$, our definition of stationary points in equation 3 reduces to the usual zero-subgradient condition.

We call a stationary point $\widetilde{\boldsymbol{\beta}}$ *reasonable* once $\|\widetilde{\boldsymbol{\gamma}}\|_1, \|\widetilde{\Theta}\|_1 \leq \sqrt{\log n}$—again to avoid unnecessary complication (we refer to the Appendix Section F for a detailed description of the reasonability assumption). Due to the ambiguity of neural networks, there are infinitely many equivalent stationary and reasonable stationary points; importantly, our guarantees hold for every (reasonable) stationary point and target $\boldsymbol{\beta}^*$.

We say that a network indexed by $(\widetilde{\boldsymbol{\gamma}}, \widetilde{\Theta})$ generalizes well if

$$\operatorname{risk}[\widetilde{\boldsymbol{\gamma}}, \widetilde{\Theta}] \ \approx \ \operatorname{risk}[\boldsymbol{\gamma}^*, \Theta^*],$$

that is, the network generalizes essentially as well as the best network. In the following, we show that not only the "statistical" network indexed by $(\widehat{\boldsymbol{\gamma}}, \widehat{\Theta})$ but also every "practical" network indexed by a reasonable stationary point $(\widetilde{\boldsymbol{\gamma}}, \widetilde{\Theta})$ of the objective function in equation 2 generalizes well.

Moreover, we call the total number of parameters in the network $p := w + w \cdot d$ the problem's effective dimension and

$$r_{\text{orc}} := \nu(\log n)^{3/2}\sqrt{\frac{\log(np)}{n}} \tag{4}$$

the oracle tuning parameter, where $\nu \in (0,\infty)$ is a constant that depends only on the distributions of the inputs and noise. It has been shown that $r_{\text{orc}}$ is indeed an optimal tuning parameter of equation 2 in some sense (Taheri et al., 2021).

We then get the following result for a new sample pair $(\boldsymbol{x}, y)$ with the same distribution as $\boldsymbol{x}_1, \ldots, \boldsymbol{x}_n$ and $y_1, \ldots, y_n$.

**Theorem 1** (Statistical Guarantees for Reasonable Stationary Points of Shallow Linear Networks). *Under the Assumption 1 any reasonable stationary point $(\widetilde{\boldsymbol{\gamma}}, \widetilde{\Theta})$ of the objective function in equation 2 with $r \geq r_{\text{orc}}$ satisfies the risk bound*

$$\text{risk}[\widetilde{\boldsymbol{\gamma}}, \widetilde{\Theta}] \leq \text{risk}[\boldsymbol{\gamma}^*, \Theta^*] + 5r\sqrt{\log n} \tag{5}$$

*with probability at least $1 - 1/2n$. If $r = r_{\text{orc}}$, the bound becomes*

$$\text{risk}[\widetilde{\boldsymbol{\gamma}}, \widetilde{\Theta}] \leq \text{risk}[\boldsymbol{\gamma}^*, \Theta^*] + \nu(\log n)^2\sqrt{\frac{\log(np)}{n}}. \tag{6}$$

Theorem 1 proves the fact that for properly chosen tuning parameter $r$ and large enough sample sizes, any reasonable stationary point of equation 2 generalizes essentially as well as $\boldsymbol{\beta}^*$. Our results essentially have the same rates as the ones in the literature (Taheri et al., 2021, Theorem 3);(Lederer, 2022a, Proposition 3), who prove that the prediction risk is at most of order $O((L/2)^{1/2-L}\log(p)\log(n)/\sqrt{n}))$ for $\ell_1$-regularized neural networks with depth $L$ and $p$ parameters. However, in stark contrast to previous results, our theories apply to all reasonable stationary points (including saddle points) rather than to the global optimum of the objective function only. Although works like Kawaguchi (2016) and Zhou & Liang (2018) argue about the absence of local minima in linear networks, saddle points still exist in linear neural networks (see Zhou & Liang (2018, Theorem 2)). Furthermore, saddle points continue to pose challenges: Lee et al. (2019) demonstrate that gradient-based algorithms can escape strict saddle points, but non-strict saddle points are problematic and also exist in linear neural networks in general (Zhou & Liang, 2018, Paragraph following their Theorem 2). We refer to our illustrative Example 1 (in the Appendix) to clearly illustrate the presence of sub-optimal critical points in our considered setup. Also, a recent study by Achour et al. (2024) demonstrates that for shallow linear neural networks with least squares loss, all saddle points are strict under some assumptions (see Achour et al. (2024, Assumption 1).

To emphasize the significance of using regularized objectives, it's worth mentioning that the rate of ordinary least-squares in linear regression is $O(d/n)$, where $d$ gives the number of parameters and $n$ the number of data examples (Lederer, 2022b, Equation 1.5). But for high-dimensional settings with $d \gg n$, least-squares are prone to overfitting, so regularization can be employed for improvement. For example, lasso with sufficiently large tuning parameter (in linear regression) gives predictions bounds at most bounded by $\sqrt{\log(d)/n}$ (Lederer, 2022b, Page 174). Also, a different prediction bound for lasso called "power-two bound" is presented in Lederer (2022b, Page 188) that holds under strong conditions but it is far from the context of this paper. Overfitting is even more problematic for complex models like neural networks with a huge number of parameters $p$. The focus has just shifted to networks involving sparsity to improve prediction bounds from $p/n$ to $\sqrt{\log(p)/n}$, which also appears in our results (see equation 6 for example).

Note that in finite time, stationary points can be computed just approximately using gradient-based algorithms. Now, we extend our results in Theorem 1 to the points that are close but not necessarily equal to a stationary points. We define a pair $(\widetilde{\widetilde{\boldsymbol{\gamma}}}, \widetilde{\widetilde{\Theta}})$ as a $\tau-$approximate stationary point if it satisfies

$$\left| \text{risk}_X[\widetilde{\widetilde{\boldsymbol{\gamma}}}, \widetilde{\widetilde{\Theta}}] + r\|\widetilde{\widetilde{\boldsymbol{\beta}}}\|_1 - \text{risk}_X[\widetilde{\boldsymbol{\gamma}}, \widetilde{\Theta}] - r\|\widetilde{\boldsymbol{\beta}}\|_1 \right| \leq \tau \tag{7}$$

for a $\tau \in [0,\infty)$. Our definition of approximate stationary points in equation 7 is closely related to the typical definitions in the literature that impose some bounds on the norm of the gradient vectors (see Appendix

Section G for a detailed description). Employing gradient-based algorithms (in finite time), we can expect to get close to a stationary point in the sense that $\widetilde{\widetilde{\boldsymbol{\beta}}} \approx \widetilde{\boldsymbol{\beta}}$ (Ghadimi & Lan, 2013; Lei et al., 2019). Then also $\|\widetilde{\widetilde{\boldsymbol{\beta}}}\|_1 \approx \|\widetilde{\boldsymbol{\beta}}\|_1$, which means that an approximation of a reasonable stationary point is also reasonable once $\tau$ is small enough. Then, we extract statistical guarantees for every practical network indexed by an approximate-reasonable stationary as follows:

**Theorem 2** (Statistical Guarantees for Approximate Stationary Points of Shallow Linear Networks). *Suppose that $(\widetilde{\widetilde{\boldsymbol{\gamma}}}, \widetilde{\widetilde{\Theta}})$ is a $\tau-$approximate stationary point and that the conditions of Theorem 1 are satisfied. Then, we have*

$$\mathrm{risk}[\widetilde{\widetilde{\boldsymbol{\gamma}}}, \widetilde{\widetilde{\Theta}}] \ \leq \ \mathrm{risk}[\boldsymbol{\gamma}^*, \Theta^*] + 8r\sqrt{\log n} + \tau \tag{8}$$

*with probability at least $1 - 1/n$. If $r = r_{\mathrm{orc}}$, the bound becomes*

$$\mathrm{risk}[\widetilde{\widetilde{\boldsymbol{\gamma}}}, \widetilde{\widetilde{\Theta}}] \ \leq \ \mathrm{risk}[\boldsymbol{\gamma}^*, \Theta^*] + \nu(\log n)^2\sqrt{\frac{\log(np)}{n}} + \tau \,. \tag{9}$$

The bounds match the earlier ones with only two small differences: 1. a summand $\tau$ is added to our statistical bounds and 2. the factor 5 in equation 5 is replaced by a factor of 8 in equation 8. Let's note that gradient-based algorithms with sufficiently many steps $O(n^2)$ ensure that $\tau \ll 1/\sqrt{n}$ (Ghadimi & Lan, 2013, Theorem 2.1). We refer to our Appendix Section G for more details regarding the dynamical accessibility of approximate stationary points. Theorem 2 might look like a simple extension of Theorem 1, but the fact that equation 7 involves the (in-sample-)prediction risk and the sparsity factors makes the proof considerably more involved.

## 3 Statistical guarantees for shallow ReLU neural networks

This section generalizes our theories in Section 2 to shallow ReLU neural networks of the form

$$\boldsymbol{x} \ \mapsto \ \boldsymbol{\gamma}^\top \boldsymbol{\sigma}(\Theta\boldsymbol{x}) \,,$$

for $(\boldsymbol{\gamma}, \Theta) \in \mathcal{B} = \{(\boldsymbol{\gamma}, \Theta) \in \mathbb{R}^w \times \mathbb{R}^{w \times d}\}$. The activation function $\boldsymbol{\sigma}(\cdot)$ corresponds to the well-known ReLU defined as $\boldsymbol{\sigma}(\boldsymbol{z}) := (\max(0, z_1), \ldots, \max(0, z_w))$ for $\boldsymbol{z} \in \mathbb{R}^w$, which its efficacy has been extensively studied (Pan & Srikumar, 2016; Raghu et al., 2017). We then approximate the unknown target function $f$ in equation 1 employing shallow ReLU neural networks. For simplifying the proofs, we assume in this section that $d = w$ that implies matrix $\Theta$ to be squared. We then consider least-squares complemented by $\ell_1$-regularization for shallow ReLU neural networks:

$$(\widehat{\boldsymbol{\gamma}}, \widehat{\Theta}) \ \in \ \underset{(\boldsymbol{\gamma}, \Theta) \in \mathcal{B}}{\arg\min}\left\{\frac{1}{n}\sum_{i=1}^{n}\left(y_i - \boldsymbol{\gamma}^\top \boldsymbol{\sigma}(\Theta\boldsymbol{x}_i)\right)^2 + r\|\mathrm{vec}(\boldsymbol{\gamma}, \Theta)\|_1\right\}. \tag{10}$$

**Assumption 2** (Model Assumptions (ReLU)). *We assume that the target function can be approximated by such a neural network, that is, there is a pair $(\boldsymbol{\gamma}^*, \Theta^*) \in \mathcal{B}$ such that $\|\boldsymbol{\gamma}^*\|_1, \|\Theta^*\|_1 \leq \sqrt{\log n}$ and active rows in $\Theta^*$ are approximately perpendicular to each other and that $f[\boldsymbol{x}] = \boldsymbol{\gamma}^{*\top} \boldsymbol{\sigma}(\Theta^*\boldsymbol{x})$ for all $\boldsymbol{x} \in \mathbb{R}^d$.*

The term active rows in a matrix $\Theta$ are approximately perpendicular to each other in our assumption above means, for any two distinct active rows $\Theta_{j,\cdot}$ and $\Theta_{j',\cdot}$ (where $\Theta_{j,\cdot}, \Theta_{j',\cdot} \neq \boldsymbol{0}$), their inner product is negligible, that is $\langle \Theta_{j,\cdot}, \Theta_{j',\cdot} \rangle \approx 0.0$. Assumption 2 stipulates that the target function is itself a shallow ReLU neural network with reasonably small parameters and that the active rows of the first layer are approximately orthogonal. Versions of these assumptions are very common in the literature (Hardt & Ma, 2016; Bartlett et al., 2018b); we discuss this assumption further in the paragraph following Theorem 3. We then define the (in-sample-)prediction and generalization risk for shallow ReLU neural networks as (we employ the same notation as used in the linear case)

$$\mathrm{risk}_X[\boldsymbol{\gamma}, \Theta] \ := \ \frac{1}{n}\sum_{i=1}^{n}\left(y_i - \boldsymbol{\gamma}^\top \boldsymbol{\sigma}(\Theta\boldsymbol{x})\right)^2$$

and

$$\text{risk}[\boldsymbol{\gamma}, \Theta] \; := \; \mathbb{E}_{(\boldsymbol{x}, y)}\left[\left(y - \boldsymbol{\gamma}^\top \boldsymbol{\sigma}(\Theta \boldsymbol{x})\right)^2\right].$$

We then get the following result for a new sample pair $(\boldsymbol{x}, y)$ with the same distribution as $\boldsymbol{x}_1, \dots, \boldsymbol{x}_n$ and $y_1, \dots, y_n$.

**Theorem 3** (Statistical Guarantees for Reasonable Stationary Points of Shallow ReLU Networks)**.** *Under the second and third parts of Assumption 1 and Assumption 2, any reasonable stationary point $(\widetilde{\boldsymbol{\gamma}}, \widetilde{\Theta})$ of the objective function in equation 10 where active rows of $\widetilde{\Theta}$ are approximately perpendicular to each other and that off-diagonal elements of $\widetilde{\Theta}\Theta^{*\top}$ and $\Theta^*\widetilde{\Theta}^\top$ are approximately zero ($|(\widetilde{\Theta}\Theta^{*\top})_{jj'}| \approx |(\Theta^*\widetilde{\Theta}^\top)_{jj'}| \approx 0.0$ for $j \neq j'$) with $r \geq r_{\mathrm{orc}}$ satisfies the risk bound*

$$\text{risk}[\widetilde{\boldsymbol{\gamma}}, \widetilde{\Theta}] \; \leq \; \text{risk}[\boldsymbol{\gamma}^*, \Theta^*] + 5r\sqrt{\log n} \tag{11}$$

*with probability at least $1 - 1/2n$.*

Note that Theorem 3 is an extension of our Theorem 1 for shallow ReLU neural networks under the assumption that the active rows of the first layer weight matrix (for stationary point and the target) being approximately orthogonal (one simple example is near-identity matrices). Orthogonal weight matrices is needed mainly for studying Hessian behavior for shallow ReLU network's (proof of Proposition 2, Step 2). Employing this assumption, we can prove that for cases with small correlation between rows and with Gaussian input, we can find a closed-form solution (approximately) for $\mathbb{E}_{\boldsymbol{x}}[\boldsymbol{\sigma}(\Theta \boldsymbol{x})_j \boldsymbol{\sigma}(\Theta \boldsymbol{x})_{j'}]$. Our Assumption 2 is weaker than it seems as previous works have studied variants of this assumption for neural networks from different perspectives: for example, Hardt & Ma (2016) shows that certain networks have a global minimum close to the identity parameterization. They study the expressiveness of Residual Networks under the assumption that enough neurons are available (Hardt & Ma, 2016, Theorem 3.2). Interesting is that, since our rates grow just in $\log p$, our framework is perfectly fit for such wide networks. Additionally, Bartlett et al. (2018a) explore the representation of smooth functions as compositions of near-identity functions, highlighting implications for deep network optimization. Bartlett et al. (2018b) prove the rate of convergence of gradient-based optimization under identity initialization for deep linear networks. Li & Yuan (2017) analyze the convergence of stochastic gradient descent for shallow ReLU networks, with nearly identity initialization; they state that "(ReLU) networks with small average spectral norm already have good performance." Altogether, we believe that our assumption makes sense not only from an expressivity standpoint (Hardt & Ma, 2016, Theorem 3.2) but also regarding the optimization landscape (Li & Yuan, 2017). Yet, of course, it would be interesting to study the subtleties even further. While studies demonstrate the existence of local minima and saddle points in ReLU networks (Fukumizu & Amari, 2000; Safran & Shamir, 2018; Yun et al., 2019), we argue that some of those suboptimals still yield satisfactory results. Essentially, Theorem 3 suggests that for a sufficiently large tuning parameter, the optimization explores locally well-curved network spaces in the vicinity of specific stationary points, such that any stationary point generalizes as effectively as a global minimum. In fact, our work concerns local curvature around the ground truth in neural networks, which we believe is valuable given the infinite number of such ground truths in neural networks, while globally favorable curvature is far from practical reality in deep learning. We employ our result in Proposition 2 and Remark 1 proving our Theorem 3. Also, an extension of Theorem 2 to shallow ReLU networks can be obtained by combining Theorem 3 with additional machinery from empirical process theory, following the same line of reasoning as in the proof of Theorem 2. However, we omit this extension here to avoid redundancy. Also, we conjecture that our main theories can be extended to deep neural networks (see our simulations in Section D), provided that suitable local curvature properties of the corresponding networks can be established. This presents an intriguing direction for future research.

**Further discussion of our Assumption 2**  Our results suggest that low correlation between the rows of the first-layer weight matrix $\widetilde{\Theta}$ is desirable, as it leads to a well-conditioned Hessian and better generalization. This observation is closely related to the benefits of random initialization: for large $d$, random Gaussian weights yield nearly orthogonal rows with high probability (see Vershynin (2018, Remark 3.2.5)). However, orthogonality is not only needed at initialization but also for the estimator $\widetilde{\Theta}$ after training, which motivates arguments ensuring that training preserves this structure. Related studies show that fixing the first layer at

its random initialization while only training the last layer can still achieve good generalization (Rosenfeld & Tsotsos, 2019), suggesting that there exist network configurations where the first-layer rows form an approximately orthogonal system, leading to favorable error bounds. Finally, while some of the literature attributes low-rank structure in shallow networks to strong correlations among rows (Kou et al., 2023), in our norm-one regularized setting low rank instead arises through sparsity: many rows might become inactive, while the surviving rows remain diverse and nearly orthogonal, which is enough for our results to hold. One can also considers group lasso to offer an alternative means of promoting structured sparsity. This alternative path to low-rank structure avoids redundancy, preserves conditioning, and further explains why such solutions generalize well.

## 4 Numerical observations

We provide here some numerical observations to clarify theories of Section 2 and Section 3. We minimize a least-squares complemented by $\ell_1$-regularization for shallow neural networks with linear and ReLU activation functions. We set our tuning parameter on the order of $\log(np)/\sqrt{n}$ based on our experiments. We consider neural networks with $d = w = 10$, that are trained over 500 and tested over 300 data sample generated from a standard normal distribution and labeled by a sparse-target network (having the same structure as the considered model) plus a Gaussian noise. Note that here, we train the networks in a finite time, that means, trained networks are just an approximation of a stationary point (due to the non-convexity). We report the relative training error and the relative test error for a potential global optimum, an approximate stationary point, and a randomly generated network (a network with randomly assigned weights) for linear and ReLU networks in Table 1, that is, the training (test) error of the "approximate stationary point" divided by the training (test) error of the "potential global optimum" (for the corresponding network). Potential global optimum and approximate stationary point (for each setting, linear or ReLU) are reached over multiple times of training on a fixed data set and assigned by the trained networks with the lowest and highest training error, respectively. More precisely, we do the optimization (solving equation 2 and equation 10) from multiple, diverse initial points (1000 times). We use PyTorch's default initialization, where weights are drawn from a uniform distribution in $[-1/\sqrt{p}, 1/\sqrt{p}]$, with $p$ denoting the number of input features to the layer (see our results with different initialization methods in our Section D). This helps explore different regions of the search space and increases the chances of finding different local and global optimum. Note that there are infinitely many critical points for neural networks in view of the network's rescaling properties. We use stochastic gradient descent with a small convergence threshold to ensure that the optimization process does not stop early. We analyze the distribution of the reached training errors (over the 1000 different optimization runs with random initialization). For this, we divide the training errors into two clusters via k-means. Then, we do a t-test over the training errors in the two classes. The t-test reveals a statistically significant difference between the training errors in two groups ($p_{\text{value}} < 0.0001$), which supports the claim that the "potential global optimum" and "approximate stationary points" differ, that is, the approximate stationary points are not just other global optima. We then report the parameters that lead to the lowest training error as a "potential global optimum" and the parameters that lead to the highest training error as "approximate stationary point". We reference to Figure 3 in the Appendix Section D for a graphical view of convergence in training. Results reveal that the test error for a potential global optimum and an approximate stationary point are very close in both linear and ReLU networks (relative errors for approximate stationary points are close to one for both linear and ReLU networks). Also note that the reported numbers in Table 1 are just relative errors to compare between training and test performance of a specific network so, a comparison between the performance of linear and ReLU networks here is not meaningful.

These observations reveal that: First, global optimization for neural networks is far reaching even for very simple neural networks. Second, very practical outputs in deep learning (approximate stationary points) can still generalize well—for linear networks and beyond. We provide the similar result for a larger network in Table 2 and more detailed experiment explanations in Appendix Section D.

Table 1: Relative training error and test error for trained shallow neural networks (with $d = 10, w = 10$) with linear and ReLU activations in a potential global optimum, an approximate stationary point, and a randomly generated network.

| | Linear | | ReLU | |
|---|---|---|---|---|
| | Training Error | Test Error | Training Error | Test Error |
| Potential Global Optimum | 1.000 | 1.000 | 1.000 | 1.000 |
| Approximate Stationary Point | 1.001 | 1.001 | 1.003 | 1.004 |
| Randomly Generated Network | 79618.240 | 58198.240 | 2120.060 | 1980.060 |

## 5 Related literature

Another interesting direction is studying optimization landscape of non-convex objectives in deep learning (Eftekhari, 2020; Hardt & Ma, 2016; Lederer, 2020; Zhou & Liang, 2018; Zhang et al., 2016; Bah et al., 2022; Trager et al., 2020). Yun et al. (2017) study the optimization landscape of deep and linear neural networks. They extract necessary and sufficient conditions for a critical point to be the global optima of the least-squares loss under some assumptions (input dimensions upper bounded by the number of data examples, $XX^\top$ and $YX^\top$ have full rank). Kawaguchi (2016, Theorem 2.3) proves that for deep and linear neural networks and under some assumptions ($XX^\top$ and $XY^\top$ have full rank), every local minimum is a global minimum and every critical point that is not a global minimum is a saddle point. They also prove that the same results hold for nonlinear-neural networks but under unrealistic assumptions (Kawaguchi, 2016, Corollary 3.2). Zhou & Liang (2018, Theorem 2) also prove that linear neural networks with least-squares loss have no spurious local minimum. Haeffele & Vidal (2017) established sufficient conditions ensuring that any local minimum of a non-convex factorization problem is also a global minimum. Moreover, they showed that when the factorization is parameterized with sufficiently large factors, one can always reach a global minimizer from any feasible initialization using purely local descent methods. Nguyen & Hein (2017) studied the loss surface of deep and wide neural networks and proved that, under mild overparameterization conditions, every local minimum is also a global minimum. Nguyen & Hein (2018) analyzed the optimization landscape and expressivity of deep convolutional neural networks. They established conditions under which all local minima are globally optimal and characterized how network depth and architecture affect the expressivity of the network. But in general, the absence of spurious local minima is rejected for non-linear networks (Fukumizu & Amari, 2000; Safran & Shamir, 2018).

More broadly, non-convexity and computational problems of neural networks have widely been studied in recent years from different perspectives, including optimization algorithms (Lovas et al., 2020; Bach & Chizat, 2021), theory of overparameterized networks (Chizat & Bach, 2018), and hyperparameter calibration (Yang et al., 2021). Liang et al. (2018) studied modified neuron activation for an arbitrary deep neural network in binary classification proving that no bad local-min exists (see also Sun et al. (2020)). Choromanska et al. (2015) studied the loss landscape of neural networks from a statistical physics perspective, establishing a connection between neural networks and spin-glass models.

## 6 Technical results

This section provides technical results needed for proving our main theories. All the proofs as well as more related auxiliary results are deferred to the Appendix.

**Additional notations** For vectors $\boldsymbol{\beta} = \text{vec}(\boldsymbol{\gamma}, \Theta) \in \mathbb{R}^p$ and $\boldsymbol{\alpha} := (\alpha_1, \ldots, \alpha_w) \in \mathbb{R}^w$ with $\alpha_j \neq 0$ for all $j \in \{1, \ldots, w\}$, we define $\boldsymbol{\beta_\alpha} := \text{vec}(\boldsymbol{\gamma_\alpha}, \Theta_{\boldsymbol{\alpha}}) \in \mathbb{R}^p$ as a rescaled version of $\boldsymbol{\beta}$ with $(\boldsymbol{\gamma_\alpha})_j := \gamma_j \cdot \alpha_j$ and $(\Theta_{\boldsymbol{\alpha}})_{jk} := \theta_{jk}/\alpha_j$ for all $j \in \{1, \ldots, w\}$ and $k \in \{1, \ldots, d\}$. We tabulate the second order partial derivatives (subdifferentials) of $\text{risk}[\boldsymbol{\gamma}, \Theta]$ with respect to the $\boldsymbol{\beta} = \text{vec}(\boldsymbol{\gamma}, \Theta)$ in a matrix called $\nabla^2 \text{risk}[\boldsymbol{\gamma}, \Theta] \in \mathbb{R}^{p \times p}$. We use $e_{\min}[\cdot]$ to generate the smallest eigenvalue of a matrix. We use the notation $\mathbf{0}$ to generate a vector of zeros.

## 6.1 Technical results for shallow linear neural networks

Here, we provide technical results that are essential for proving our main theories for shallow linear networks but might also be of interest by themselves. We first study the behavior of the Hessian matrix for shallow linear networks in a rescaled network as follows:

**Proposition 1** (Hessian Behavior for Shallow Linear Network). *Suppose Assumption 1 is verified and that* $(\boldsymbol{\gamma}, \Theta) \in \mathcal{B}$ *with* $\Theta\Theta^\top$ *invertible. Let* $\boldsymbol{a} := [(\boldsymbol{a}^1)^\top, (\boldsymbol{a}^2)^\top]^\top \in \mathbb{R}^p$ *be a vector with* $\|\boldsymbol{a}\|_2 = 1$, $\boldsymbol{a}^1 \in \mathbb{R}^w$, *and* $\boldsymbol{a}^2 \in \mathbb{R}^{w \cdot d}$. *If* $\boldsymbol{a}^1 = \boldsymbol{0}$ *or* $\boldsymbol{a}^2 = \boldsymbol{0}$, *we have for all* $\boldsymbol{\alpha} \in \mathbb{R}^w \setminus \{\boldsymbol{0}\}$

$$\boldsymbol{a}^\top \nabla^2 \mathrm{risk}[\boldsymbol{\gamma_\alpha}, \Theta_{\boldsymbol{\alpha}}] \boldsymbol{a} \ \geq \ 0;$$

*Otherwise, above inequality holds for all* $\boldsymbol{\alpha} := (1/c, \ldots, 1/c) \in \mathbb{R}^w$ *with* $c \in [1, \infty)$ *such that*

$$c^2 \geq \frac{2\|\boldsymbol{\gamma}\|_2^2 \|\boldsymbol{a}^2\|_2^2 + 4\|\boldsymbol{a}^1\|_2 \|\boldsymbol{a}^2\|_2 \|\boldsymbol{\gamma}^\top \Theta - \boldsymbol{\gamma}^{*\top}\Theta^*\|_2}{e_{\min}[\Theta\Theta^\top]\|\boldsymbol{a}^1\|_2^2} \, .$$

Note that if $\boldsymbol{a}^1 = \boldsymbol{0}$ or $\boldsymbol{a}^2 = \boldsymbol{0}$, the quadratic product on the Hessian matrix (in a rescaled network with parameters $(\boldsymbol{\gamma_\alpha}, \Theta_{\boldsymbol{\alpha}})$) is non-negative for all $\boldsymbol{\alpha}$, otherwise, it is non-negative just for $\boldsymbol{\alpha}$ with large enough $c$. Proposition 1 is employed for the proof of Theorem 1.

**Lemma 1** (Empirical Processes). *Under the Assumption 1 it holds for each reasonable stationary point* $\widetilde{\boldsymbol{\beta}} = \mathrm{vec}(\widetilde{\boldsymbol{\gamma}}, \widetilde{\Theta})$ *of the objective function in equation 2 that*

$$\left| \left( \nabla \mathrm{risk}_X[\widetilde{\boldsymbol{\gamma}}, \widetilde{\Theta}] - \nabla \mathrm{risk}[\widetilde{\boldsymbol{\gamma}}, \widetilde{\Theta}] \right)^\top (\boldsymbol{\beta}^* - \widetilde{\boldsymbol{\beta}}) \right| \leq r_{\mathrm{orc}} \|\boldsymbol{\beta}^* - \widetilde{\boldsymbol{\beta}}\|_1 + \frac{r_{\mathrm{orc}}}{2n}$$

*with probability at least* $1 - 1/2n$, *where* $r_{\mathrm{orc}}$ *is the oracle tuning parameter defined in equation 4.*

The result above establishes a bound for the absolute difference between $\nabla \mathrm{risk}_X[\widetilde{\boldsymbol{\gamma}}, \widetilde{\Theta}]$ and $\nabla \mathrm{risk}[\widetilde{\boldsymbol{\gamma}}, \widetilde{\Theta}]$ for every reasonable stationary point $(\widetilde{\boldsymbol{\gamma}}, \widetilde{\Theta}) \in \mathcal{B}$ for shallow linear networks (a similar result can also be reached for shallow ReLU networks; see Remark 1). We employ Lemma 1 choosing the optimal tuning parameter for the objective function equation 2.

## 6.2 Technical results for shallow ReLU neural networks

Now, we study the behavior of the Hessian matrix for shallow ReLU networks in a rescaled network. Since ReLU networks are non-differentiable at zero, we employ subdifferentials in this section (instead of partial derivatives) using the same notation as used for linear networks. We suppose that $\nexists \, \boldsymbol{x}$ with $(\Theta\boldsymbol{x})_j = 0$, where $j \in \{1, \ldots, w\}$, then we have

**Proposition 2** (Hessian Behavior for Shallow ReLU Networks). *Suppose Assumption 2 and the second and third parts of Assumption 1 are verified, and that* $(\boldsymbol{\gamma}, \Theta) \in \mathcal{B}$ *with active rows of* $\Theta$ *being approximately perpendicular. Let* $\boldsymbol{a} := [(\boldsymbol{a}^1)^\top, (\boldsymbol{a}^2)^\top]^\top \in \mathbb{R}^p$ *be a vector with* $\|\boldsymbol{a}\|_2 = 1$, $\boldsymbol{a}^1 \in \mathbb{R}^w$, *and* $\boldsymbol{a}^2 \in \mathbb{R}^{w \cdot d}$. *If* $\boldsymbol{a}^2 = \boldsymbol{0}$, *we have for all* $\boldsymbol{\alpha} \in \mathbb{R}^w \setminus \{\boldsymbol{0}\}$

$$\boldsymbol{a}^\top \nabla^2 \mathrm{risk}[\boldsymbol{\gamma_\alpha}, \Theta_{\boldsymbol{\alpha}}] \boldsymbol{a} \ \geq \ 0;$$

*Otherwise, above inequality holds for all* $\boldsymbol{\alpha} := (1/c, \ldots, 1/c) \in \mathbb{R}^w$ *with* $c \in [1, \infty)$ *large enough.*

Note that Proposition 2 is an extension of our Propostion 1 for ReLU networks, that holds under an extra assumption over the first layer weight matrix.

**Remark 1** (Empirical Processes for Shallow ReLU Neural Networks). *Under the Assumption 2 and the second and third parts of the Assumption 1, almost the same bound (up to a constant and log factor) as stated in Lemma 1 can hold for each reasonable stationary point* $\widetilde{\boldsymbol{\beta}} = \mathrm{vec}(\widetilde{\boldsymbol{\gamma}}, \widetilde{\Theta})$ *of the objective function in equation 10.*

As stated in Remark 1, the tuning parameter for ReLU networks can be calibrated similarly to linear networks (although there's potential for improvement, we omit that to avoid unnecessary complication.)

## 7  Heavy-tailed noise

This section puts a focus on heavy-tailed noise. We limit ourselves to linear networks for simplicity, but the same techniques also work in the ReLU case. More generally, this section illustrates the much larger generality—and technical difficulty—of our regression setup as compared to the common classification setups, which are bounded by design.

**Definition 1** (Tails). *Let $I : \mathbb{R} \to \mathbb{R}$ be an increasing function. The function $I$ captures the right tail of the random variable $z$ if*

$$\mathbb{P}(z > t) \leq \exp\big(-I(t)\big), \qquad \forall t \in (0, \infty).$$

In this section, we assume that noise is heavy-tailed, having a right tail as defined in Definition 1 with $I_\alpha(t) = c_\alpha t^{1/\alpha}$ for $c_\alpha \in (0, \infty)$ (for example $\alpha = 1$ for sub-gaussian noise and $\alpha = 2$ for sub-exponential noise). We also define

$$r_{\text{orc},\alpha} := \nu (\log n)^{3/2} \frac{\big(\log(np)\big)^\alpha}{\sqrt{n}}, \tag{12}$$

where $\alpha \in [2, \infty)$ and $\nu, c \in (0, \infty)$ are constants depending on the distributions of inputs and noise. Now, we extend our results in Theorem 1 for heavy-tailed noise.

**Theorem 4** (Statistical Guarantees for Reasonable Stationary Points for Heavy-tailed Noise). *Under the first two parts of Assumption 1, any reasonable stationary point $(\widetilde{\gamma}, \widetilde{\Theta})$ of the objective function in equation 2 with $r \geq r_{\text{orc},\alpha}$ satisfies the risk bound*

$$\text{risk}[\widetilde{\gamma}, \widetilde{\Theta}] \leq \text{risk}[\gamma^*, \Theta^*] + 5r\sqrt{\log n} \tag{13}$$

*with a probability at least $1 - 1/n$. If $r = r_{\text{orc},\alpha}$, the bound becomes*

$$\text{risk}[\widetilde{\gamma}, \widetilde{\Theta}] \leq \text{risk}[\gamma^*, \Theta^*] + \nu(\log n)^2 \frac{\big(\log(np)\big)^\alpha}{\sqrt{n}}. \tag{14}$$

The above results show that our theories still hold under heavy tails; the bounds and the optimal tuning parameter (see Theorem 1) then simply entail a power of $\alpha$ (depending on the noise) for $\log(np)$. This is an important step forward, as usual inputs to neural networks (images, text, ...) are often very noisy.

## 8  Discussion

We have established statistical guarantees for approximate stationary points of regularized shallow linear neural networks. We have then extended our theories to shallow ReLU neural networks under the assumption over the first layer weight matrix. Despite being limited to shallow networks, our theory is a large step forward in four ways: 1. Several papers consider the existence or non-existence of critical points that are not global optima in linear neural networks under certain assumptions. In contrast, our theories apply regardless of whether such local minima or saddle points exist in the objective under consideration. 2. Our extensions to ReLU neural networks not only provide theoretical insights but also highlight the importance of effective initialization, such as near-identity initialization, for ReLU networks (Hardt & Ma, 2016). 3. While works like Bach & Chizat (2021) consider convergence of specific optimization algorithms in deep learning, our results are agnostic to the optimization algorithm and do not require infinite-width networks, making our findings more general. 4. And finally, our new statistical approach inspired by high-dimensional statistics is expected to spark further progress in the mathematical understanding of deep learning.

### Acknowledgments

J. Lederer and M. Taheri are grateful for partial funding by the Deutsche Forschungsgemeinschaft (DFG, German Research Foundation) under project numbers 541176257 and 520388526 (TRR391). The authors thank Ali Mohades for constructive discussions at various stages of the project. F. Xie is supported in part by the Guangdong Basic and Applied Basic Research Foundation (No. 2023A1515110469), the Guangdong Provincial Key Laboratory IRADS (No. 2022B1212010006), and the grant of Higher Education Enhancement Plan of "Rushing to the Top, Making Up Shortcomings and Strengthening Special Features" (No. 2025KTSCX186).

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

## A    Appendix: Example and auxiliary results

Here we provide an illustrative and simple example to clearly show the existence of sub-optimal critical points for the regularized objective functions (equation 2 and equation 10) with linear and ReLU activations.

**Example 1** (Existence of sub-optimal critical points for regularized shallow networks)**.** *Let consider a toy linear shallow neural network with just two neurons $(a_1, a_2)$, and consider the loss function $f_{(a_1, a_2)}(X) = \sum_{i=1}^{n}(a_1 a_2 x_i - y_i)^2/2 + |a_1| + |a_2|$. Then, we suppose two training samples $(x_1 = 2, y_1 = 2)$ and $(x_2 = 4, y_2 = 1)$ that makes the objective function $\min_{(a_1, a_2)} f_{(a_1, a_2)}(X)$ non-convex, including local and global minimum and saddle point. One can confirm that $A = (a_1 = 0, a_2 = 0)$ is a local min with $f_A = 2.5$, while $A' = (a_1 \approx 0.55, a_2 \approx 0.55)$ is a global min with $f_{A'} \approx 2.1$ (see the left panel of Figure 2). This simple example illustrates that there are critical points even for simple regularized linear neural networks that are not global optima in our considered setup. Note that if the optimization algorithm (for example gradient descent) starts with weight initialization close to zero, it is high likely that we stuck in the vicinity of the local min $(0, 0)$. A similar example also holds for ReLU networks (see the right panel of Figure 2).*

Here we provide more technical results that are used to prove our main theorems.

First, we derive a uniform bound on the absolute difference between $\nabla \mathrm{risk}_X[\gamma, \Theta]$ and $\nabla \mathrm{risk}[\gamma, \Theta]$ for linear shallow networks. We use the notation $\|\Theta\|_\infty := \max_{j \in \{1, \dots, w\}} \sum_{k=1}^{d} |\theta_{jk}|$.

**Lemma 2** (Uniform Bound on the Difference Between $\nabla \mathrm{risk}_X[\gamma, \Theta]$ and $\nabla \mathrm{risk}[\gamma, \Theta]$ for Linear Networks)**.** *Under the Assumption 1 it holds for each $t, \eta, \epsilon \in (0, \infty)$ and $\boldsymbol{\beta} \in \mathcal{C}_{\eta, \epsilon} := \{\boldsymbol{\beta} = \mathrm{vec}(\boldsymbol{\gamma}, \Theta) \in \mathbb{R}^p : \|\boldsymbol{\beta}^* - \boldsymbol{\beta}\|_1 \leq \eta$ and $\|\boldsymbol{\gamma}^\top \Theta - \boldsymbol{\gamma}^{*\top} \Theta^*\|_1 \leq \epsilon\}$ that*

$$\sup_{\boldsymbol{\beta} \in \mathcal{C}_{\eta, \epsilon}} \left| \left( \nabla \mathrm{risk}_X[\boldsymbol{\gamma}, \Theta] - \nabla \mathrm{risk}[\boldsymbol{\gamma}, \Theta] \right)^\top (\boldsymbol{\beta}^* - \boldsymbol{\beta}) \right| \leq 2t\eta \big( \eta + \max\{\|\boldsymbol{\gamma}^*\|_\infty, \|\Theta^*\|_\infty\} \big) (1 + \epsilon)$$

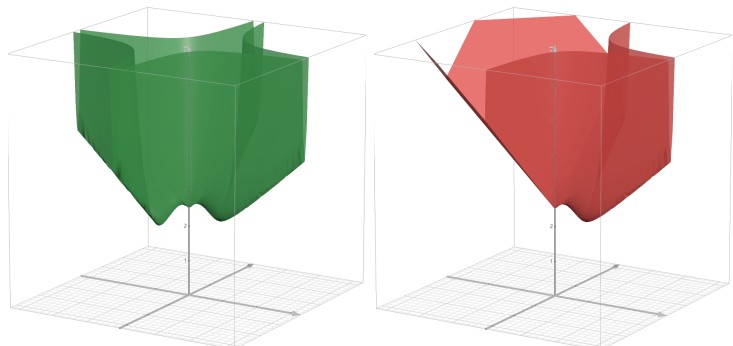

Figure 2: Non-convex objective function $\min_{(a_1,a_2)} f_{(a_1,a_2)}(X) = \sum_{i=1}^{n}(a_1\sigma(a_2 x_i) - y_i)^2/2 + |a_1| + |a_2|$ for two training samples $(x_1 = 2, y_1 = 2)$ and $(x_2 = 4, y_2 = 1)$ includes critical points that are not global optima. The left panel illustrates the objective for linear activation function, and the right panel shows the objective for the ReLU.

*with probability at least $1 - 4d^2p \exp(-\kappa n \min\{t^2/\nu^2, t/\nu\})$ with constants $\nu, \kappa \in (0, \infty)$ depending only on the distributions of the inputs and noise.*

The set $\mathcal{C}_{\eta,\epsilon}$ contains all parameters in a neighborhood of $\boldsymbol{\beta}^*$; in particular, the bound applies to $\boldsymbol{\beta}^* = \text{vec}(\boldsymbol{\gamma}^*, \Theta^*)$ itself—without any further assumption on $\boldsymbol{\beta}^*$. The lemma is the main ingredient of our proof for Lemma 1.

We also derive a uniform bound on the absolute difference between $\text{risk}_X[\boldsymbol{\gamma}, \Theta]$ and $\text{risk}[\boldsymbol{\gamma}, \Theta]$ (for linear shallow networks.)

**Lemma 3** (Uniform Bound on the Difference Between $\text{risk}_X[\boldsymbol{\gamma}, \Theta]$ and $\text{risk}[\boldsymbol{\gamma}, \Theta]$ for Linear Networks). *Suppose Assumption 1 is verified and that $\sup_{(\boldsymbol{\gamma},\Theta)\in\mathcal{B}} \|(\boldsymbol{\gamma}^{*\top}\Theta^* - \boldsymbol{\gamma}^\top\Theta)^2\|_\infty \leq \epsilon'$ for an $\epsilon' \in (0, \infty)$. Then, we have for each $t \in [0, \infty)$ that*

$$\sup_{(\boldsymbol{\gamma},\Theta)\in\mathcal{B}} \left|\text{risk}_X[\boldsymbol{\gamma}, \Theta] - \text{risk}[\boldsymbol{\gamma}, \Theta]\right| \leq t\left(1 + 4\epsilon' + 4\sqrt{\epsilon'}\right)$$

*with probability at least $1 - 18d^2 \exp(-\kappa n \min\{t^2/\nu^2, t/\nu\})$, with constants $\nu, \kappa \in (0, \infty)$ depending only on the distributions of the inputs and noise.*

Lemma 3 is the main ingredient of our proof of Theorem 2.

Then, we derive a lemma studying the invertibility of the line segment between two matrices. This lemma is employed in the proof of Theorem 1.

**Lemma 4** (Invertibility of the Line Segment Between Two Matrices). *Let's define $H(t) := (A + tC)(A + tC)^\top$ for $A, C \in \mathbb{R}^{w' \times d'}$ with $w' \leq d'$ and $t \in (0, 1)$, where $A$ has full (row) rank. Then, $H(t)$ is not invertible at most in finitely many $t \in (0, 1)$.*

Here, we differentiate the empirical risk $\text{risk}_X[\boldsymbol{\gamma}, \Theta]$ with respect to the parameters $\boldsymbol{\beta} = \text{vec}(\boldsymbol{\gamma}, \Theta)$. We use the indices $j, k$ for the first-order partial derivatives and indices $j', k'$ for the second-order partial derivatives. We use the notation $\mathbf{1}\{\cdot\}$ as an indicator function.

**Lemma 5** (First- and Second-Order Partial Derivatives of the Empirical Risk for Linear Networks). *It holds for each $j, j' \in \{1, \ldots, w\}$ and $k, k' \in \{1, \ldots, d\}$ that*

$$\frac{\partial}{\partial\gamma_j}\text{risk}_X[\boldsymbol{\gamma}, \Theta] = -\frac{2}{n}\sum_{i=1}^{n}\left((y_i - \boldsymbol{\gamma}^\top\Theta\boldsymbol{x}_i)(\Theta\boldsymbol{x}_i)_j\right),$$

$$\frac{\partial}{\partial\theta_{jk}}\text{risk}_X[\boldsymbol{\gamma}, \Theta] = -\frac{2}{n}\sum_{i=1}^{n}\left((y_i - \boldsymbol{\gamma}^\top\Theta\boldsymbol{x}_i)\gamma_j(\boldsymbol{x}_i)_k\right);$$

*and*

$$\frac{\partial^2}{\partial\gamma_{j'}\partial\gamma_j}\mathrm{risk}_X[\boldsymbol{\gamma},\Theta] = \frac{2}{n}\sum_{i=1}^{n}\big((\Theta\boldsymbol{x}_i)_{j'}(\Theta\boldsymbol{x}_i)_j\big),$$

$$\frac{\partial^2}{\partial\theta_{j'k'}\partial\theta_{jk}}\mathrm{risk}_X[\boldsymbol{\gamma},\Theta] = \frac{2}{n}\gamma_{j'}\gamma_j\sum_{i=1}^{n}\big((\boldsymbol{x}_i)_{k'}(\boldsymbol{x}_i)_k\big).$$

*Moreover, if $j' = j$, it holds that*

$$\frac{\partial^2}{\partial\theta_{jk'}\partial\gamma_j}\mathrm{risk}_X[\boldsymbol{\gamma},\Theta] = \frac{2}{n}\sum_{i=1}^{n}\Big(\gamma_j(\boldsymbol{x}_i)_{k'}(\Theta\boldsymbol{x}_i)_j - (y_i - \boldsymbol{\gamma}^\top\Theta\boldsymbol{x}_i)(\boldsymbol{x}_i)_{k'}\Big)$$

*and*

$$\frac{\partial^2}{\partial\gamma_j\partial\theta_{jk}}\mathrm{risk}_X[\boldsymbol{\gamma},\Theta] = \frac{2}{n}\sum_{i=1}^{n}\Big(\gamma_j(\boldsymbol{x}_i)_k(\Theta\boldsymbol{x}_i)_j - (y_i - \boldsymbol{\gamma}^\top\Theta\boldsymbol{x}_i)(\boldsymbol{x}_i)_k\Big),$$

*and if $j' \neq j$, it holds that*

$$\frac{\partial^2}{\partial\gamma_{j'}\partial\theta_{jk}}\mathrm{risk}_X[\boldsymbol{\gamma},\Theta] = \frac{2}{n}\gamma_j\sum_{i=1}^{n}(\boldsymbol{x}_i)_k(\Theta\boldsymbol{x}_i)_{j'}$$

*and*

$$\frac{\partial^2}{\partial\theta_{j'k'}\partial\gamma_j}\mathrm{risk}_X[\boldsymbol{\gamma},\Theta] = \frac{2}{n}\gamma_{j'}\sum_{i=1}^{n}(\boldsymbol{x}_i)_{k'}(\Theta\boldsymbol{x}_i)_j.$$

These derivatives are basic tools for us given that we work with stationary points.

The next result is essentially a population version of the partial derivatives in Lemma 5, that is, sums are replaced by expectations.

**Lemma 6** (First- and Second-Order Partial Derivatives of the Population Risk for Linear Networks). *It holds for each $j, j' \in \{1, \ldots, w\}$ and $k, k' \in \{1, \ldots, d\}$ that*

$$\frac{\partial}{\partial\gamma_j}\mathrm{risk}[\boldsymbol{\gamma},\Theta] = -2\mathbb{E}_{(\boldsymbol{x},y)}\big[(y - \boldsymbol{\gamma}^\top\Theta\boldsymbol{x})(\Theta\boldsymbol{x})_j\big],$$

$$\frac{\partial}{\partial\theta_{jk}}\mathrm{risk}[\boldsymbol{\gamma},\Theta] = -2\mathbb{E}_{(\boldsymbol{x},y)}\big[(y - \boldsymbol{\gamma}^\top\Theta\boldsymbol{x})\gamma_j(\boldsymbol{x})_k\big];$$

*and*

$$\frac{\partial^2}{\partial\gamma_{j'}\partial\gamma_j}\mathrm{risk}[\boldsymbol{\gamma},\Theta] = 2\,\mathbb{E}_{(\boldsymbol{x},y)}\big[(\Theta\boldsymbol{x})_{j'}(\Theta\boldsymbol{x})_j\big],$$

$$\frac{\partial^2}{\partial\theta_{j'k'}\partial\theta_{jk}}\mathrm{risk}[\boldsymbol{\gamma},\Theta] = 2\gamma_{j'}\gamma_j\mathbb{E}_{(\boldsymbol{x},y)}\big[(\boldsymbol{x})_{k'}(\boldsymbol{x})_k\big].$$

*Moreover, if $j' = j$, it holds that*

$$\frac{\partial^2}{\partial\theta_{jk'}\partial\gamma_j}\mathrm{risk}[\boldsymbol{\gamma},\Theta] = 2\mathbb{E}_{(\boldsymbol{x},y)}\big[\gamma_j(\boldsymbol{x})_{k'}(\Theta\boldsymbol{x})_j - (y - \boldsymbol{\gamma}^\top\Theta\boldsymbol{x})(\boldsymbol{x})_{k'}\big]$$

*and*

$$\frac{\partial^2}{\partial\gamma_j\partial\theta_{jk}}\mathrm{risk}[\boldsymbol{\gamma},\Theta] = 2\mathbb{E}_{(\boldsymbol{x},y)}\big[\gamma_j(\boldsymbol{x})_k(\Theta\boldsymbol{x})_j - (y - \boldsymbol{\gamma}^\top\Theta\boldsymbol{x})(\boldsymbol{x})_k\big],$$

*and if $j' \neq j$, it holds that*

$$\frac{\partial^2}{\partial\gamma_{j'}\partial\theta_{jk}}\mathrm{risk}[\boldsymbol{\gamma},\Theta] = 2\gamma_j\mathbb{E}_{(\boldsymbol{x},y)}\big[(\boldsymbol{x})_k(\Theta\boldsymbol{x})_{j'}\big]$$

*and*

$$\frac{\partial^2}{\partial\theta_{j'k'}\partial\gamma_j}\mathrm{risk}[\boldsymbol{\gamma},\Theta] = 2\gamma_{j'}\mathbb{E}_{(\boldsymbol{x},y)}\big[(\boldsymbol{x})_{k'}(\Theta\boldsymbol{x})_j\big].$$

We use these results in the proofs of Theorem 1 and Proposition 1.

**Lemma 7** (First- and Second-Order Subdifferentials of the Empirical Risk for ReLU Networks)**.** *It holds for each $j, j' \in \{1, \dots, w\}$ and $k, k' \in \{1, \dots, d\}$ that*

$$\frac{\partial}{\partial \gamma_j} \mathrm{risk}_X[\boldsymbol{\gamma}, \Theta] = -\frac{2}{n} \sum_{i=1}^n \left( \left( y_i - \boldsymbol{\gamma}^\top \boldsymbol{\sigma}(\Theta \boldsymbol{x}_i) \right) \boldsymbol{\sigma}(\Theta \boldsymbol{x}_i)_j \right),$$

$$\frac{\partial^2}{\partial \gamma_{j'} \partial \gamma_j} \mathrm{risk}_X[\boldsymbol{\gamma}, \Theta] = \frac{2}{n} \sum_{i=1}^n \left( (\boldsymbol{\sigma}(\Theta \boldsymbol{x}_i)_{j'} \boldsymbol{\sigma}(\Theta \boldsymbol{x}_i)_j \right).$$

*And*

$$\frac{\partial}{\partial \theta_{jk}} \mathrm{risk}_X[\boldsymbol{\gamma}, \Theta] = -\frac{2}{n} \sum_{i=1}^n \left( \left( y_i - \boldsymbol{\gamma}^\top \boldsymbol{\sigma}(\Theta \boldsymbol{x}_i) \right) \gamma_j (\boldsymbol{x}_i)_k \kappa(\boldsymbol{x}_i, j) \right)$$

*with*

$$\kappa(\boldsymbol{x}_i, j) := \begin{cases} \mathbf{1}\{(\Theta \boldsymbol{x}_i)_j > 0\}, & \textit{if } (\Theta \boldsymbol{x}_i)_j \neq 0. \\ [0, 1], & \textit{otherwise.} \end{cases}$$

*If $j = j'$ and $\exists\, i \in \{1, \dots, n\}$ with $(\Theta \boldsymbol{x}_i)_j = 0$ then, $\partial^2 \mathrm{risk}_X[\boldsymbol{\gamma}, \Theta]/\partial \theta_{j'k'} \partial \theta_{jk}$ doesn't exists otherwise,*

$$\frac{\partial^2}{\partial \theta_{j'k'} \partial \theta_{jk}} \mathrm{risk}_X[\boldsymbol{\gamma}, \Theta] = \frac{2}{n} \gamma_j \gamma_{j'} \sum_{i=1}^n \left( (\boldsymbol{x}_i)_{k'} (\boldsymbol{x}_i)_k \kappa(\boldsymbol{x}_i, j') \kappa(\boldsymbol{x}_i, j) \right).$$

*For $j' = j$*

$$\frac{\partial^2}{\partial \theta_{jk'} \partial \gamma_j} \mathrm{risk}_X[\boldsymbol{\gamma}, \Theta] = \frac{2}{n} \sum_{i=1}^n \left( \gamma_j (\boldsymbol{x}_i)_{k'} \boldsymbol{\sigma}(\Theta \boldsymbol{x}_i)_j \kappa(\boldsymbol{x}_i, j) - \left( y_i - \boldsymbol{\gamma}^\top \boldsymbol{\sigma}(\Theta \boldsymbol{x}_i) \right) (\boldsymbol{x}_i)_{k'} \kappa(\boldsymbol{x}_i, j) \right)$$

*and if $j' \neq j$*

$$\frac{\partial^2}{\partial \theta_{j'k'} \partial \gamma_j} \mathrm{risk}_X[\boldsymbol{\gamma}, \Theta] = \frac{2}{n} \gamma_{j'} \sum_{i=1}^n \left( (\boldsymbol{x}_i)_{k'} \boldsymbol{\sigma}(\Theta \boldsymbol{x}_i)_j \kappa(\boldsymbol{x}_i, j') \right).$$

The next result is essentially a population version of the subdifferentials in Lemma 7, that is, sums are replaced by expectations.

**Lemma 8** (Second-Order Subdifferentials of the Population Risk for ReLU Networks)**.** *It holds for each $j, j' \in \{1, \dots, w\}$ and $k, k' \in \{1, \dots, d\}$*

$$\frac{\partial^2}{\partial \gamma_{j'} \partial \gamma_j} \mathrm{risk}[\boldsymbol{\gamma}, \Theta] = 2 \mathbb{E}_{\boldsymbol{x}} \left[ (\Theta \boldsymbol{x})_{j'} (\Theta \boldsymbol{x})_j \mathbf{1}\{(\Theta \boldsymbol{x})_{j'} > 0, (\Theta \boldsymbol{x})_j > 0\} \right].$$

*If $j = j'$ and $\exists\, \boldsymbol{x}$ with $(\Theta \boldsymbol{x})_j = 0$ then, $\partial^2 \mathrm{risk}[\boldsymbol{\gamma}, \Theta]/\partial \theta_{j'k'} \partial \theta_{jk}$ doesn't exists otherwise,*

$$\frac{\partial^2}{\partial \theta_{j'k'} \partial \theta_{jk}} \mathrm{risk}[\boldsymbol{\gamma}, \Theta] = 2 \gamma_j \gamma_{j'} \mathbb{E}_{\boldsymbol{x}} \left[ (\boldsymbol{x})_{k'} (\boldsymbol{x})_k \kappa(\boldsymbol{x}, j') \kappa(\boldsymbol{x}, j) \right],$$

*where*

$$\kappa(\boldsymbol{x}, j) := \begin{cases} \mathbf{1}\{(\Theta \boldsymbol{x})_j > 0\}, & \textit{if } (\Theta \boldsymbol{x})_j \neq 0. \\ [0, 1], & \textit{otherwise.} \end{cases}$$

*For $j' = j$, it holds that*

$$\frac{\partial^2}{\partial \theta_{jk'} \partial \gamma_j} \mathrm{risk}[\boldsymbol{\gamma}, \Theta] = 2 \mathbb{E}_{\boldsymbol{x}, \boldsymbol{y}} [\gamma_j (\boldsymbol{x})_{k'} \boldsymbol{\sigma}(\Theta \boldsymbol{x})_j \kappa(\boldsymbol{x}, j) - \left( \boldsymbol{y} - \boldsymbol{\gamma}^\top \boldsymbol{\sigma}(\Theta \boldsymbol{x}) \right) (\boldsymbol{x})_{k'} \kappa(\boldsymbol{x}, j)],$$

*and if $j' \neq j$, it holds that*

$$\frac{\partial^2}{\partial \theta_{j'k'} \partial \gamma_j} \mathrm{risk}[\boldsymbol{\gamma}, \Theta] = 2 \gamma_{j'} \mathbb{E}_{\boldsymbol{x}} [(\boldsymbol{x})_{k'} \boldsymbol{\sigma}(\Theta \boldsymbol{x})_j \kappa(\boldsymbol{x}, j')].$$

**Lemma 9** (Expected Value of the Joint Density Over Two Half Spaces). *For two mean-zero Gaussian random variables $Z$ and $Z'$ with unit variance and with small enough $\rho = \mathbb{E}[ZZ']$ ($|\rho| \leq 0.2$) we have*

$$\mathbb{E}[ZZ'\mathbf{1}\{Z > 0\}\mathbf{1}\{Z' > 0\}] \approx \frac{1}{2\pi}\left(1 + \frac{\pi\rho}{2} - \frac{3\rho^2}{2}\right).$$

The proof is based on computing the integral using the joint density. We then apply a change of variables and switch to polar coordinates, evaluate the radial integral, and approximate the angular integral via a binomial expansion under the assumption that the correlation is small. We skip the detailed proof as it just involves linear algebra.

## B    Appendix: Proofs for shallow linear networks

Here, we provide the proofs of our main claims for linear networks.

### B.1    Proof of Theorem 1

*Proof.* The proof approach is based on Taylor's theorem and the definition of stationary points.

Let's introduce some notations: We use the notation $\boldsymbol{\gamma}^\top \Theta_A \bar{\boldsymbol{x}} := \boldsymbol{\gamma}^\top[\Theta, A]\bar{\boldsymbol{x}}$ to generate an extended network indexed by $(\boldsymbol{\gamma}, \Theta_A)$ with $\bar{\boldsymbol{x}} := (\boldsymbol{x}^\top, \tilde{\boldsymbol{x}}^\top)^\top \in \mathbb{R}^{d+w-1}$, $\tilde{\boldsymbol{x}}$ having the same distribution as $\boldsymbol{x}$, and $A = [\boldsymbol{v}_1, \ldots, \boldsymbol{v}_{w-1}] \in \mathbb{R}^{w \times w-1}$, with $\boldsymbol{v}_1, \ldots, \boldsymbol{v}_{w-1} \in \mathbb{R}^w$, is a matrix whose columns are basis of $\mathbb{R}^{w-1}$ such that $\boldsymbol{\gamma}^\top \boldsymbol{v}_1 = \cdots = \boldsymbol{\gamma}^\top \boldsymbol{v}_{w-1} = 0$. It means, the input's dimension of the network is extended from $d$ to $d + w - 1$ and so the inner-layer matrix need also to be extended from $\Theta \in \mathbb{R}^{w \times d}$ to $[\Theta, A] \in \mathbb{R}^{w \times (d+w-1)}$. We also use the notation $\boldsymbol{\gamma}_{\boldsymbol{\alpha}}^\top \Theta_{\boldsymbol{\alpha},A} \bar{\boldsymbol{x}}$ to make an extended network that is also rescaled across the layers by a suitable $\boldsymbol{\alpha}$. Note that the notation $\Theta_{\boldsymbol{\alpha},A}$ is equivalent with $(\Theta_A)_{\boldsymbol{\alpha}}$, both means we rescale a matrix $\Theta_A \in \mathbb{R}^{w \times (d+w-1)}$ with a vector $\boldsymbol{\alpha} \in \mathbb{R}^w$ (see more details about rescaled networks in Section 6). Using the above definitions, it is easy to see that $\boldsymbol{\gamma}_{\boldsymbol{\alpha}}^\top \Theta_{\boldsymbol{\alpha},A} \bar{\boldsymbol{x}} = \boldsymbol{\gamma}^\top \Theta \boldsymbol{x}$, which means, the output of the extended and rescaled network is the same as the original network (using the definition of $A$ and rescaled weights). In other words, we have a network that is first extended and then rescaled while the output of the network is still the same as the original one. We use the notation $\mathrm{risk}[\boldsymbol{\gamma}_{\boldsymbol{\alpha}}, \Theta_{\boldsymbol{\alpha},A}] := \mathbb{E}_{(\bar{\boldsymbol{x}},y)}[(y - \boldsymbol{\gamma}_{\boldsymbol{\alpha}}^\top \Theta_{\boldsymbol{\alpha},A}\bar{\boldsymbol{x}})^2]$ to compute the population risk in an extended and rescaled network. We also define $p' := w + w \cdot (d + w - 1)$ as the effective dimension of the extended network.

Now, let's start the proof by writing a second-order Taylor expansion of $\mathrm{risk}[\boldsymbol{\gamma}^*_{\boldsymbol{\alpha}}, \Theta^*_{\boldsymbol{\alpha},A'}]$ (the risk in an extended and rescaled version of the target with $\boldsymbol{\beta}^*_{\boldsymbol{\alpha},A'} = \mathrm{vec}(\boldsymbol{\gamma}^*_{\boldsymbol{\alpha}}, \Theta^*_{\boldsymbol{\alpha},A'}) \in \mathbb{R}^{p'}$) around an extended and rescaled version of a reasonable stationary $\widetilde{\boldsymbol{\beta}}_{\boldsymbol{\alpha},A} = \mathrm{vec}(\widetilde{\boldsymbol{\gamma}}_{\boldsymbol{\alpha}}, \widetilde{\Theta}_{\boldsymbol{\alpha},A}) \in \mathbb{R}^{p'}$ with suitable $\boldsymbol{\alpha} \in \mathbb{R}^w$ and $A, A' \in \mathbb{R}^{w \times w-1}$ (we see later how to assign suitable value for $\boldsymbol{\alpha}$) to get

$$\mathrm{risk}[\boldsymbol{\gamma}^*_{\boldsymbol{\alpha}}, \Theta^*_{\boldsymbol{\alpha},A'}] = \mathrm{risk}[\widetilde{\boldsymbol{\gamma}}_{\boldsymbol{\alpha}}, \widetilde{\Theta}_{\boldsymbol{\alpha},A}] + \nabla \mathrm{risk}[\widetilde{\boldsymbol{\gamma}}_{\boldsymbol{\alpha}}, \widetilde{\Theta}_{\boldsymbol{\alpha},A}]^\top (\boldsymbol{\beta}^*_{\boldsymbol{\alpha},A'} - \widetilde{\boldsymbol{\beta}}_{\boldsymbol{\alpha},A})$$
$$+ \frac{1}{2}(\boldsymbol{\beta}^*_{\boldsymbol{\alpha},A'} - \widetilde{\boldsymbol{\beta}}_{\boldsymbol{\alpha},A})^\top \nabla^2 \mathrm{risk}[\widetilde{\boldsymbol{\gamma}}_{\boldsymbol{\alpha}} + t(\boldsymbol{\gamma}^*_{\boldsymbol{\alpha}} - \widetilde{\boldsymbol{\gamma}}_{\boldsymbol{\alpha}}), \widetilde{\Theta}_{\boldsymbol{\alpha},A} + t(\Theta^*_{\boldsymbol{\alpha},A'} - \widetilde{\Theta}_{\boldsymbol{\alpha},A})]$$
$$(\boldsymbol{\beta}^*_{\boldsymbol{\alpha},A'} - \widetilde{\boldsymbol{\beta}}_{\boldsymbol{\alpha},A})$$

for some $t \in (0, 1)$ (Bertsekas et al., 2003, Proposition 1.1.13.a), where we use the notation $\nabla \mathrm{risk}[\widetilde{\boldsymbol{\gamma}}_{\boldsymbol{\alpha}}, \widetilde{\Theta}_{\boldsymbol{\alpha},A}] \in \mathbb{R}^{p'}$ and $\nabla^2 \mathrm{risk}[\widetilde{\boldsymbol{\gamma}}_{\boldsymbol{\alpha}}, \widetilde{\Theta}_{\boldsymbol{\alpha},A}] \in \mathbb{R}^{p' \times p'}$ to collect the first and second order partial derivatives of $\mathrm{risk}[\widetilde{\boldsymbol{\gamma}}_{\boldsymbol{\alpha}}, \widetilde{\Theta}_{\boldsymbol{\alpha},A}]$ with respect to the $\widetilde{\boldsymbol{\beta}}_{\boldsymbol{\alpha},A}$, respectively (note that we have no assumption on $(\boldsymbol{\gamma}^*_{\boldsymbol{\alpha}}, \Theta^*_{\boldsymbol{\alpha},A'})$ nor $(\widetilde{\boldsymbol{\gamma}}_{\boldsymbol{\alpha}}, \widetilde{\Theta}_{\boldsymbol{\alpha},A})$ to have bounded norms).

Then, we employ the property of extended and rescaled networks that is $\mathrm{risk}[\widetilde{\boldsymbol{\gamma}}_{\boldsymbol{\alpha}}, \widetilde{\Theta}_{\boldsymbol{\alpha},A}] = \mathrm{risk}[\widetilde{\boldsymbol{\gamma}}, \widetilde{\Theta}]$ and $\mathrm{risk}[\boldsymbol{\gamma}^*_{\boldsymbol{\alpha}}, \Theta^*_{\boldsymbol{\alpha},A'}] = \mathrm{risk}[\boldsymbol{\gamma}^*, \Theta^*]$, and use the shorthand notation

$$m := (\boldsymbol{\beta}^*_{\boldsymbol{\alpha},A'} - \widetilde{\boldsymbol{\beta}}_{\boldsymbol{\alpha},A})^\top \nabla^2 \mathrm{risk}[\widetilde{\boldsymbol{\gamma}}_{\boldsymbol{\alpha}} + t(\boldsymbol{\gamma}^*_{\boldsymbol{\alpha}} - \widetilde{\boldsymbol{\gamma}}_{\boldsymbol{\alpha}}), \widetilde{\Theta}_{\boldsymbol{\alpha},A} + t(\Theta^*_{\boldsymbol{\alpha},A'} - \widetilde{\Theta}_{\boldsymbol{\alpha},A})](\boldsymbol{\beta}^*_{\boldsymbol{\alpha},A'} - \widetilde{\boldsymbol{\beta}}_{\boldsymbol{\alpha},A})$$

to obtain

$$\mathrm{risk}[\boldsymbol{\gamma}^*, \Theta^*] = \mathrm{risk}[\widetilde{\boldsymbol{\gamma}}, \widetilde{\Theta}] + \nabla \mathrm{risk}[\widetilde{\boldsymbol{\gamma}}_{\boldsymbol{\alpha}}, \widetilde{\Theta}_{\boldsymbol{\alpha},A}]^\top (\boldsymbol{\beta}^*_{\boldsymbol{\alpha},A'} - \widetilde{\boldsymbol{\beta}}_{\boldsymbol{\alpha},A}) + \frac{1}{2}m.$$

Now, we are motivated to show that $\nabla\text{risk}[\widetilde{\boldsymbol{\gamma}}_{\boldsymbol{\alpha}}, \widetilde{\Theta}_{\boldsymbol{\alpha},A}]^\top(\boldsymbol{\beta}^*_{\boldsymbol{\alpha},A'} - \widetilde{\boldsymbol{\beta}}_{\boldsymbol{\alpha},A}) = \nabla\text{risk}[\widetilde{\boldsymbol{\gamma}}, \widetilde{\Theta}]^\top(\boldsymbol{\beta}^* - \widetilde{\boldsymbol{\beta}})$. To do so, we use 1. our Lemma 6 (for an extended and rescaled network), 2. the property of extended and rescaled networks, 3. linearity of expectations, 4. our assumption on $\bar{\boldsymbol{x}}$, 5. some rewriting, 6. linearity of expectations, 7. our assumption on $\bar{\boldsymbol{x}}$ (let's recall that $\bar{\boldsymbol{x}} = (\boldsymbol{x}^\top, \tilde{\boldsymbol{x}}^\top)^\top \in \mathbb{R}^{d+w-1}$ with $\tilde{\boldsymbol{x}}$ having the same distribution as $\boldsymbol{x}$ and independent of $\boldsymbol{x}$) that makes the second expectation zero, and 8. some rewriting to obtain that

$$
\begin{aligned}
\frac{\partial}{\partial(\widetilde{\boldsymbol{\gamma}}_{\boldsymbol{\alpha}})_j}\text{risk}[\widetilde{\boldsymbol{\gamma}}_{\boldsymbol{\alpha}}, \widetilde{\Theta}_{\boldsymbol{\alpha},A}] &= -2\mathbb{E}_{(\bar{\boldsymbol{x}},y)}\Big[\big(y - \widetilde{\boldsymbol{\gamma}}_{\boldsymbol{\alpha}}{}^\top\widetilde{\Theta}_{\boldsymbol{\alpha},A}\bar{\boldsymbol{x}}\big)\big(\widetilde{\Theta}_{\boldsymbol{\alpha},A}\bar{\boldsymbol{x}}\big)_j\Big] \\
&= -2\mathbb{E}_{(\bar{\boldsymbol{x}},y)}\Big[\big(y - \widetilde{\boldsymbol{\gamma}}^\top\widetilde{\Theta}\boldsymbol{x}\big)\big(\widetilde{\Theta}_{\boldsymbol{\alpha},A}\bar{\boldsymbol{x}}\big)_j\Big] \\
&= -2\mathbb{E}_{(\bar{\boldsymbol{x}},y)}\Big[y\big(\widetilde{\Theta}_{\boldsymbol{\alpha},A}\bar{\boldsymbol{x}}\big)_j\Big] + 2\mathbb{E}_{(\bar{\boldsymbol{x}},y)}\Big[\big(\widetilde{\boldsymbol{\gamma}}^\top\widetilde{\Theta}\boldsymbol{x}\big)\big(\widetilde{\Theta}_{\boldsymbol{\alpha},A}\bar{\boldsymbol{x}}\big)_j\Big] \\
&= -2\mathbb{E}_{(\boldsymbol{x},y)}\Big[y\big(\widetilde{\Theta}_{\boldsymbol{\alpha}}\boldsymbol{x}\big)_j\Big] + 2\mathbb{E}_{(\bar{\boldsymbol{x}},y)}\Big[\big(\widetilde{\boldsymbol{\gamma}}^\top\widetilde{\Theta}\boldsymbol{x}\big)\big(\widetilde{\Theta}_{\boldsymbol{\alpha},A}\bar{\boldsymbol{x}}\big)_j\Big] \\
&= -2\mathbb{E}_{(\boldsymbol{x},y)}\Big[y\big(\widetilde{\Theta}_{\boldsymbol{\alpha}}\boldsymbol{x}\big)_j\Big] + 2\mathbb{E}_{(\bar{\boldsymbol{x}},y)}\Big[\big(\widetilde{\boldsymbol{\gamma}}^\top\widetilde{\Theta}\boldsymbol{x}\big)\sum_{k=1}^{d+w-1}\big(\widetilde{\Theta}_{\boldsymbol{\alpha},A}\big)_{jk}(\bar{\boldsymbol{x}})_k\Big] \\
&= -2\mathbb{E}_{(\boldsymbol{x},y)}\Big[y\big(\widetilde{\Theta}_{\boldsymbol{\alpha}}\boldsymbol{x}\big)_j\Big] + 2\mathbb{E}_{(\bar{\boldsymbol{x}},y)}\Big[\big(\widetilde{\boldsymbol{\gamma}}^\top\widetilde{\Theta}\boldsymbol{x}\big)\sum_{k=1}^{d}\big(\widetilde{\Theta}_{\boldsymbol{\alpha},A}\big)_{jk}(\bar{\boldsymbol{x}})_k\Big] \\
&\quad + 2\mathbb{E}_{(\bar{\boldsymbol{x}},y)}\Big[\big(\widetilde{\boldsymbol{\gamma}}^\top\widetilde{\Theta}\boldsymbol{x}\big)\sum_{k=d+1}^{d+w-1}\big(\widetilde{\Theta}_{\boldsymbol{\alpha},A}\big)_{jk}(\bar{\boldsymbol{x}})_k\Big] \\
&= -2\mathbb{E}_{(\boldsymbol{x},y)}\Big[y\big(\widetilde{\Theta}_{\boldsymbol{\alpha}}\boldsymbol{x}\big)_j\Big] + 2\mathbb{E}_{(\boldsymbol{x},y)}\Big[\big(\widetilde{\boldsymbol{\gamma}}^\top\widetilde{\Theta}\boldsymbol{x}\big)\sum_{k=1}^{d}\big(\widetilde{\Theta}_{\boldsymbol{\alpha}}\big)_{jk}(\boldsymbol{x})_k\Big] \\
&= -2\mathbb{E}_{(\boldsymbol{x},y)}\Big[y\big(\widetilde{\Theta}_{\boldsymbol{\alpha}}\boldsymbol{x}\big)_j\Big] + 2\mathbb{E}_{(\boldsymbol{x},y)}\Big[\big(\widetilde{\boldsymbol{\gamma}}^\top\widetilde{\Theta}\boldsymbol{x}\big)\big(\widetilde{\Theta}_{\boldsymbol{\alpha}}\boldsymbol{x}\big)_j\Big].
\end{aligned}
$$

Then, we 1. imply our result above for all $j \in \{1, \ldots, w\}$, 2. use the definition of rescaled parameters and linearity of expectations to cancel $\boldsymbol{\alpha}$'s, and 3. use our results in Lemma 6 to obtain

$$
\begin{aligned}
\Big(\frac{\partial}{\partial\widetilde{\boldsymbol{\gamma}}_{\boldsymbol{\alpha}}}\text{risk}[\widetilde{\boldsymbol{\gamma}}_{\boldsymbol{\alpha}}, \widetilde{\Theta}_{\boldsymbol{\alpha},A}]\Big)^\top(\boldsymbol{\gamma}^*_{\boldsymbol{\alpha}} - \widetilde{\boldsymbol{\gamma}}_{\boldsymbol{\alpha}}) &= 2\Big(-\mathbb{E}_{(\boldsymbol{x},y)}\Big[y\big(\widetilde{\Theta}_{\boldsymbol{\alpha}}\boldsymbol{x}\big)_j\Big] + \mathbb{E}_{(\boldsymbol{x},y)}\Big[\big(\widetilde{\boldsymbol{\gamma}}^\top\widetilde{\Theta}\boldsymbol{x}\big)\big(\widetilde{\Theta}_{\boldsymbol{\alpha}}\boldsymbol{x}\big)\Big]\Big)^\top(\boldsymbol{\gamma}^*_{\boldsymbol{\alpha}} - \widetilde{\boldsymbol{\gamma}}_{\boldsymbol{\alpha}}) \\
&= 2\Big(-\mathbb{E}_{(\boldsymbol{x},y)}\Big[y\big(\widetilde{\Theta}\boldsymbol{x}\big)_j\Big] + \mathbb{E}_{(\boldsymbol{x},y)}\Big[\big(\widetilde{\boldsymbol{\gamma}}^\top\widetilde{\Theta}\boldsymbol{x}\big)\big(\widetilde{\Theta}\boldsymbol{x}\big)\Big]\Big)^\top(\boldsymbol{\gamma}^* - \widetilde{\boldsymbol{\gamma}}) \\
&= \Big(\frac{\partial}{\partial\widetilde{\boldsymbol{\gamma}}}\text{risk}[\widetilde{\boldsymbol{\gamma}}, \widetilde{\Theta}]\Big)^\top(\boldsymbol{\gamma}^* - \widetilde{\boldsymbol{\gamma}}).
\end{aligned}
$$

Implying a similar argument as above for all partial derivatives, we conclude that $\nabla\text{risk}[\widetilde{\boldsymbol{\gamma}}_{\boldsymbol{\alpha}}, \widetilde{\Theta}_{\boldsymbol{\alpha},A}]^\top(\boldsymbol{\beta}^*_{\boldsymbol{\alpha},A'} - \widetilde{\boldsymbol{\beta}}_{\boldsymbol{\alpha},A}) = \nabla\text{risk}[\widetilde{\boldsymbol{\gamma}}, \widetilde{\Theta}]^\top(\boldsymbol{\beta}^* - \widetilde{\boldsymbol{\beta}})$ (we omit the detailed proof). Tabulating this observation in the earlier display we obtain

$$
\text{risk}[\boldsymbol{\gamma}^*, \Theta^*] = \text{risk}[\widetilde{\boldsymbol{\gamma}}, \widetilde{\Theta}] + \nabla\text{risk}[\widetilde{\boldsymbol{\gamma}}, \widetilde{\Theta}]^\top(\boldsymbol{\beta}^* - \widetilde{\boldsymbol{\beta}}) + \frac{1}{2}m.
$$

Rearranging the display above we obtain

$$
-\nabla\text{risk}[\widetilde{\boldsymbol{\gamma}}, \widetilde{\Theta}]^\top(\boldsymbol{\beta}^* - \widetilde{\boldsymbol{\beta}}) = \text{risk}[\widetilde{\boldsymbol{\gamma}}, \widetilde{\Theta}] - \text{risk}[\boldsymbol{\gamma}^*, \Theta^*] + \frac{1}{2}m.
$$

Now, let's recall the definition of stationary points in equation 3 which implies

$$
\nabla\text{risk}_X[\widetilde{\boldsymbol{\gamma}}, \widetilde{\Theta}]^\top(\boldsymbol{\beta}^* - \widetilde{\boldsymbol{\beta}}) + r\tilde{\boldsymbol{z}}^\top(\boldsymbol{\beta}^* - \widetilde{\boldsymbol{\beta}}) \geq 0.
$$

We 1. rearrange above inequality and expand the bracket, 2. use Hölder's inequality and the fact that $\tilde{\boldsymbol{z}}^\top \widetilde{\boldsymbol{\beta}} = \|\widetilde{\boldsymbol{\beta}}\|_1$ (recall that $\tilde{\boldsymbol{z}} \in \partial\|\widetilde{\boldsymbol{\beta}}\|_1$), and 3. use $\|\tilde{\boldsymbol{z}}\|_\infty \leq 1$ to obtain

$$-\nabla\mathrm{risk}_X[\widetilde{\boldsymbol{\gamma}}, \widetilde{\Theta}]^\top(\boldsymbol{\beta}^* - \widetilde{\boldsymbol{\beta}}) \leq r\tilde{\boldsymbol{z}}^\top\boldsymbol{\beta}^* - r\tilde{\boldsymbol{z}}^\top\widetilde{\boldsymbol{\beta}}$$
$$\leq r\|\tilde{\boldsymbol{z}}\|_\infty\|\boldsymbol{\beta}^*\|_1 - r\|\widetilde{\boldsymbol{\beta}}\|_1$$
$$\leq r\|\boldsymbol{\beta}^*\|_1 - r\|\widetilde{\boldsymbol{\beta}}\|_1\,,$$

which rearranging implies

$$\nabla\mathrm{risk}_X[\widetilde{\boldsymbol{\gamma}}, \widetilde{\Theta}]^\top(\boldsymbol{\beta}^* - \widetilde{\boldsymbol{\beta}}) + r\|\boldsymbol{\beta}^*\|_1 - r\|\widetilde{\boldsymbol{\beta}}\|_1 \geq 0\,.$$

The display above demonstrates the positivity of the terms on its left-hand side, enabling us to obtain

$$-\nabla\mathrm{risk}[\widetilde{\boldsymbol{\gamma}}, \widetilde{\Theta}]^\top(\boldsymbol{\beta}^* - \widetilde{\boldsymbol{\beta}}) \leq -\nabla\mathrm{risk}[\widetilde{\boldsymbol{\gamma}}, \widetilde{\Theta}]^\top(\boldsymbol{\beta}^* - \widetilde{\boldsymbol{\beta}}) + \nabla\mathrm{risk}_X[\widetilde{\boldsymbol{\gamma}}, \widetilde{\Theta}]^\top(\boldsymbol{\beta}^* - \widetilde{\boldsymbol{\beta}}) + r\|\boldsymbol{\beta}^*\|_1 - r\|\widetilde{\boldsymbol{\beta}}\|_1\,,$$

that is,

$$-\nabla\mathrm{risk}[\widetilde{\boldsymbol{\gamma}}, \widetilde{\Theta}]^\top(\boldsymbol{\beta}^* - \widetilde{\boldsymbol{\beta}}) \leq \left(\nabla\mathrm{risk}_X[\widetilde{\boldsymbol{\gamma}}, \widetilde{\Theta}] - \nabla\mathrm{risk}[\widetilde{\boldsymbol{\gamma}}, \widetilde{\Theta}]\right)^\top(\boldsymbol{\beta}^* - \widetilde{\boldsymbol{\beta}}) + r\|\boldsymbol{\beta}^*\|_1 - r\|\widetilde{\boldsymbol{\beta}}\|_1\,.$$

Now, let's use our display earlier (obtained by Taylor expansion) to rewrite the left-hand side of the display above as

$$\mathrm{risk}[\widetilde{\boldsymbol{\gamma}}, \widetilde{\Theta}] - \mathrm{risk}[\boldsymbol{\gamma}^*, \Theta^*] + \frac{1}{2}m \leq \left(\nabla\mathrm{risk}_X[\widetilde{\boldsymbol{\gamma}}, \widetilde{\Theta}] - \nabla\mathrm{risk}[\widetilde{\boldsymbol{\gamma}}, \widetilde{\Theta}]\right)^\top(\boldsymbol{\beta}^* - \widetilde{\boldsymbol{\beta}}) + r\|\boldsymbol{\beta}^*\|_1 - r\|\widetilde{\boldsymbol{\beta}}\|_1\,.$$

Rearranging the display above we obtain

$$\mathrm{risk}[\widetilde{\boldsymbol{\gamma}}, \widetilde{\Theta}] \leq \mathrm{risk}[\boldsymbol{\gamma}^*, \Theta^*] + r\|\boldsymbol{\beta}^*\|_1 + \left(\nabla\mathrm{risk}_X[\widetilde{\boldsymbol{\gamma}}, \widetilde{\Theta}] - \nabla\mathrm{risk}[\widetilde{\boldsymbol{\gamma}}, \widetilde{\Theta}]\right)^\top(\boldsymbol{\beta}^* - \widetilde{\boldsymbol{\beta}}) - r\|\widetilde{\boldsymbol{\beta}}\|_1 - \frac{1}{2}m\,.$$

For the right-hand side of the inequality above we 1. get an absolute value of the third term, 2. add a zero-valued factor, 3. use triangle inequality, and 4. use our results in Lemma 1 to obtain

$$\mathrm{risk}[\widetilde{\boldsymbol{\gamma}}, \widetilde{\Theta}] \leq \mathrm{risk}[\boldsymbol{\gamma}^*, \Theta^*] + r\|\boldsymbol{\beta}^*\|_1 + \left|\left(\nabla\mathrm{risk}_X[\widetilde{\boldsymbol{\gamma}}, \widetilde{\Theta}] - \nabla\mathrm{risk}[\widetilde{\boldsymbol{\gamma}}, \widetilde{\Theta}]\right)^\top(\boldsymbol{\beta}^* - \widetilde{\boldsymbol{\beta}})\right| - r\|\widetilde{\boldsymbol{\beta}}\|_1 - \frac{1}{2}m$$

$$= \mathrm{risk}[\boldsymbol{\gamma}^*, \Theta^*] + 2r\|\boldsymbol{\beta}^*\|_1 + \left|\left(\nabla\mathrm{risk}_X[\widetilde{\boldsymbol{\gamma}}, \widetilde{\Theta}] - \nabla\mathrm{risk}[\widetilde{\boldsymbol{\gamma}}, \widetilde{\Theta}]\right)^\top(\boldsymbol{\beta}^* - \widetilde{\boldsymbol{\beta}})\right| - r\left(\|\widetilde{\boldsymbol{\beta}}\|_1 + \|\boldsymbol{\beta}^*\|_1\right)$$
$$- \frac{1}{2}m$$
$$\leq \mathrm{risk}[\boldsymbol{\gamma}^*, \Theta^*] + 2r\|\boldsymbol{\beta}^*\|_1 + \left|\left(\nabla\mathrm{risk}_X[\widetilde{\boldsymbol{\gamma}}, \widetilde{\Theta}] - \nabla\mathrm{risk}[\widetilde{\boldsymbol{\gamma}}, \widetilde{\Theta}]\right)^\top(\boldsymbol{\beta}^* - \widetilde{\boldsymbol{\beta}})\right| - r\|\boldsymbol{\beta}^* - \widetilde{\boldsymbol{\beta}}\|_1 - \frac{1}{2}m$$
$$\leq \mathrm{risk}[\boldsymbol{\gamma}^*, \Theta^*] + 2r\|\boldsymbol{\beta}^*\|_1 + r_{\mathrm{orc}}\|\boldsymbol{\beta}^* - \widetilde{\boldsymbol{\beta}}\|_1 + \frac{r_{\mathrm{orc}}}{2n} - r\|\boldsymbol{\beta}^* - \widetilde{\boldsymbol{\beta}}\|_1 - \frac{1}{2}m$$

with probability at least $1 - 1/2n$.

The third and fifth terms in the last inequality above can be canceled if we choose the tuning parameter large enough. Hence, we obtain

$$\mathrm{risk}[\widetilde{\boldsymbol{\gamma}}, \widetilde{\Theta}] \leq \mathrm{risk}[\boldsymbol{\gamma}^*, \Theta^*] + 2r\|\boldsymbol{\beta}^*\|_1 + \frac{r_{\mathrm{orc}}}{2n} - \frac{1}{2}m$$

for $r \geq r_{\mathrm{orc}}$.

The rest of the proof is analyzing the behavior of $m$. Let's rewrite $m = \|\boldsymbol{\beta}^*_{\boldsymbol{\alpha},A'} - \widetilde{\boldsymbol{\beta}}_{\boldsymbol{\alpha},A}\|_2^2\, m'$ with

$$m' := \frac{(\boldsymbol{\beta}^*_{\boldsymbol{\alpha},A'} - \widetilde{\boldsymbol{\beta}}_{\boldsymbol{\alpha},A})^\top}{\|\boldsymbol{\beta}^*_{\boldsymbol{\alpha},A'} - \widetilde{\boldsymbol{\beta}}_{\boldsymbol{\alpha},A}\|_2}\nabla^2\mathrm{risk}[\widetilde{\boldsymbol{\gamma}}_{\boldsymbol{\alpha}} + t(\boldsymbol{\gamma}^*_{\boldsymbol{\alpha}} - \widetilde{\boldsymbol{\gamma}}_{\boldsymbol{\alpha}}), \widetilde{\Theta}_{\boldsymbol{\alpha},A} + t(\Theta^*_{\boldsymbol{\alpha},A'} - \widetilde{\Theta}_{\boldsymbol{\alpha},A})]\frac{(\boldsymbol{\beta}^*_{\boldsymbol{\alpha},A'} - \widetilde{\boldsymbol{\beta}}_{\boldsymbol{\alpha},A})}{\|\boldsymbol{\beta}^*_{\boldsymbol{\alpha},A'} - \widetilde{\boldsymbol{\beta}}_{\boldsymbol{\alpha},A}\|_2}\,.$$

Now, we are motivated to employ our results in Proposition 1. To do so, we need to make sure about the invertibility of the matrix $(\widetilde{\Theta}_A + t(\Theta^*_{A'} - \widetilde{\Theta}_A))(\widetilde{\Theta}_A + t(\Theta^*_{A'} - \widetilde{\Theta}_A))^\top$. Using the definition of the extended

networks, it is easy to see that $\widetilde{\Theta}_A$ and $\Theta^*_{A'}$ have full row rank. Then, using Lemma 4, we obtain that the line segment between two matrices $\widetilde{\Theta}_A$ and $\Theta^*_{A'}$ is not invertible at most in finitely many $t$. It means, if we shift $t$ by a tiny value $\varsigma \approx 0$ then, we can make sure that in the new point $t' = t - \varsigma$ the corresponding matrix is invertible, that is,

$$
\begin{aligned}
m' &:= \frac{(\boldsymbol{\beta}^*_{\boldsymbol{\alpha},A'} - \widetilde{\boldsymbol{\beta}}_{\boldsymbol{\alpha},A})^\top}{\|\boldsymbol{\beta}^*_{\boldsymbol{\alpha},A'} - \widetilde{\boldsymbol{\beta}}_{\boldsymbol{\alpha},A}\|_2} \nabla^2 \mathrm{risk}[\widetilde{\boldsymbol{\gamma}}_{\boldsymbol{\alpha}} + (t - \varsigma + \varsigma)(\boldsymbol{\gamma}^*_{\boldsymbol{\alpha}} - \widetilde{\boldsymbol{\gamma}}_{\boldsymbol{\alpha}}), \widetilde{\Theta}_{\boldsymbol{\alpha},A} + (t - \varsigma + \varsigma)(\Theta^*_{\boldsymbol{\alpha},A'} - \widetilde{\Theta}_{\boldsymbol{\alpha},A})] \\
&\quad \frac{(\boldsymbol{\beta}^*_{\boldsymbol{\alpha},A'} - \widetilde{\boldsymbol{\beta}}_{\boldsymbol{\alpha},A})}{\|\boldsymbol{\beta}^*_{\boldsymbol{\alpha},A'} - \widetilde{\boldsymbol{\beta}}_{\boldsymbol{\alpha},A}\|_2} \\
&\approx \frac{(\boldsymbol{\beta}^*_{\boldsymbol{\alpha},A'} - \widetilde{\boldsymbol{\beta}}_{\boldsymbol{\alpha},A})^\top}{\|\boldsymbol{\beta}^*_{\boldsymbol{\alpha},A'} - \widetilde{\boldsymbol{\beta}}_{\boldsymbol{\alpha},A}\|_2} \nabla^2 \mathrm{risk}[\widetilde{\boldsymbol{\gamma}}_{\boldsymbol{\alpha}} + (t - \varsigma)(\boldsymbol{\gamma}^*_{\boldsymbol{\alpha}} - \widetilde{\boldsymbol{\gamma}}_{\boldsymbol{\alpha}}), \widetilde{\Theta}_{\boldsymbol{\alpha},A} + (t - \varsigma)(\Theta^*_{\boldsymbol{\alpha},A'} - \widetilde{\Theta}_{\boldsymbol{\alpha},A})] \\
&\quad \frac{(\boldsymbol{\beta}^*_{\boldsymbol{\alpha},A'} - \widetilde{\boldsymbol{\beta}}_{\boldsymbol{\alpha},A})}{\|\boldsymbol{\beta}^*_{\boldsymbol{\alpha},A'} - \widetilde{\boldsymbol{\beta}}_{\boldsymbol{\alpha},A}\|_2} \\
&= \frac{(\boldsymbol{\beta}^*_{\boldsymbol{\alpha},A'} - \widetilde{\boldsymbol{\beta}}_{\boldsymbol{\alpha},A})^\top}{\|\boldsymbol{\beta}^*_{\boldsymbol{\alpha},A'} - \widetilde{\boldsymbol{\beta}}_{\boldsymbol{\alpha},A}\|_2} \nabla^2 \mathrm{risk}[\widetilde{\boldsymbol{\gamma}}_{\boldsymbol{\alpha}} + t'(\boldsymbol{\gamma}^*_{\boldsymbol{\alpha}} - \widetilde{\boldsymbol{\gamma}}_{\boldsymbol{\alpha}}), \widetilde{\Theta}_{\boldsymbol{\alpha},A} + t'(\Theta^*_{\boldsymbol{\alpha},A'} - \widetilde{\Theta}_{\boldsymbol{\alpha},A})] \frac{(\boldsymbol{\beta}^*_{\boldsymbol{\alpha},A'} - \widetilde{\boldsymbol{\beta}}_{\boldsymbol{\alpha},A})}{\|\boldsymbol{\beta}^*_{\boldsymbol{\alpha},A'} - \widetilde{\boldsymbol{\beta}}_{\boldsymbol{\alpha},A}\|_2} \,,
\end{aligned}
$$

where the second equation is reached by assuming $\varsigma$ is very close to zero and so we can ignore the remaining terms. Then, we have $(\widetilde{\Theta}_A + t'(\Theta^*_{A'} - \widetilde{\Theta}_A))(\widetilde{\Theta}_A + t'(\Theta^*_{A'} - \widetilde{\Theta}_A))^\top$ as an invertible matrix.

Implying Proposition 1 (with $\boldsymbol{a} = (\boldsymbol{\beta}^*_{\boldsymbol{\alpha},A'} - \widetilde{\boldsymbol{\beta}}_{\boldsymbol{\alpha},A})/\|\boldsymbol{\beta}^*_{\boldsymbol{\alpha},A'} - \widetilde{\boldsymbol{\beta}}_{\boldsymbol{\alpha},A}\|_2$ and $d + w - 1$ and $p'$ as the dimension of the input and the effective dimension, respectively) we obtain that $m' \in [0, \infty)$ for appropriate $\boldsymbol{\alpha}$, that is, $\boldsymbol{\alpha}$ with large enough $c$). The observation that $m' \in [0, \infty)$ together with the definition of $m$ implies that $m \in [0, \infty)$ as well.

Tabulating this observation to the display earlier together with our assumption on $\boldsymbol{\beta}^*$ ($\|\boldsymbol{\beta}^*\|_1 = \|\boldsymbol{\gamma}^*\|_1 + \|\Theta^*\|_1 \leq 2\sqrt{\log n}$) and the fact that $1/2n \leq \sqrt{\log n}$, we obtain for all $r \geq r_{\mathrm{orc}}$ that

$$
\begin{aligned}
\mathrm{risk}[\widetilde{\boldsymbol{\gamma}}, \widetilde{\Theta}] &\leq \mathrm{risk}[\boldsymbol{\gamma}^*, \Theta^*] + 2r\|\boldsymbol{\beta}^*\|_1 + \frac{r_{\mathrm{orc}}}{2n} - \frac{1}{2}m \\
&\leq \mathrm{risk}[\boldsymbol{\gamma}^*, \Theta^*] + 2r\|\boldsymbol{\beta}^*\|_1 + \frac{r_{\mathrm{orc}}}{2n} \\
&\leq \mathrm{risk}[\boldsymbol{\gamma}^*, \Theta^*] + 5r\sqrt{\log n}
\end{aligned}
$$

with probability at least $1 - 1/2n$.

The second claim is a trivial consequence of the first claim by 1. using $r = r_{\mathrm{orc}}$ and 2. absorbing the constant 5 in $\nu$ and simplifying to obtain

$$
\begin{aligned}
\mathrm{risk}[\widetilde{\boldsymbol{\gamma}}, \widetilde{\Theta}] &\leq \mathrm{risk}[\boldsymbol{\gamma}^*, \Theta^*] + \nu(\log n)^{3/2}\sqrt{\frac{\log(np)}{n}}\left(5\sqrt{\log n}\right) \\
&= \mathrm{risk}[\boldsymbol{\gamma}^*, \Theta^*] + \nu(\log n)^2\sqrt{\frac{\log(np)}{n}} \,,
\end{aligned}
$$

with probability at least $1 - 1/2n$, which completes the proof. $\square$

## B.2 Proof of Theorem 2

*Proof.* The main ingredients of the proof are the definition of $\tau-$approximate stationary point and our Lemma 3.

We start the proof using the definition of a $\tau-$approximate stationary point in equation 7 that implies

$$
\mathrm{risk}_X[\widetilde{\boldsymbol{\gamma}}, \widetilde{\widetilde{\Theta}}] + r\|\widetilde{\widetilde{\boldsymbol{\beta}}}\|_1 \leq \mathrm{risk}_X[\widetilde{\boldsymbol{\gamma}}, \widetilde{\Theta}] + r\|\widetilde{\boldsymbol{\beta}}\|_1 + \tau \,.
$$

We add zero-valued terms to the both sides of the inequality above to obtain

$$\text{risk}_X[\widetilde{\widetilde{\gamma}}, \widetilde{\widetilde{\Theta}}] - \text{risk}[\widetilde{\widetilde{\gamma}}, \widetilde{\widetilde{\Theta}}] + \text{risk}[\widetilde{\widetilde{\gamma}}, \widetilde{\widetilde{\Theta}}] + r\|\widetilde{\widetilde{\beta}}\|_1 \leq \text{risk}_X[\widetilde{\gamma}, \widetilde{\Theta}] - \text{risk}[\widetilde{\gamma}, \widetilde{\Theta}] + \text{risk}[\widetilde{\gamma}, \widetilde{\Theta}] + r\|\widetilde{\beta}\|_1 + \tau.$$

Then, we 1. rearrange the terms, get an absolute value of the two terms, and use the properties of absolute values, 2. get a supremum over the reasonable parameter space $\mathcal{B}_{\text{res}}$ using our assumptions that $(\widetilde{\widetilde{\gamma}}, \widetilde{\widetilde{\Theta}}), (\widetilde{\gamma}, \widetilde{\Theta}) \in \mathcal{B}_{\text{res}} := \{(\gamma, \Theta) \in \mathcal{B} : \|\gamma\|_1, \|\Theta\|_1 \leq \sqrt{\log n}\}$ (we use our assumption that the stationary is reasonable and our argument in the paragraph above Theorem 2 to reach that $(\widetilde{\widetilde{\gamma}}, \widetilde{\widetilde{\Theta}})$ is reasonable as well), 3. simplify, and 4. leave a negative term to obtain

$$\begin{aligned}
\text{risk}[\widetilde{\widetilde{\gamma}}, \widetilde{\widetilde{\Theta}}] &\leq \text{risk}[\widetilde{\gamma}, \widetilde{\Theta}] + \left|\text{risk}_X[\widetilde{\widetilde{\gamma}}, \widetilde{\widetilde{\Theta}}] - \text{risk}[\widetilde{\widetilde{\gamma}}, \widetilde{\widetilde{\Theta}}]\right| + \left|\text{risk}_X[\widetilde{\gamma}, \widetilde{\Theta}] - \text{risk}[\widetilde{\gamma}, \widetilde{\Theta}]\right| + r\|\widetilde{\beta}\|_1 - r\|\widetilde{\widetilde{\beta}}\|_1 + \tau \\
&\leq \text{risk}[\widetilde{\gamma}, \widetilde{\Theta}] + \sup_{(\gamma,\Theta)\in\mathcal{B}_{\text{res}}} \left|\text{risk}_X[\gamma, \Theta] - \text{risk}[\gamma, \Theta]\right| + \sup_{(\gamma,\Theta)\in\mathcal{B}_{\text{res}}} \left|\text{risk}_X[\gamma, \Theta] - \text{risk}[\gamma, \Theta]\right| \\
&\quad + r\|\widetilde{\beta}\|_1 - r\|\widetilde{\widetilde{\beta}}\|_1 + \tau \\
&= \text{risk}[\widetilde{\gamma}, \widetilde{\Theta}] + 2\sup_{(\gamma,\Theta)\in\mathcal{B}_{\text{res}}} \left|\text{risk}_X[\gamma, \Theta] - \text{risk}[\gamma, \Theta]\right| + r\|\widetilde{\beta}\|_1 - r\|\widetilde{\widetilde{\beta}}\|_1 + \tau \\
&\leq \text{risk}[\widetilde{\gamma}, \widetilde{\Theta}] + 2\sup_{(\gamma,\Theta)\in\mathcal{B}_{\text{res}}} \left|\text{risk}_X[\gamma, \Theta] - \text{risk}[\gamma, \Theta]\right| + r\|\widetilde{\beta}\|_1 + \tau.
\end{aligned}$$

Then, we use 1. our result above, 2. Lemma 3 bounding the second term with $t = \nu\sqrt{\log(32nd^2)/\kappa n}$ and $\mathcal{B} = \mathcal{B}_{\text{res}}$ (with probability at least $1 - 1/2n$), 3. the definition of $\mathcal{B}_{\text{res}}$ to replace $\sup_{(\gamma,\Theta)\in\mathcal{B}_{\text{res}}} \left\|\gamma^{*\top}\Theta^* - \gamma^\top\Theta\right\|_\infty^2 \leq \sup_{(\gamma,\Theta)\in\mathcal{B}_{\text{res}}} 2\|\gamma\|_\infty^2\|\Theta\|_1^2 \leq 2(\log n)^2 =: \epsilon'$, 4. our Theorem 1 upper bounding the first term (for $r \geq r_{\text{orc}}$ with probability at least $1 - 1/2n$), 5. our assumption that stationary is reasonable, 6. simplifying, 7. an assumption that $n \geq 3$ (just for simplifying the terms), and 8. the assumption that $r \geq r_{\text{orc}}$ and the definition of $r_{\text{orc}}$ (note that for simplicity, we absorb all the constants in $\nu$) to obtain

$$\begin{aligned}
\text{risk}[\widetilde{\widetilde{\gamma}}, \widetilde{\widetilde{\Theta}}] &\leq \text{risk}[\widetilde{\gamma}, \widetilde{\Theta}] + 2\sup_{(\gamma,\Theta)\in\mathcal{B}_{\text{res}}} \left|\text{risk}_X[\gamma, \Theta] - \text{risk}[\gamma, \Theta]\right| + r\|\widetilde{\beta}\|_1 + \tau \\
&\leq \text{risk}[\widetilde{\gamma}, \widetilde{\Theta}] + 2\nu\sqrt{\frac{\log(32nd^2)}{\kappa n}}\left(1 + 4\epsilon' + 4\sqrt{\epsilon'}\right) + r\|\widetilde{\beta}\|_1 + \tau \\
&\leq \text{risk}[\widetilde{\gamma}, \widetilde{\Theta}] + 2\nu\sqrt{\frac{\log(32nd^2)}{\kappa n}}\left(1 + 8(\log n)^2 + 8\log n\right) + r\|\widetilde{\beta}\|_1 + \tau \\
&\leq \text{risk}[\gamma^*, \Theta^*] + 5r\sqrt{\log n} + 2\nu\sqrt{\frac{\log(32nd^2)}{\kappa n}}\left(1 + 8(\log n)^2 + 8\log n\right) + r\|\widetilde{\beta}\|_1 + \tau \\
&\leq \text{risk}[\gamma^*, \Theta^*] + 5r\sqrt{\log n} + 2\nu\sqrt{\frac{\log(32nd^2)}{\kappa n}}\left(1 + 8(\log n)^2 + 8\log n\right) + 2r\sqrt{\log n} + \tau \\
&= \text{risk}[\gamma^*, \Theta^*] + 7r\sqrt{\log n} + 2\nu\sqrt{\frac{\log(32nd^2)}{\kappa n}}\left(1 + 8(\log n)^2 + 8\log n\right) + \tau \\
&\leq \text{risk}[\gamma^*, \Theta^*] + 7r\sqrt{\log n} + 34\nu\sqrt{\frac{\log(32nd^2)}{\kappa n}}(\log n)^2 + \tau \\
&\leq \text{risk}[\gamma^*, \Theta^*] + 8r\sqrt{\log n} + \tau
\end{aligned}$$

with probability at least $1 - (1+1)/2n$, which is obtained by the fact that if $a \leq z_1 + z_2$

$$\begin{aligned}
\mathbb{P}(a \leq c_1 + c_2) &\geq \mathbb{P}(z_1 + z_2 \leq c_1 + c_2) \\
&= 1 - \mathbb{P}(z_1 + z_2 > c_1 + c_2) \\
&\geq 1 - \left(\mathbb{P}(z_1 > c_1) + \mathbb{P}(z_2 > c_2)\right),
\end{aligned}$$

where $a, z_1, z_2$ are random variables and $c_1, c_2$ are constants, as desired.

The second claim is a trivial consequence of the first claim by 1. using $r = r_{\mathrm{orc}}$ and 2. absorbing the constant 8 in $\nu$ to obtain

$$
\mathrm{risk}[\widetilde{\boldsymbol{\gamma}}, \widetilde{\Theta}] \ \ \mathrm{risk}[\boldsymbol{\gamma}^*, \Theta^*] + \nu(\log n)^{3/2}\sqrt{\frac{\log(np)}{n}}\left(8\sqrt{\log n}\right) + \tau
$$

$$
= \ \mathrm{risk}[\boldsymbol{\gamma}^*, \Theta^*] + \nu(\log n)^2\sqrt{\frac{\log(np)}{n}} + \tau\,,
$$

with probability at least $1 - 1/n$, which completes the proof. $\qquad\square$

## B.3 Proof of Proposition 1

*Proof.* The proof is based on basic algebra and property of scaling weights across the layers in neural networks. Without loss of generality, we assume that $\boldsymbol{x}_i \in \mathcal{N}(\boldsymbol{0}, I_{d\times d})$ (the proof for independent and centered sub-Gaussian random vectors $\boldsymbol{x}_i$ with independent coordinates is the same, just some constants may change, which doesn't affect the main results).

Let's consider all the network parameters as a vector of length $p$ (recall that $p = w + w \cdot d$). Then, we can tabulate the second-order partial derivatives of $\mathrm{risk}[\boldsymbol{\gamma}, \Theta]$ in a matrix called $\nabla^2\mathrm{risk}[\boldsymbol{\gamma}, \Theta] \in \mathbb{R}^{p\times p}$ (for notational simplicity, we focus on $\nabla^2\mathrm{risk}[\boldsymbol{\gamma}, \Theta]$ for the moment and then we move to $\nabla^2\mathrm{risk}[\boldsymbol{\gamma_\alpha}, \Theta_{\boldsymbol{\alpha}}]$ at the end of the proof) of the form

$$
\nabla^2\mathrm{risk}[\boldsymbol{\gamma}, \Theta] = \left[\begin{array}{cc} A & C \\ B & D \end{array}\right]
$$

with $A \in \mathbb{R}^{w\times w}$, $B \in \mathbb{R}^{(w\cdot d)\times w}$, $C \in \mathbb{R}^{w\times(w\cdot d)}$, and $D \in \mathbb{R}^{(w\cdot d)\times(w\cdot d)}$, where

$$
A_{j',j} := \frac{\partial^2}{\partial\gamma_{j'}\partial\gamma_j}\mathrm{risk}[\boldsymbol{\gamma}, \Theta]\,,
$$

$$
B_{(j'-1)d+k',j} := \frac{\partial^2}{\partial\theta_{j'k'}\partial\gamma_j}\mathrm{risk}[\boldsymbol{\gamma}, \Theta]\,,
$$

$$
C_{j',(j-1)d+k} := \frac{\partial^2}{\partial\gamma_{j'}\partial\theta_{jk}}\mathrm{risk}[\boldsymbol{\gamma}, \Theta]\,,
$$

$$
D_{(j'-1)d+k',(j-1)d+k} := \frac{\partial^2}{\partial\theta_{j'k'}\partial\theta_{jk}}\mathrm{risk}[\boldsymbol{\gamma}, \Theta]
$$

for $j, j' \in \{1, \ldots, w\}$ and $k, k' \in \{1, \ldots, d\}$.

Applying the block-wise structure of $\nabla^2\mathrm{risk}[\boldsymbol{\gamma}, \Theta]$, we are motivated to analyze the behavior of

$$
\boldsymbol{a}^\top\nabla^2\mathrm{risk}[\boldsymbol{\gamma}, \Theta]\boldsymbol{a} = (\boldsymbol{a}^1)^\top A\boldsymbol{a}^1 + (\boldsymbol{a}^1)^\top C\boldsymbol{a}^2 + (\boldsymbol{a}^2)^\top B\boldsymbol{a}^1 + (\boldsymbol{a}^2)^\top D\boldsymbol{a}^2\,.
$$

Note that $C = B^\top$ (see Lemma 6), so, we are left to analyze the behavior of

$$
\boldsymbol{a}^\top\nabla^2\mathrm{risk}[\boldsymbol{\gamma}, \Theta]\boldsymbol{a} = (\boldsymbol{a}^1)^\top A\boldsymbol{a}^1 + 2(\boldsymbol{a}^1)^\top C\boldsymbol{a}^2 + (\boldsymbol{a}^2)^\top D\boldsymbol{a}^2
$$

for all $\boldsymbol{a} \in \mathbb{R}^p$ with $\|\boldsymbol{a}\|_2 = 1$.

We do the proof in steps: We start by going through the three terms on the right-hand side of the display above separately, to write them in a mathematically nice formulation (Steps 1:3). In Step 4, we sum up the results calculated in Steps 1:3. Finally in Step 5, we use our results in Steps 1:4 to prove the main claims of the proposition.

*Step 1:* We show that for $\boldsymbol{a}^2 \in \mathbb{R}^{w\cdot d}$ and $D \in \mathbb{R}^{(w\cdot d)\times(w\cdot d)}$,

$$
(\boldsymbol{a}^2)^\top D\boldsymbol{a}^2 = 2\sum_{k=1}^d\left(\boldsymbol{\gamma}^\top(\boldsymbol{a}^2)^k\right)^2\,,
$$

where we denote $(\boldsymbol{a}^2)^k := \big((\boldsymbol{a}^2)_k, (\boldsymbol{a}^2)_{d+k}, \ldots, (\boldsymbol{a}^2)_{(w-1)d+k}\big)^\top \in \mathbb{R}^w$ (as a sub-vector of $\boldsymbol{a}^2$) for each $k \in \{1, \ldots, d\}$.

We start by writing matrix product in the form of sums and fill the entries of matrix $D$ with the corresponding values from the definition to get

$$
(\boldsymbol{a}^2)^\top D \boldsymbol{a}^2
$$

$$
= \sum_{j=1}^w \sum_{k=1}^d \left( \sum_{j'=1}^w \sum_{k'=1}^d \left( (\boldsymbol{a}^2)_{(j'-1)d+k'} \frac{\partial^2}{\partial \theta_{j'k'} \partial \theta_{jk}} \mathrm{risk}[\boldsymbol{\gamma}, \Theta] \right) (\boldsymbol{a}^2)_{(j-1)d+k} \right).
$$

By Lemma 6 we have

$$
\frac{\partial^2}{\partial \theta_{j'k'} \partial \theta_{jk}} \mathrm{risk}[\boldsymbol{\gamma}, \Theta] = 2\gamma_{j'} \gamma_j \mathbb{E}_{(\boldsymbol{x}, y)} \big[ (\boldsymbol{x})_k (\boldsymbol{x})_{k'} \big],
$$

which using our assumption on $\boldsymbol{x}$ (identity covariance matrix) implies

$$
\frac{\partial^2}{\partial \theta_{j'k'} \partial \theta_{jk}} \mathrm{risk}[\boldsymbol{\gamma}, \Theta] = 2\gamma_{j'} \gamma_j
$$

for $k = k'$ and zero otherwise (for $k \neq k'$). We use 1. our display earlier, 2. the result above, 3. the linearity of sums, 4. some rewriting (using multinomial theorem), and 5. implying our notation $(\boldsymbol{a}^2)^k$ for writing the sum in the form of product to obtain

$$
(\boldsymbol{a}^2)^\top D \boldsymbol{a}^2 = \sum_{j=1}^w \sum_{k=1}^d \left( \sum_{j'=1}^w \sum_{k'=1}^d \left( (\boldsymbol{a}^2)_{(j'-1)d+k'} \frac{\partial^2}{\partial \theta_{j'k'} \partial \theta_{jk}} \mathrm{risk}[\boldsymbol{\gamma}, \Theta] \right) \boldsymbol{a}^2_{(j-1)d+k} \right)
$$

$$
= 2 \sum_{j=1}^w \sum_{k=1}^d \left( \sum_{j'=1}^w \left( (\boldsymbol{a}^2)_{(j'-1)d+k} \gamma_{j'} \gamma_j \right) \boldsymbol{a}^2_{(j-1)d+k} \right)
$$

$$
= 2 \sum_{j=1}^w \sum_{k=1}^d \sum_{j'=1}^w \left( (\boldsymbol{a}^2)_{(j'-1)d+k} \gamma_{j'} \gamma_j \boldsymbol{a}^2_{(j-1)d+k} \right)
$$

$$
= 2 \sum_{k=1}^d \left( \sum_{j=1}^w (\boldsymbol{a}^2)_{(j-1)d+k} \gamma_j \right)^2
$$

$$
= 2 \sum_{k=1}^d \left( \boldsymbol{\gamma}^\top (\boldsymbol{a}^2)^k \right)^2.
$$

*Step 2:* We prove that for $\boldsymbol{a}^1 \in \mathbb{R}^w$ and $A \in \mathbb{R}^{w \times w}$,

$$
(\boldsymbol{a}^1)^\top A \boldsymbol{a}^1 = 2 \sum_{k=1}^d \left( (\Theta_{.,k})^\top \boldsymbol{a}^1 \right)^2,
$$

where $\Theta_{.,k}$ denotes the $k$-th column of $\Theta$.

For each $j, j' \in \{1, \ldots, w\}$, we use 1. the result of Lemma 6, 2. the definition of covariance, 3. the fact that $\mathrm{Cov}(\Theta \boldsymbol{x}) = \Theta \mathrm{Cov}(\boldsymbol{x}) \Theta^\top$, 4. the assumption on $\boldsymbol{x}$ (identity covariance), and 5. rewriting to obtain

$$
\frac{\partial^2}{\partial \gamma_{j'} \partial \gamma_j} \mathrm{risk}[\boldsymbol{\gamma}, \Theta] = 2\mathbb{E}_{(\boldsymbol{x}, y)} \big[ (\Theta \boldsymbol{x})_{j'} (\Theta \boldsymbol{x})_j \big]
$$

$$
= 2 \big( \mathrm{Cov}(\Theta \boldsymbol{x}) \big)_{j'j}
$$

$$
= 2 \big( \Theta \mathrm{Cov}(\boldsymbol{x}) \Theta^\top \big)_{j'j}
$$

$$
= 2 \big( \Theta \Theta^\top \big)_{j'j}
$$

$$
= 2 \sum_{k=1}^d \theta_{j'k} \theta_{jk}.
$$

We use 1. the definition of sub-matrix $A$ to write the matrix product in the form of a sum, 2. tabulating above result and using the linearity of sums, 3. some rewriting (using the multinomial theorem), and 4. writing the sum in the form of product to obtain

$$
\begin{aligned}
(\boldsymbol{a}^1)^\top A \boldsymbol{a}^1 &= \sum_{j=1}^w \sum_{j'=1}^w \left( (\boldsymbol{a}^1)_{j'} \frac{\partial^2}{\partial \gamma_{j'} \partial \gamma_j} \mathrm{risk}[\boldsymbol{\gamma}, \Theta](\boldsymbol{a}^1)_j \right) \\
&= \sum_{k=1}^d \sum_{j=1}^w \sum_{j'=1}^w 2(\boldsymbol{a}^1)_{j'} \theta_{j'k} \theta_{jk} (\boldsymbol{a}^1)_j \\
&= 2 \sum_{k=1}^d \left( \sum_{j=1}^w (\theta_{jk}(\boldsymbol{a}^1)_j) \right)^2 \\
&= 2 \sum_{k=1}^d \left( (\Theta_{.,k})^\top \boldsymbol{a}^1 \right)^2 .
\end{aligned}
$$

*Step 3:* We show that for $\boldsymbol{a}^1 \in \mathbb{R}^w$, $\boldsymbol{a}^2 \in R^{w \cdot d}$, and $C \in \mathbb{R}^{w \times (w \cdot d)}$,

$$
\begin{aligned}
(\boldsymbol{a}^1)^\top C \boldsymbol{a}^2 &= 2 \sum_{k=1}^d \left( \boldsymbol{\gamma}^\top (\boldsymbol{a}^2)^k \right) \left( (\Theta_{.,k})^\top \boldsymbol{a}^1 \right) \\
&\quad + 2 \sum_{k=1}^d \left( \left( (\boldsymbol{\gamma}^\top \Theta - \boldsymbol{\gamma}^{*\top} \Theta^*)_k - \mathbb{E}_{(\boldsymbol{x},y)}\left[ (y - \boldsymbol{\gamma}^{*\top} \Theta^* \boldsymbol{x})(\boldsymbol{x})_k \right] \right) (\boldsymbol{a}^1)^\top (\boldsymbol{a}^2)^k \right).
\end{aligned}
$$

Expanding $(\boldsymbol{a}^1)^\top C \boldsymbol{a}^2$ yields

$$
(\boldsymbol{a}^1)^\top C \boldsymbol{a}^2 = \sum_{j=1}^w \sum_{k=1}^d \left( \sum_{j'=1}^w \left( (\boldsymbol{a}^1)_{j'} \frac{\partial^2}{\partial \gamma_{j'} \partial \theta_{jk}} \mathrm{risk}[\boldsymbol{\gamma}, \Theta] \right) (\boldsymbol{a}^2)_{(j-1)d+k} \right).
$$

Now, we need to consider two different cases:

*Case 1:* $(j \neq j')$

We use 1. the result of Lemma 6, 2. writing matrix product in the form of a sum, 3. linearity of sums and expectations, and 4. our assumption on $\boldsymbol{x}$ to get for each $j, j' \in \{1, \ldots, w\}$ and $k \in \{1, \ldots, d\}$ with $j \neq j'$ that

$$
\begin{aligned}
\frac{\partial^2}{\partial \gamma_{j'} \partial \theta_{jk}} \mathrm{risk}[\boldsymbol{\gamma}, \Theta] &= 2 \gamma_j \mathbb{E}_{(\boldsymbol{x},y)}\left[ (\boldsymbol{x})_k (\Theta \boldsymbol{x})_{j'} \right] \\
&= 2 \gamma_j \mathbb{E}_{(\boldsymbol{x},y)}\left[ (\boldsymbol{x})_k \sum_{k'=1}^d (\theta_{j'k'}(\boldsymbol{x})_{k'}) \right] \\
&= 2 \gamma_j \sum_{k'=1}^d \left( \theta_{j'k'} \mathbb{E}_{(\boldsymbol{x},y)}\left[ (\boldsymbol{x})_k (\boldsymbol{x})_{k'} \right] \right) \\
&= 2 \gamma_j \theta_{j'k} .
\end{aligned}
$$

*Case 2:* $(j = j')$

We use 1. the result of Lemma 6, 2. linearity of expectations, 3. linearity of expectations and our assumption on $\boldsymbol{x}$ (same argument as above), 4. linearity of expectations, 5. linearity of expectations and our assumption on $\boldsymbol{x}$, 6. adding a zero-valued term, and 7. again linearity of expectations, our assumption on $\boldsymbol{x}$, and rearranging to obtain

$$
\frac{\partial^2}{\partial \gamma_j \partial \theta_{jk}} \mathrm{risk}[\boldsymbol{\gamma}, \Theta] = 2 \mathbb{E}_{(\boldsymbol{x},y)}\left[ \gamma_j (\boldsymbol{x})_k (\Theta \boldsymbol{x})_j - (y - \boldsymbol{\gamma}^\top \Theta \boldsymbol{x})(\boldsymbol{x})_k \right]
$$

$$= 2\mathbb{E}_{(\boldsymbol{x},y)}\big[\gamma_j(\boldsymbol{x})_k(\Theta\boldsymbol{x})_j\big] + 2\mathbb{E}_{(\boldsymbol{x},y)}\big[(\boldsymbol{\gamma}^\top\Theta\boldsymbol{x})(\boldsymbol{x})_k - y(\boldsymbol{x})_k\big]$$

$$= 2\gamma_j\theta_{jk} + 2\mathbb{E}_{(\boldsymbol{x},y)}\big[(\boldsymbol{\gamma}^\top\Theta\boldsymbol{x})(\boldsymbol{x})_k - y(\boldsymbol{x})_k\big]$$

$$= 2\gamma_j\theta_{jk} + 2\mathbb{E}_{(\boldsymbol{x},y)}\big[(\boldsymbol{\gamma}^\top\Theta\boldsymbol{x})(\boldsymbol{x})_k\big] - 2\mathbb{E}_{(\boldsymbol{x},y)}\big[y(\boldsymbol{x})_k\big]$$

$$= 2\gamma_j\theta_{jk} + 2(\boldsymbol{\gamma}^\top\Theta)_k - 2\mathbb{E}_{(\boldsymbol{x},y)}\big[y(\boldsymbol{x})_k\big]$$

$$= 2\gamma_j\theta_{jk} + 2(\boldsymbol{\gamma}^\top\Theta)_k - 2\mathbb{E}_{(\boldsymbol{x},y)}\big[(y + \boldsymbol{\gamma}^{*\top}\Theta^*\boldsymbol{x} - \boldsymbol{\gamma}^{*\top}\Theta^*\boldsymbol{x})(\boldsymbol{x})_k\big]$$

$$= 2\gamma_j\theta_{jk} + 2(\boldsymbol{\gamma}^\top\Theta - \boldsymbol{\gamma}^{*\top}\Theta^*)_k - 2\mathbb{E}_{(\boldsymbol{x},y)}\big[(y - \boldsymbol{\gamma}^{*\top}\Theta^*\boldsymbol{x})(\boldsymbol{x})_k\big].$$

Now, we 1. use our earlier expansion, 2. separate the innermost sum in two cases, 3. use the result above (Case 1 and Case 2), 4. rearranging, 5. use linearity of sums and some rewriting, and 6. write sums in the form of vector products and rearranging to obtain

$$(\boldsymbol{a}^1)^\top C \boldsymbol{a}^2$$

$$= \sum_{j=1}^{w}\sum_{k=1}^{d}\left(\sum_{j'=1}^{w}\left((\boldsymbol{a}^1)_{j'}\frac{\partial^2}{\partial\gamma_{j'}\partial\theta_{jk}}\mathrm{risk}[\boldsymbol{\gamma},\Theta]\right)(\boldsymbol{a}^2)_{(j-1)d+k}\right)$$

$$= \sum_{j=1}^{w}\sum_{k=1}^{d}\left(\sum_{j'=1,j'\neq j}^{w}\left((\boldsymbol{a}^1)_{j'}\frac{\partial^2}{\partial\gamma_{j'}\partial\theta_{jk}}\mathrm{risk}[\boldsymbol{\gamma},\Theta]\right)(\boldsymbol{a}^2)_{(j-1)d+k}\right)$$

$$+ \sum_{j=1}^{w}\sum_{k=1}^{d}\left((\boldsymbol{a}^1)_{j}\frac{\partial^2}{\partial\gamma_{j}\partial\theta_{jk}}\mathrm{risk}[\boldsymbol{\gamma},\Theta](\boldsymbol{a}^2)_{(j-1)d+k}\right)$$

$$= 2\sum_{j=1}^{w}\sum_{k=1}^{d}\sum_{j'=1,j'\neq j}^{w}\left((\boldsymbol{a}^1)_{j'}\gamma_j\theta_{j'k}(\boldsymbol{a}^2)_{(j-1)d+k}\right)$$

$$+ 2\sum_{j=1}^{w}\sum_{k=1}^{d}\left((\boldsymbol{a}^1)_{j}\Big(\gamma_j\theta_{jk} + (\boldsymbol{\gamma}^\top\Theta - \boldsymbol{\gamma}^{*\top}\Theta^*)_k - \mathbb{E}_{(\boldsymbol{x},y)}\big[(y - \boldsymbol{\gamma}^{*\top}\Theta^*\boldsymbol{x})(\boldsymbol{x})_k\big]\Big)(\boldsymbol{a}^2)_{(j-1)d+k}\right)$$

$$= 2\sum_{j=1}^{w}\sum_{k=1}^{d}\sum_{j'=1}^{w}\left((\boldsymbol{a}^1)_{j'}\gamma_j\theta_{j'k}(\boldsymbol{a}^2)_{(j-1)d+k}\right)$$

$$+ 2\sum_{j=1}^{w}\sum_{k=1}^{d}\left((\boldsymbol{a}^1)_{j}\Big((\boldsymbol{\gamma}^\top\Theta - \boldsymbol{\gamma}^{*\top}\Theta^*)_k - \mathbb{E}_{(\boldsymbol{x},y)}\big[(y - \boldsymbol{\gamma}^{*\top}\Theta^*\boldsymbol{x})(\boldsymbol{x})_k\big]\Big)(\boldsymbol{a}^2)_{(j-1)d+k}\right)$$

$$= 2\sum_{k=1}^{d}\left(\sum_{j'=1}^{w}(\boldsymbol{a}^1)_{j'}\theta_{j'k}\right)\left(\sum_{j=1}^{w}\gamma_j(\boldsymbol{a}^2)_{(j-1)d+k}\right)$$

$$+ 2\sum_{k=1}^{d}\Big((\boldsymbol{\gamma}^\top\Theta - \boldsymbol{\gamma}^{*\top}\Theta^*)_k - \mathbb{E}_{(\boldsymbol{x},y)}\big[(y - \boldsymbol{\gamma}^{*\top}\Theta^*\boldsymbol{x})(\boldsymbol{x})_k\big]\Big)\left(\sum_{j=1}^{w}(\boldsymbol{a}^1)_{j}(\boldsymbol{a}^2)_{(j-1)d+k}\right)$$

$$= 2\sum_{k=1}^{d}\Big(\boldsymbol{\gamma}^\top(\boldsymbol{a}^2)^k\Big)\Big((\Theta_{\cdot,k})^\top(\boldsymbol{a}^1)\Big)$$

$$+ 2\sum_{k=1}^{d}\Big((\boldsymbol{\gamma}^\top\Theta - \boldsymbol{\gamma}^{*\top}\Theta^*)_k - \mathbb{E}_{(\boldsymbol{x},y)}\big[(y - \boldsymbol{\gamma}^{*\top}\Theta^*\boldsymbol{x})(\boldsymbol{x})_k\big]\Big)(\boldsymbol{a}^1)^\top(\boldsymbol{a}^2)^k.$$

*Step 4:* We prove that for any $\boldsymbol{a} = [(\boldsymbol{a}^1)^\top, (\boldsymbol{a}^2)^\top]^\top \in \mathbb{R}^p$ and $(\boldsymbol{\gamma}, \Theta) \in \mathcal{B}$, it holds that

$$\boldsymbol{a}^\top\nabla^2\mathrm{risk}[\boldsymbol{\gamma},\Theta]\boldsymbol{a} = 2\sum_{k=1}^{d}\Big((\Theta_{\cdot,k})^\top(\boldsymbol{a}^1) + \boldsymbol{\gamma}^\top(\boldsymbol{a}^2)^k\Big)^2 + 4\sum_{k=1}^{d}(\boldsymbol{\gamma}^\top\Theta - \boldsymbol{\gamma}^{*\top}\Theta^*)_k(\boldsymbol{a}^1)^\top(\boldsymbol{a}^2)^k$$

$$- 4\sum_{k=1}^{d}\mathbb{E}_{(\boldsymbol{x},y)}\big[(y - \boldsymbol{\gamma}^{*\top}\Theta^*\boldsymbol{x})(\boldsymbol{x})_k\big](\boldsymbol{a}^1)^\top(\boldsymbol{a}^2)^k.$$

We use 1. the block-wise structure of the Hessian matrix and rearranging, 2. our results in Steps 1:3, and 3. multinomial theorem to obtain

$$\boldsymbol{a}^\top \nabla^2 \mathrm{risk}[\boldsymbol{\gamma}, \Theta]\boldsymbol{a} = (\boldsymbol{a}^1)^\top A\boldsymbol{a}^1 + (\boldsymbol{a}^2)^\top D\boldsymbol{a}^2 + 2(\boldsymbol{a}^1)^\top C\boldsymbol{a}^2$$

$$= 2\sum_{k=1}^d \left(\boldsymbol{\gamma}^\top (\boldsymbol{a}^2)^k\right)^2 + 2\sum_{k=1}^d \left((\Theta_{.,k})^\top \boldsymbol{a}^1\right)^2 + 4\sum_{k=1}^d \left(\boldsymbol{\gamma}^\top (\boldsymbol{a}^2)^k\right)\left((\Theta_{.,k})^\top \boldsymbol{a}^1\right)$$

$$+ 4\sum_{k=1}^d \left(\left(\left(\boldsymbol{\gamma}^\top \Theta - \boldsymbol{\gamma}^{*\top}\Theta^*\right)_k - \mathbb{E}_{(\boldsymbol{x},y)}\left[(y - \boldsymbol{\gamma}^{*\top}\Theta^*\boldsymbol{x})(\boldsymbol{x})_k\right]\right)(\boldsymbol{a}^1)^\top (\boldsymbol{a}^2)^k\right)$$

$$= 2\sum_{k=1}^d \left((\Theta_{.,k})^\top \boldsymbol{a}^1 + \boldsymbol{\gamma}^\top (\boldsymbol{a}^2)^k\right)^2 + 4\sum_{k=1}^d (\boldsymbol{\gamma}^\top \Theta - \boldsymbol{\gamma}^{*\top}\Theta^*)_k (\boldsymbol{a}^1)^\top (\boldsymbol{a}^2)^k$$

$$- 4\sum_{k=1}^d \mathbb{E}_{(\boldsymbol{x},y)}\left[(y - \boldsymbol{\gamma}^{*\top}\Theta^*\boldsymbol{x})(\boldsymbol{x})_k\right](\boldsymbol{a}^1)^\top (\boldsymbol{a}^2)^k \,.$$

*Step 5:* Now, we employ our results in Steps 1–4 to prove the main claims of the proposition.

*Claim 1:* ($\boldsymbol{a}^1 = \boldsymbol{0}$ and $\boldsymbol{a}^2 \neq \boldsymbol{0}$)

We use 1. the block-wise structure of the Hessian, 2. the assumption that $\boldsymbol{a}^1 = \boldsymbol{0}$, 3. our result in Step 1, and 4. the fact that sum of non-negative terms is also non-negative to obtain

$$\boldsymbol{a}^\top \nabla^2 \mathrm{risk}[\boldsymbol{\gamma}, \Theta]\boldsymbol{a} = (\boldsymbol{a}^1)^\top A\boldsymbol{a}^1 + 2(\boldsymbol{a}^1)^\top C\boldsymbol{a}^2 + (\boldsymbol{a}^2)^\top D\boldsymbol{a}^2$$

$$= (\boldsymbol{a}^2)^\top D\boldsymbol{a}^2$$

$$= 2\sum_{k=1}^d \left(\boldsymbol{\gamma}^\top (\boldsymbol{a}^2)^k\right)^2$$

$$\geq 0 \,.$$

The above display can also reveal that for all $\boldsymbol{\alpha} \in \mathbb{R}^w \setminus \{\boldsymbol{0}\}$ (moving to a scaled version of the parameters)

$$\boldsymbol{a}^\top \nabla^2 \mathrm{risk}[\boldsymbol{\gamma_\alpha}, \Theta_{\boldsymbol{\alpha}}]\boldsymbol{a} = 2\sum_{k=1}^d \left((\boldsymbol{\gamma_\alpha})^\top (\boldsymbol{a}^2)^k\right)^2 \geq 0 \,,$$

as desired.

*Claim 2:* ($\boldsymbol{a}^1 \neq \boldsymbol{0}$ and $\boldsymbol{a}^2 = \boldsymbol{0}$)

The proof is similar to *Claim 1* so we omit the proof.

*Claim 3:* ($\boldsymbol{a}^1 \neq \boldsymbol{0}$ and $\boldsymbol{a}^2 \neq \boldsymbol{0}$)

We use our results in Step 4 together with getting an absolute value of the two last terms to obtain

$$\boldsymbol{a}^\top \nabla^2 \mathrm{risk}[\boldsymbol{\gamma}, \Theta]\boldsymbol{a} = 2\sum_{k=1}^d \left((\Theta_{.,k})^\top \boldsymbol{a}^1 + \boldsymbol{\gamma}^\top (\boldsymbol{a}^2)^k\right)^2 + 4\sum_{k=1}^d (\boldsymbol{\gamma}^\top \Theta - \boldsymbol{\gamma}^{*\top}\Theta^*)_k (\boldsymbol{a}^1)^\top (\boldsymbol{a}^2)^k$$

$$- 4\sum_{k=1}^d \mathbb{E}_{(\boldsymbol{x},y)}\left[(y - \boldsymbol{\gamma}^{*\top}\Theta^*\boldsymbol{x})(\boldsymbol{x})_k\right](\boldsymbol{a}^1)^\top (\boldsymbol{a}^2)^k$$

$$\geq 2\sum_{k=1}^d \left((\Theta_{.,k})^\top \boldsymbol{a}^1 + \boldsymbol{\gamma}^\top (\boldsymbol{a}^2)^k\right)^2 - 4\left|\sum_{k=1}^d (\boldsymbol{\gamma}^\top \Theta - \boldsymbol{\gamma}^{*\top}\Theta^*)_k (\boldsymbol{a}^1)^\top (\boldsymbol{a}^2)^k\right|$$

$$- 4\left|\sum_{k=1}^d \mathbb{E}_{(\boldsymbol{x},y)}\left[(y - \boldsymbol{\gamma}^{*\top}\Theta^*\boldsymbol{x})(\boldsymbol{x})_k\right](\boldsymbol{a}^1)^\top (\boldsymbol{a}^2)^k\right| \,.$$

First, let's concentrate on the second term of display above and 1. use the triangle inequality and properties of absolute values, 2. use Hölder inequality, 3. get a factor $\|\boldsymbol{a}^1\|_2$ out of the summation, 4. use Cauchy–Schwarz inequality, and 5. some rewriting to obtain

$$
4\left|\sum_{k=1}^{d}(\boldsymbol{\gamma}^\top\Theta - \boldsymbol{\gamma}^{*\top}\Theta^*)_k(\boldsymbol{a}^1)^\top(\boldsymbol{a}^2)^k\right| \leq 4\sum_{k=1}^{d}\left|(\boldsymbol{\gamma}^\top\Theta - \boldsymbol{\gamma}^{*\top}\Theta^*)_k\right|\left|(\boldsymbol{a}^1)^\top(\boldsymbol{a}^2)^k\right|
$$

$$
\leq 4\sum_{k=1}^{d}\left|(\boldsymbol{\gamma}^\top\Theta - \boldsymbol{\gamma}^{*\top}\Theta^*)_k\right|\|\boldsymbol{a}^1\|_2\|(\boldsymbol{a}^2)^k\|_2
$$

$$
= 4\|\boldsymbol{a}^1\|_2\sum_{k=1}^{d}\left|(\boldsymbol{\gamma}^\top\Theta - \boldsymbol{\gamma}^{*\top}\Theta^*)_k\right|\|(\boldsymbol{a}^2)^k\|_2
$$

$$
\leq 4\|\boldsymbol{a}^1\|_2\sqrt{\sum_{k=1}^{d}\left|(\boldsymbol{\gamma}^\top\Theta - \boldsymbol{\gamma}^{*\top}\Theta^*)_k\right|^2}\sqrt{\sum_{k=1}^{d}\|(\boldsymbol{a}^2)^k\|_2^2}
$$

$$
= 4\|\boldsymbol{a}^1\|_2\|\boldsymbol{a}^2\|_2\|\boldsymbol{\gamma}^\top\Theta - \boldsymbol{\gamma}^{*\top}\Theta^*\|_2\,.
$$

Then, we use 1. our assumption that $y = \boldsymbol{\gamma}^{*\top}\Theta^*\boldsymbol{x} + u$, 2. independence of $u$ and $\boldsymbol{x}$, and 3. our assumption that $\mathbb{E}[\boldsymbol{x}] = \boldsymbol{0}$ (also we have $\mathbb{E}[u] = 0$) to obtain

$$
4\left|\sum_{k=1}^{d}\mathbb{E}_{(\boldsymbol{x},y)}\big[(y - \boldsymbol{\gamma}^{*\top}\Theta^*\boldsymbol{x})(\boldsymbol{x})_k\big](\boldsymbol{a}^1)^\top(\boldsymbol{a}^2)^k\right| = 4\left|\sum_{k=1}^{d}\mathbb{E}_{(\boldsymbol{x},y)}\big[u(\boldsymbol{x})_k\big](\boldsymbol{a}^1)^\top(\boldsymbol{a}^2)^k\right|
$$

$$
= 4\left|\sum_{k=1}^{d}\mathbb{E}_{(\boldsymbol{x},y)}\big[u\big]\mathbb{E}_{(\boldsymbol{x},y)}\big[(\boldsymbol{x})_k\big](\boldsymbol{a}^1)^\top(\boldsymbol{a}^2)^k\right|
$$

$$
= 0\,.
$$

Tabulating two observations above in the previous display we obtain

$$
\boldsymbol{a}^\top\nabla^2\mathrm{risk}[\boldsymbol{\gamma},\Theta]\boldsymbol{a} \geq 2\sum_{k=1}^{d}\Big((\Theta_{.,k})^\top\boldsymbol{a}^1 + \boldsymbol{\gamma}^\top(\boldsymbol{a}^2)^k\Big)^2 - 4\|\boldsymbol{a}^1\|_2\|\boldsymbol{a}^2\|_2\|\boldsymbol{\gamma}^\top\Theta - \boldsymbol{\gamma}^{*\top}\Theta^*\|_2\,.
$$

Now, let's define for each $k \in \{1,\ldots,d\}$ that $A_k := (\Theta_{.,k})^\top\boldsymbol{a}^1$, $B_k := \boldsymbol{\gamma}^\top(\boldsymbol{a}^2)^k$, and using the fact $(A_k + B_k)^2 \geq \frac{1}{2}(A_k)^2 - (B_k)^2$ to obtain

$$
\boldsymbol{a}^\top\nabla^2\mathrm{risk}[\boldsymbol{\gamma},\Theta]\boldsymbol{a}
$$

$$
\geq 2\sum_{k=1}^{d}(A_k + B_k)^2 - 4\|\boldsymbol{a}^1\|_2\|\boldsymbol{a}^2\|_2\|\boldsymbol{\gamma}^\top\Theta - \boldsymbol{\gamma}^{*\top}\Theta^*\|_2
$$

$$
\geq \sum_{k=1}^{d}(A_k)^2 - 2\sum_{k=1}^{d}(B_k)^2 - 4\|\boldsymbol{a}^1\|_2\|\boldsymbol{a}^2\|_2\|\boldsymbol{\gamma}^\top\Theta - \boldsymbol{\gamma}^{*\top}\Theta^*\|_2\,.
$$

Now, we analyze the first two terms on the right-hand side of the last inequality above. We use 1. the definition of $A_k$, 2. some rewritings, 3. the linearity of sums, 4. the definition of matrix product, 5. property of eigenvalues ($e_{\min}[\Theta\Theta^\top]$ denotes the smallest eigenvalue of $\Theta\Theta^\top$), and 6. the norm definition to obtain

$$
\sum_{k=1}^{d}(A_k)^2 = \sum_{k=1}^{d}\Big((\Theta_{.,k})^\top\boldsymbol{a}^1\Big)^2
$$

$$
= \sum_{k=1}^{d}(\boldsymbol{a}^1)^\top\Theta_{.,k}(\Theta_{.,k})^\top\boldsymbol{a}^1
$$

$$
\begin{aligned}
&= (\boldsymbol{a}^1)^\top \left( \sum_{k=1}^{d} \Theta_{\cdot,k}(\Theta_{\cdot,k})^\top \right) \boldsymbol{a}^1 \\
&= (\boldsymbol{a}^1)^\top \Theta\Theta^\top \boldsymbol{a}^1 \\
&\geq e_{\min}\left[\Theta\Theta^\top\right](\boldsymbol{a}^1)^\top \boldsymbol{a}^1 \\
&= e_{\min}\left[\Theta\Theta^\top\right]\|\boldsymbol{a}^1\|_2^2 .
\end{aligned}
$$

Also, using 1. the definition of $B_k$, 2. the Cauchy–Schwarz inequality, 3. the linearity of sums, and 4. the definition of norms we obtain

$$
2\sum_{k=1}^{d}(B_k)^2 = 2\sum_{k=1}^{d}\left(\boldsymbol{\gamma}^\top(\boldsymbol{a}^2)^k\right)^2 \leq 2\sum_{k=1}^{d}\|\boldsymbol{\gamma}\|_2^2\|(\boldsymbol{a}^2)^k\|_2^2 = 2\|\boldsymbol{\gamma}\|_2^2\sum_{k=1}^{d}\|(\boldsymbol{a}^2)^k\|_2^2 = 2\|\boldsymbol{\gamma}\|_2^2\|\boldsymbol{a}^2\|_2^2 .
$$

Collecting two displays above together with the earlier one we obtain

$$
\boldsymbol{a}^\top \nabla^2 \mathrm{risk}[\boldsymbol{\gamma}, \Theta]\boldsymbol{a} \geq e_{\min}\left[\Theta\Theta^\top\right]\|\boldsymbol{a}^1\|_2^2 - 2\|\boldsymbol{\gamma}\|_2^2\|\boldsymbol{a}^2\|_2^2 - 4\|\boldsymbol{a}^1\|_2\|\boldsymbol{a}^2\|_2\|\boldsymbol{\gamma}^\top\Theta - \boldsymbol{\gamma}^{*\top}\Theta^*\|_2 .
$$

Now, it is time to concentrate on the Hessian behavior of $\nabla^2 \mathrm{risk}[\boldsymbol{\gamma_\alpha}, \Theta_{\boldsymbol{\alpha}}]$ (and not $\nabla^2 \mathrm{risk}[\boldsymbol{\gamma}, \Theta]$). We use the known fact in neural networks that weights can be rescaled across the layers once activations are nonnegative-homogeneous. It says for a neural network parameterized by $(\boldsymbol{\gamma}, \Theta)$, there is another network with the same objective value such that the covariates of $\boldsymbol{\gamma}$ are multiplied by the covariates of $\boldsymbol{\alpha}$ and the covariates in each column of $\Theta$ are divided by the covariates of $\boldsymbol{\alpha}$. We use this fact with $\alpha_j = 1/c$ for all $j \in \{1, \dots, w\}$, which $c \in (1, \infty)$, together with the above result to analyze the behavior of Hessian in $(\boldsymbol{\gamma_\alpha}, \Theta_{\boldsymbol{\alpha}})$ and get

$$
\begin{aligned}
\boldsymbol{a}^\top \nabla^2 \mathrm{risk}[\boldsymbol{\gamma_\alpha}, \Theta_{\boldsymbol{\alpha}}]\boldsymbol{a} &\geq e_{\min}\left[\Theta_{\boldsymbol{\alpha}}\Theta_{\boldsymbol{\alpha}}^\top\right]\|\boldsymbol{a}^1\|_2^2 - 2\|\boldsymbol{\gamma_\alpha}\|_2^2\|\boldsymbol{a}^2\|_2^2 - 4\|\boldsymbol{a}^1\|_2\|\boldsymbol{a}^2\|_2\|\boldsymbol{\gamma_\alpha}^\top\Theta_{\boldsymbol{\alpha}} - \boldsymbol{\gamma^*_\alpha}^\top\Theta^*_{\boldsymbol{\alpha}}\|_2 \\
&= c^2 e_{\min}\left[\Theta\Theta^\top\right]\|\boldsymbol{a}^1\|_2^2 - \frac{2}{c^2}\|\boldsymbol{\gamma}\|_2^2\|\boldsymbol{a}^2\|_2^2 - 4\|\boldsymbol{a}^1\|_2\|\boldsymbol{a}^2\|_2\|\boldsymbol{\gamma}^\top\Theta - \boldsymbol{\gamma}^{*\top}\Theta^*\|_2 ,
\end{aligned}
$$

where for the last line we use factorizing and the definition of scaled parameters. Using above display, we can guarantee positive semidefinite Hessian once $c$ is selected large enough because, the first term can dominate the other two terms. So, we use $c \in [1, \infty)$ and our assumption on $\Theta\Theta^\top$ to obtain that for

$$
c^2 \geq \frac{2\|\boldsymbol{\gamma}\|_2^2\|\boldsymbol{a}^2\|_2^2 + 4\|\boldsymbol{a}^1\|_2\|\boldsymbol{a}^2\|_2\|\boldsymbol{\gamma}^\top\Theta - \boldsymbol{\gamma}^{*\top}\Theta^*\|_2}{e_{\min}\left[\Theta\Theta^\top\right]\|\boldsymbol{a}^1\|_2^2} ,
$$

we can guarantee positive semidefinite Hessian, as desired. $\qquad\square$

### B.4 Proof of Lemma 1

*Proof.* The proof idea is inspired by Elsener & van de Geer (2018, Lemma 14) and main ingredients are our Lemma 2 and union bounds.

Let's define $\tilde{r}(t) := 2t$ for $t \in (0, \infty)$, $s_{\mathcal{C}_{\eta,\epsilon}} := (\eta + \max\{\|\boldsymbol{\gamma}^*\|_\infty, \|\Theta^*\|_\infty\})(1 + \epsilon)$, which is basically defined by parameters $\epsilon$ and $\eta$ of $\mathcal{C}_{\eta,\epsilon}$ (recall that $\mathcal{C}_{\eta,\epsilon} = \{\boldsymbol{\beta} = \mathrm{vec}(\boldsymbol{\gamma}, \Theta) \in \mathbb{R}^p : \|\boldsymbol{\beta}^* - \boldsymbol{\beta}\|_1 \leq \eta \text{ and } \|\boldsymbol{\gamma}^\top\Theta - \boldsymbol{\gamma}^{*\top}\Theta^*\|_1 \leq \epsilon\}$), and $Z(\boldsymbol{\beta}, \boldsymbol{\beta}^*)$ as a function of two vectors $\boldsymbol{\beta}$ and $\boldsymbol{\beta}^*$ (with $\boldsymbol{\beta} = \mathrm{vec}(\boldsymbol{\gamma}, \Theta)$) defined as

$$
Z(\boldsymbol{\beta}, \boldsymbol{\beta}^*) := \left|\left(\nabla\mathrm{risk}_X[\boldsymbol{\gamma}, \Theta] - \nabla\mathrm{risk}[\boldsymbol{\gamma}, \Theta]\right)^\top(\boldsymbol{\beta}^* - \boldsymbol{\beta})\right| .
$$

Using Lemma 2 and notations above and with assuming $\widetilde{\boldsymbol{\beta}} \in \mathcal{C}_{\eta,\epsilon}$ (specific values of $\epsilon$ and $\eta$ be assigned at the end of the proof) we obtain for each $t \in (0, \infty)$ that

$$
\begin{aligned}
\mathbb{P}\left(Z(\widetilde{\boldsymbol{\beta}}, \boldsymbol{\beta}^*) \geq \eta\tilde{r}(t)s_{\mathcal{C}_{\eta,\epsilon}}\right) &\leq \mathbb{P}\left(\sup_{\boldsymbol{\beta} \in \mathcal{C}_{\eta,\epsilon}} Z(\boldsymbol{\beta}, \boldsymbol{\beta}^*) \geq \eta\tilde{r}(t)s_{\mathcal{C}_{\eta,\epsilon}}\right) \\
&\leq 4d^2p\exp(-\kappa n\min\{t^2/\nu^2, t/\nu\})
\end{aligned}
$$

with $\nu, \kappa \in (0, \infty)$ constants depending only on the distributions of the inputs and noise.

We assume without loss of generality that $1/n \leq \eta$ and continue the proof in two different cases:

*Case 1:* $(\|\widetilde{\boldsymbol{\beta}} - \boldsymbol{\beta}^*\|_1 \leq 1/n)$

In this case, we use 1. the fact that $\|\widetilde{\boldsymbol{\beta}} - \boldsymbol{\beta}^*\|_1 \tilde{r}(t) s_{\mathcal{C}_{\eta,\epsilon}} \geq 0$, 2. our assumption that $1/n \leq \eta$ and the definition of $s_{\mathcal{C}_{\eta,\epsilon}}$, and 3. our assumption that $\|\widetilde{\boldsymbol{\beta}} - \boldsymbol{\beta}^*\|_1 \leq 1/n$, which implies that $\widetilde{\boldsymbol{\beta}} \in \mathcal{C}_{1/n,\epsilon}$ and our argument above to obtain for each $t \in (0, \infty)$ that

$$
\begin{aligned}
\mathbb{P}\left( Z(\widetilde{\boldsymbol{\beta}}, \boldsymbol{\beta}^*) \geq 2\|\widetilde{\boldsymbol{\beta}} - \boldsymbol{\beta}^*\|_1 \tilde{r}(t) s_{\mathcal{C}_{\eta,\epsilon}} + \frac{\tilde{r}(t)}{n} s_{\mathcal{C}_{\eta,\epsilon}} \right) &\leq \mathbb{P}\left( Z(\widetilde{\boldsymbol{\beta}}, \boldsymbol{\beta}^*) \geq \frac{\tilde{r}(t)}{n} s_{\mathcal{C}_{\eta,\epsilon}} \right) \\
&\leq \mathbb{P}\left( Z(\widetilde{\boldsymbol{\beta}}, \boldsymbol{\beta}^*) \geq \frac{\tilde{r}(t)}{n} s_{\mathcal{C}_{1/n,\epsilon}} \right) \\
&\leq 4d^2 p \exp(-\kappa n \min\{t^2/\nu^2, t/\nu\}) .
\end{aligned}
$$

*Case 2:* $(1/n < \|\widetilde{\boldsymbol{\beta}} - \boldsymbol{\beta}^*\|_1 \leq \eta)$

In this case, we use 1. the fact that for mutually exclusive events $H_1, \ldots, H_n$: $\mathbb{P}(\cup_{i=1}^n H_i) = \sum_{i=1}^n \mathbb{P}(H_i)$, 2. lower bound of $\|\widetilde{\boldsymbol{\beta}} - \boldsymbol{\beta}^*\|_1$, 3. the fact that $\tilde{r}(t) s_{\mathcal{C}_{\eta,\epsilon}}/n \geq 0$ and removing the lower bound, 4. the fact that $2^{i+1}/n \leq \eta$, and 5. the fact that $\widetilde{\boldsymbol{\beta}} \in s_{\mathcal{C}_{2^{i+1}/n,\epsilon}}$ and our earlier argument to obtain for each $t \in (0, \infty)$ that

$$
\begin{aligned}
&\mathbb{P}\left( Z(\widetilde{\boldsymbol{\beta}}, \boldsymbol{\beta}^*) \geq 2\|\widetilde{\boldsymbol{\beta}} - \boldsymbol{\beta}^*\|_1 \tilde{r}(t) s_{\mathcal{C}_{\eta,\epsilon}} + \frac{\tilde{r}(t)}{n} s_{\mathcal{C}_{\eta,\epsilon}} \quad \text{for} \quad \frac{1}{n} < \|\widetilde{\boldsymbol{\beta}} - \boldsymbol{\beta}^*\|_1 \leq \eta \right) \\
&= \sum_{i=0}^{\lceil \log_2 (n\eta) \rceil - 1} \mathbb{P}\left( Z(\widetilde{\boldsymbol{\beta}}, \boldsymbol{\beta}^*) \geq 2\|\widetilde{\boldsymbol{\beta}} - \boldsymbol{\beta}^*\|_1 \tilde{r}(t) s_{\mathcal{C}_{\eta,\epsilon}} + \frac{\tilde{r}(t)}{n} s_{\mathcal{C}_{\eta,\epsilon}} \quad \text{for} \quad \frac{2^i}{n} < \|\widetilde{\boldsymbol{\beta}} - \boldsymbol{\beta}^*\|_1 \leq \frac{2^{i+1}}{n} \right) \\
&\leq \sum_{i=0}^{\lceil \log_2 (n\eta) \rceil - 1} \mathbb{P}\left( Z(\widetilde{\boldsymbol{\beta}}, \boldsymbol{\beta}^*) \geq \frac{2^{i+1}}{n} \tilde{r}(t) s_{\mathcal{C}_{\eta,\epsilon}} + \frac{\tilde{r}(t)}{n} s_{\mathcal{C}_{\eta,\epsilon}} \quad \text{for} \quad \frac{2^i}{n} < \|\widetilde{\boldsymbol{\beta}} - \boldsymbol{\beta}^*\|_1 \leq \frac{2^{i+1}}{n} \right) \\
&\leq \sum_{i=0}^{\lceil \log_2 (n\eta) \rceil - 1} \mathbb{P}\left( Z(\widetilde{\boldsymbol{\beta}}, \boldsymbol{\beta}^*) \geq \frac{2^{i+1}}{n} \tilde{r}(t) s_{\mathcal{C}_{\eta,\epsilon}} \quad \text{for} \quad \|\widetilde{\boldsymbol{\beta}} - \boldsymbol{\beta}^*\|_1 \leq \frac{2^{i+1}}{n} \right) \\
&\leq \sum_{i=0}^{\lceil \log_2 (n\eta) \rceil - 1} \mathbb{P}\left( Z(\widetilde{\boldsymbol{\beta}}, \boldsymbol{\beta}^*) \geq \frac{2^{i+1}}{n} \tilde{r}(t) s_{\mathcal{C}_{2^{i+1}/n,\epsilon}} \quad \text{for} \quad \|\widetilde{\boldsymbol{\beta}} - \boldsymbol{\beta}^*\|_1 \leq \frac{2^{i+1}}{n} \right) \\
&\leq 4\lceil \log_2 (n\eta) \rceil d^2 p \exp(-\kappa n \min\{t^2/\nu^2, t/\nu\}) .
\end{aligned}
$$

We collect all pieces of the proof (*Case 1* and *Case 2*), set $t = \nu\sqrt{\log (8nd^2p\lceil \log_2 (n\eta) \rceil)/(\kappa n)}$ (we use the notation log as natural logarithm), and use the union bounds to obtain (we also need to assume $n$ is large enough to get rid of the min operator)

$$
\begin{aligned}
\mathbb{P}\Big( Z(\widetilde{\boldsymbol{\beta}}, \boldsymbol{\beta}^*) &\geq 2\|\widetilde{\boldsymbol{\beta}} - \boldsymbol{\beta}^*\|_1 \tilde{r}\left( \nu\sqrt{\log (8nd^2p\lceil \log_2 (n\eta) \rceil)/(\kappa n)} \right) s_{\mathcal{C}_{\eta,\epsilon}} \\
&+ \frac{\tilde{r}\left( \nu\sqrt{\log (8nd^2p\lceil \log_2 (n\eta) \rceil)/(\kappa n)} \right)}{n} s_{\mathcal{C}_{\eta,\epsilon}} \Big) \\
&\leq 4\lceil \log_2 (n\eta) \rceil d^2 p \exp(- \log(8nd^2p\lceil \log_2 (n\eta) \rceil)) \\
&= \frac{1}{2n} .
\end{aligned}
$$

Now, we use the results above and the definitions of $Z(\widetilde{\boldsymbol{\beta}}, \boldsymbol{\beta}^*)$ and $\tilde{r}(t)$ to obtain

$$
\mathbb{P}\Bigg(\Big|\big(\nabla \mathrm{risk}_X[\widetilde{\boldsymbol{\gamma}}, \widetilde{\Theta}] - \nabla \mathrm{risk}[\widetilde{\boldsymbol{\gamma}}, \widetilde{\Theta}]\big)^\top(\boldsymbol{\beta}^* - \widetilde{\boldsymbol{\beta}})\Big| \geq 4\nu s_{\mathcal{C}_{\eta,\epsilon}}\|\widetilde{\boldsymbol{\beta}} - \boldsymbol{\beta}^*\|_1 \sqrt{\frac{\log\big(8nd^2p\lceil \log_2(n\eta)\rceil\big)}{\kappa n}}
$$

$$
+ 2\nu s_{\mathcal{C}_{\eta,\epsilon}}\sqrt{\frac{\log\big(8nd^2p\lceil \log_2(n\eta)\rceil\big)}{\kappa n^3}}\Bigg)
$$

$$
\leq \frac{1}{2n}.
$$

Then, we use our assumption that the stationary point $(\widetilde{\boldsymbol{\gamma}}, \widetilde{\Theta})$ is reasonable to obtain: $\|\widetilde{\boldsymbol{\gamma}}^\top\widetilde{\Theta} - \boldsymbol{\gamma}^{*\top}\Theta^*\|_1 \leq \|\widetilde{\boldsymbol{\gamma}}^\top\widetilde{\Theta}\|_1 + \|\boldsymbol{\gamma}^{*\top}\Theta^*\|_1 \leq \|\widetilde{\boldsymbol{\gamma}}\|_1\|\widetilde{\Theta}\|_\infty + \|\boldsymbol{\gamma}^*\|_1\|\Theta^*\|_\infty \leq 2\log n$ (using triangle inequality, Hölder's inequality, and our assumption on reasonable target and stationary) and $\|\widetilde{\boldsymbol{\beta}} - \boldsymbol{\beta}^*\|_1 \leq \|\widetilde{\boldsymbol{\beta}}\|_1 + \|\boldsymbol{\beta}^*\|_1 = \|\widetilde{\boldsymbol{\gamma}}\|_1 + \|\widetilde{\Theta}\|_1 + \|\boldsymbol{\gamma}^*\|_1 + \|\Theta^*\|_1 \leq 4\sqrt{\log n}$ (using triangle inequality, our definition of norm, and our assumption on reasonable target and stationary), which means we can assign $\epsilon = 2\log n$ and $\eta = 4\sqrt{\log n}$ (for $n \geq 2$ we can make sure that $1/n \leq \eta$ is satisfied).

Now, we plug in the values of $\epsilon = 2\log n$, $\eta = 4\sqrt{\log n}$, and $s_{\mathcal{C}_{\eta,\epsilon}} = (\eta + \max\{\|\boldsymbol{\gamma}^*\|_\infty, \|\Theta^*\|_\infty\})(1 + \epsilon) \leq (5\sqrt{\log n})(1 + 2\log n) \leq 15(\log n)^{3/2}$ (for $n \geq 2$) to conclude that

$$
\mathbb{P}\Bigg(\Big|\big(\nabla \mathrm{risk}_X[\widetilde{\boldsymbol{\gamma}}, \widetilde{\Theta}] - \nabla \mathrm{risk}[\widetilde{\boldsymbol{\gamma}}, \widetilde{\Theta}]\big)^\top(\boldsymbol{\beta}^* - \widetilde{\boldsymbol{\beta}})\Big|
$$

$$
\geq \frac{60}{\sqrt{\kappa n}}\nu\|\widetilde{\boldsymbol{\beta}} - \boldsymbol{\beta}^*\|_1(\log n)^{3/2}\sqrt{\log\big(8nd^2p\lceil \log_2(4n\sqrt{\log n})\rceil\big)}
$$

$$
+ \frac{30}{\sqrt{\kappa n^3}}\nu(\log n)^{3/2}\sqrt{\log\big(8nd^2p\lceil \log_2(4n\sqrt{\log n})\rceil\big)}\Bigg)
$$

$$
\leq \frac{1}{2n}.
$$

Then, we use the fact that $d \leq p$ and simplifying display above to obtain

$$
\mathbb{P}\Bigg(\Big|\big(\nabla \mathrm{risk}_X[\widetilde{\boldsymbol{\gamma}}, \widetilde{\Theta}] - \nabla \mathrm{risk}[\widetilde{\boldsymbol{\gamma}}, \widetilde{\Theta}]\big)^\top(\boldsymbol{\beta}^* - \widetilde{\boldsymbol{\beta}})\Big|
$$

$$
\geq \frac{180}{\sqrt{\kappa n}}\nu\|\widetilde{\boldsymbol{\beta}} - \boldsymbol{\beta}^*\|_1(\log n)^{3/2}\sqrt{\log(np)} + \frac{90}{\sqrt{\kappa n^3}}\nu(\log n)^{3/2}\sqrt{\log(np)}\Bigg)
$$

$$
\leq \frac{1}{2n}.
$$

We finally absorb all the constants $(180/\sqrt{\kappa})$ in $\nu$ and use the definition of $r_{\mathrm{orc}}$ to complete the proof. $\square$

### B.5 Proof of Lemma 2

*Proof.* We start the proof with Hölder's inequality and the definition of $\mathcal{C}_{\eta,\epsilon}$, which implies $\|\boldsymbol{\beta}^* - \boldsymbol{\beta}\|_1 \leq \eta$ for all $\boldsymbol{\beta} \in \mathcal{C}_{\eta,\epsilon}$ to obtain

$$
\sup_{\boldsymbol{\beta}=\mathrm{vec}(\boldsymbol{\gamma},\Theta)\in\mathcal{C}_{\eta,\epsilon}}\Big|\big(\nabla \mathrm{risk}_X[\boldsymbol{\gamma}, \Theta] - \nabla \mathrm{risk}[\boldsymbol{\gamma}, \Theta]\big)^\top(\boldsymbol{\beta}^* - \boldsymbol{\beta})\Big|
$$

$$
\leq \sup_{\boldsymbol{\beta}=\mathrm{vec}(\boldsymbol{\gamma},\Theta)\in\mathcal{C}_{\eta,\epsilon}}\big(\big\|\nabla \mathrm{risk}_X[\boldsymbol{\gamma}, \Theta] - \nabla \mathrm{risk}[\boldsymbol{\gamma}, \Theta]\big\|_\infty\|\boldsymbol{\beta}^* - \boldsymbol{\beta}\|_1\big)
$$

$$
\leq \eta \sup_{\boldsymbol{\beta}=\mathrm{vec}(\boldsymbol{\gamma},\Theta)\in\mathcal{C}_{\eta,\epsilon}}\big\|\nabla \mathrm{risk}_X[\boldsymbol{\gamma}, \Theta] - \nabla \mathrm{risk}[\boldsymbol{\gamma}, \Theta]\big\|_\infty.
$$

The rest of the proof is using our Lemma 5 and Bernstein's inequality (Vershynin, 2018, Corollary 2.8.3) to find an upper bound for $\sup_{\boldsymbol{\beta}=\text{vec}(\boldsymbol{\gamma},\Theta)\in\mathcal{C}_{\eta,\epsilon}}\|\nabla\text{risk}_X[\boldsymbol{\gamma},\Theta]-\nabla\text{risk}[\boldsymbol{\gamma},\Theta]\|_\infty$. Note that for simplifying the notation, we use $\mathbb{E}[\cdot]$ as a shorthand notation of $\mathbb{E}_{(\boldsymbol{x}_1,y_1),\ldots,(\boldsymbol{x}_n,y_n)}[\cdot]$ throughout this proof.

We use 1. our result in Lemma 5 and i.i.d. assumption on the data, 2. equation 1 and our assumption that $f[\boldsymbol{x}]=\boldsymbol{\gamma}^{*\top}\Theta^*\boldsymbol{x}$, zero-mean noise, linearity of expectations, and factorizing, 3. the definition of sup-norm, triangle inequality, and Hölder's inequality, 4. the definition of $\mathcal{C}_{\eta,\epsilon}$, which implies $\|\boldsymbol{\gamma}^{*\top}\Theta^*-\boldsymbol{\gamma}^\top\Theta\|_1\leq\epsilon$, 5. adding a zero-valued term and rewriting, and 6. the triangle inequality and the definition of $\mathcal{C}_{\eta,\epsilon}$, which implies $\|\boldsymbol{\gamma}-\boldsymbol{\gamma}^*\|_1\leq\|\boldsymbol{\beta}-\boldsymbol{\beta}^*\|_1\leq\eta$, to obtain for each $j\in\{1,\ldots,w\}$ and $k\in\{1,\ldots,d\}$ that

$$\left|\frac{\partial}{\partial\theta_{jk}}\text{risk}_X[\boldsymbol{\gamma},\Theta]-\frac{\partial}{\partial\theta_{jk}}\text{risk}[\boldsymbol{\gamma},\Theta]\right|$$

$$=\left|-\frac{2}{n}\sum_{i=1}^n(y_i-\boldsymbol{\gamma}^\top\Theta\boldsymbol{x}_i)\gamma_j(\boldsymbol{x}_i)_k+\mathbb{E}\left[\frac{2}{n}\sum_{i=1}^n(y_i-\boldsymbol{\gamma}^\top\Theta\boldsymbol{x}_i)\gamma_j(\boldsymbol{x}_i)_k\right]\right|$$

$$=2|\gamma_j|\left|\frac{1}{n}\sum_{i=1}^n\Big(u_i(\boldsymbol{x}_i)_k+(\boldsymbol{\gamma}^{*\top}\Theta^*-\boldsymbol{\gamma}^\top\Theta)\big(\boldsymbol{x}_i(\boldsymbol{x}_i)_k-\mathbb{E}[\boldsymbol{x}_i(\boldsymbol{x}_i)_k]\big)\Big)\right|$$

$$\leq2\|\boldsymbol{\gamma}\|_\infty\left(\left|\frac{1}{n}\sum_{i=1}^nu_i(\boldsymbol{x}_i)_k\right|+\|\boldsymbol{\gamma}^\top\Theta-\boldsymbol{\gamma}^{*\top}\Theta^*\|_1\left\|\frac{1}{n}\sum_{i=1}^n\big(\mathbb{E}[\boldsymbol{x}_i(\boldsymbol{x}_i)_k]-\boldsymbol{x}_i(\boldsymbol{x}_i)_k\big)\right\|_\infty\right)$$

$$\leq2\|\boldsymbol{\gamma}\|_\infty\left(\left|\frac{1}{n}\sum_{i=1}^nu_i(\boldsymbol{x}_i)_k\right|+\epsilon\left\|\frac{1}{n}\sum_{i=1}^n\big(\mathbb{E}[\boldsymbol{x}_i(\boldsymbol{x}_i)_k]-\boldsymbol{x}_i(\boldsymbol{x}_i)_k\big)\right\|_\infty\right)$$

$$=2\|\boldsymbol{\gamma}-\boldsymbol{\gamma}^*+\boldsymbol{\gamma}^*\|_\infty\left(\left|\frac{1}{n}\sum_{i=1}^nu_i(\boldsymbol{x}_i)_k\right|+\epsilon\left\|\frac{1}{n}\sum_{i=1}^n\big(\boldsymbol{x}_i(\boldsymbol{x}_i)_k-\mathbb{E}[\boldsymbol{x}_i(\boldsymbol{x}_i)_k]\big)\right\|_\infty\right)$$

$$\leq2(\eta+\|\boldsymbol{\gamma}^*\|_\infty)\left(\left|\frac{1}{n}\sum_{i=1}^nu_i(\boldsymbol{x}_i)_k\right|+\epsilon\left\|\frac{1}{n}\sum_{i=1}^n\big(\boldsymbol{x}_i(\boldsymbol{x}_i)_k-\mathbb{E}[\boldsymbol{x}_i(\boldsymbol{x}_i)_k]\big)\right\|_\infty\right).$$

We continue to work on the absolute value and sup-norm term in the last inequality above separately. For each $i\in\{1,\ldots,n\}$ and $k\in\{1,\ldots,d\}$, we use our assumptions on $\boldsymbol{x}_i$ and $u_i$ to obtain that $z_i:=u_i(\boldsymbol{x}_i)_k$ are independent and sub-exponential random variables with zero-mean (Vershynin, 2018, Lemma 2.7.7) and so, we can employ Bernstein's inequality in Vershynin (2018, Corollary 2.8.3) to obtain for each $t\in[0,\infty)$ that

$$\mathbb{P}\left(\left|\frac{1}{n}\sum_{i=1}^nu_i(\boldsymbol{x}_i)_k\right|\geq t\right)\leq2\exp(-\kappa\min\{t^2/\nu^2,t/\nu\}n)$$

with $\kappa\in(0,\infty)$ an absolute constant and $\nu:=\max_{i\in\{1,\ldots,n\}}\|u_i(\boldsymbol{x}_i)_k\|_{\psi_1}\in(0,\infty)$ a constant that depends on the distributions of $\boldsymbol{x}$ and $u$ (for a sub-exponential random variable $z$, we define $\|z\|_{\psi_1}:=\inf\{q\in(0,\infty):\mathbb{E}\exp((|z|/q))\leq2\}$).

Now we study the behavior of the sup-norm term in the last inequality of the earlier display. Let's rewrite the sup-norm in the form of a max as

$$\left\|\frac{1}{n}\sum_{i=1}^n\big(\boldsymbol{x}_i(\boldsymbol{x}_i)_k-\mathbb{E}[\boldsymbol{x}_i(\boldsymbol{x}_i)_k]\big)\right\|_\infty=\max_{k'\in\{1,\ldots,d\}}\left|\frac{1}{n}\sum_{i=1}^n\big((\boldsymbol{x}_i)_{k'}(\boldsymbol{x}_i)_k-\mathbb{E}[(\boldsymbol{x}_i)_{k'}(\boldsymbol{x}_i)_k]\big)\right|.$$

Following the same argument as earlier and for each $i\in\{1,\ldots,n\}$ and $k,k'\in\{1,\ldots,d\}$, we use our assumption on $\boldsymbol{x}_i$ to obtain that $z_i':=(\boldsymbol{x}_i)_{k'}(\boldsymbol{x}_i)_k-\mathbb{E}[(\boldsymbol{x}_i)_{k'}(\boldsymbol{x}_i)_k]$ are independent sub-exponential random variables with zero-mean and again we can employ Bernstein's inequality (Vershynin, 2018, Corollary 2.8.3) to obtain for each $t'\in[0,\infty)$ that

$$\mathbb{P}\left(\left|\frac{1}{n}\sum_{i=1}^n\big((\boldsymbol{x}_i)_{k'}(\boldsymbol{x}_i)_k-\mathbb{E}[(\boldsymbol{x}_i)_{k'}(\boldsymbol{x}_i)_k]\big)\right|\geq t'\right)\leq2\exp(-\kappa'\min\{t'^2/\nu'^2,t'/\nu'\}n)$$

with $\kappa' \in (0, \infty)$ an absolute constant and $\nu' := \max_{i \in \{1,\dots,n\}} \|(\boldsymbol{x}_i)_{k'}(\boldsymbol{x}_i)_k - \mathbb{E}[(\boldsymbol{x}_i)_{k'}(\boldsymbol{x}_i)_k]\|_{\psi_1} \in (0, \infty)$ a constant that depends on the distribution of $\boldsymbol{x}$.

Then, we use our result above together with the fact that if $\mathbb{P}(|b_i| \geq t) \leq a$ holds for all $i \in \{1,\dots p\}$, then we also have $\mathbb{P}(\max_{i \in \{1,\dots p\}} |b_i| \geq t) \leq pa$ to obtain

$$\mathbb{P}\left(\max_{k' \in \{1,\dots,d\}} \left|\frac{1}{n}\sum_{i=1}^{n}\left((\boldsymbol{x}_i)_{k'}(\boldsymbol{x}_i)_k - \mathbb{E}[(\boldsymbol{x}_i)_{k'}(\boldsymbol{x}_i)_k]\right)\right| \geq t'\right) \leq 2d\exp(-\kappa' \min\{t'^2/\nu'^2, t'/\nu'\}n).$$

Collecting all pieces above together with considering $t = t'$, we obtain for each $j \in \{1, \dots, w\}$ and $k \in \{1, \dots, d\}$ that

$$\left|\frac{\partial}{\partial \theta_{jk}}\mathrm{risk}_X[\boldsymbol{\gamma}, \Theta] - \frac{\partial}{\partial \theta_{jk}}\mathrm{risk}[\boldsymbol{\gamma}, \Theta]\right| \leq 2t(\eta + \|\boldsymbol{\gamma}^*\|_\infty)(1 + \epsilon)$$

with probability at least $1 - 2\exp(-\kappa \min\{t^2/\nu^2, t/\nu\}n) - 2d\exp(-\kappa' \min\{t^2/\nu'^2, t/\nu'\}n)$, which is obtained using the fact that

$$P(A + bD \leq t + bt) = 1 - P(A + bD > t + bt) \geq 1 - P(A > t) - P(D > t)$$

for any $b \in (0, \infty)$ and $t \in \mathbb{R}$.

Then, we follow the same argument as earlier and use 1. our result in Lemma 5 and i.i.d. assumption on the data, 2. the properties of absolute values and linearity of expectations, 3. some rewriting, 4. Hölder's inequality, 5. equation 1 and our assumptions that $f[\boldsymbol{x}] = \boldsymbol{\gamma}^{*\top}\Theta^*\boldsymbol{x}$, zero-mean noise, and definition of sup-norm, 6. triangle inequality, compatible norms (for a matrix $A \in \mathbb{R}^{d\times d}$, we define $\|A\|_{\infty,1} := \max_{k\in\{1,\dots,d\}}\sum_{k'=1}^{d}|A_{k',k}|$), and the definition of $\mathcal{C}_{\eta,\epsilon}$, which implies $\|\boldsymbol{\gamma}^{*\top}\Theta^* - \boldsymbol{\gamma}^\top\Theta\|_1 \leq \epsilon$, 7. adding a zero-valued term, 8. the triangle inequality and the definition of $\mathcal{C}_{\eta,\epsilon}$, which implies $\|\Theta - \Theta^*\|_1 \leq \|\boldsymbol{\beta} - \boldsymbol{\beta}^*\|_1 \leq \eta$ to obtain for each $j \in \{1, \dots, w\}$ that

$$\left|\frac{\partial}{\partial \gamma_j}\mathrm{risk}_X[\boldsymbol{\gamma}, \Theta] - \frac{\partial}{\partial \gamma_j}\mathrm{risk}[\boldsymbol{\gamma}, \Theta]\right|$$

$$= \left|-\frac{2}{n}\sum_{i=1}^{n}\left((y_i - \boldsymbol{\gamma}^\top\Theta\boldsymbol{x}_i)(\Theta\boldsymbol{x}_i)_j\right) + \mathbb{E}\left[\frac{2}{n}\sum_{i=1}^{n}\left((y_i - \boldsymbol{\gamma}^\top\Theta\boldsymbol{x}_i)(\Theta\boldsymbol{x}_i)_j\right)\right]\right|$$

$$= \left|\frac{2}{n}\sum_{i=1}^{n}\left((y_i - \boldsymbol{\gamma}^\top\Theta\boldsymbol{x}_i)(\Theta\boldsymbol{x}_i)_j - \mathbb{E}[(y_i - \boldsymbol{\gamma}^\top\Theta\boldsymbol{x}_i)(\Theta\boldsymbol{x}_i)_j]\right)\right|$$

$$= \left|\frac{2}{n}\sum_{i=1}^{n}\left((y_i - \boldsymbol{\gamma}^\top\Theta\boldsymbol{x}_i)\boldsymbol{x}_i^\top\Theta_{j,\cdot} - \mathbb{E}[(y_i - \boldsymbol{\gamma}^\top\Theta\boldsymbol{x}_i)\boldsymbol{x}_i^\top\Theta_{j,\cdot}]\right)\right|$$

$$\leq \left\|\frac{2}{n}\sum_{i=1}^{n}\left((y_i - \boldsymbol{\gamma}^\top\Theta\boldsymbol{x}_i)\boldsymbol{x}_i^\top - \mathbb{E}[(y_i - \boldsymbol{\gamma}^\top\Theta\boldsymbol{x}_i)\boldsymbol{x}_i^\top]\right)\right\|_\infty \|\Theta_{j,\cdot}\|_1$$

$$\leq 2\|\Theta\|_\infty \left(\left\|\frac{1}{n}\sum_{i=1}^{n}\left(u_i\boldsymbol{x}_i^\top + (\boldsymbol{\gamma}^{*\top}\Theta^* - \boldsymbol{\gamma}^\top\Theta)(\boldsymbol{x}_i\boldsymbol{x}_i^\top - \mathbb{E}[\boldsymbol{x}_i\boldsymbol{x}_i^\top])\right)\right\|_\infty\right.$$

$$\leq 2\|\Theta\|_\infty \left(\left\|\frac{1}{n}\sum_{i=1}^{n}u_i\boldsymbol{x}_i^\top\right\|_\infty + \epsilon\left\|\frac{1}{n}\sum_{i=1}^{n}(\boldsymbol{x}_i\boldsymbol{x}_i^\top - \mathbb{E}[\boldsymbol{x}_i\boldsymbol{x}_i^\top])\right\|_{\infty,1}\right)$$

$$\leq 2\|\Theta - \Theta^* + \Theta^*\|_\infty \left(\left\|\frac{1}{n}\sum_{i=1}^{n}u_i\boldsymbol{x}_i^\top\right\|_\infty + \epsilon\left\|\frac{1}{n}\sum_{i=1}^{n}(\boldsymbol{x}_i\boldsymbol{x}_i^\top - \mathbb{E}[\boldsymbol{x}_i\boldsymbol{x}_i^\top])\right\|_{\infty,1}\right)$$

$$\leq 2(\eta + \|\Theta^*\|_\infty)\left(\left\|\frac{1}{n}\sum_{i=1}^{n}u_i\boldsymbol{x}_i^\top\right\|_\infty + \epsilon\left\|\frac{1}{n}\sum_{i=1}^{n}(\boldsymbol{x}_i\boldsymbol{x}_i^\top - \mathbb{E}[\boldsymbol{x}_i\boldsymbol{x}_i^\top])\right\|_{\infty,1}\right).$$

Then, we use the same argument as earlier to treat the sup-norm terms above (we use our assumptions on $\boldsymbol{x}_i$ and $u_i$ and application of Bernstein's inequality) to obtain that

$$\left|\frac{\partial}{\partial \gamma_j}\mathrm{risk}_X[\boldsymbol{\gamma}, \Theta] - \frac{\partial}{\partial \gamma_j}\mathrm{risk}[\boldsymbol{\gamma}, \Theta]\right| \leq 2t(\eta + \|\Theta^*\|_\infty)(1 + \epsilon)$$

with probability at least $1 - 2d \exp(-\kappa \min\{t^2/\nu^2, t/\nu\}n) - 2d^2 \exp(-\kappa' \min\{t^2/\nu'^2, t/\nu'\}n)$ ($\kappa$, $\nu$, $\kappa'$, $\nu'$ are constants depending only on the distributions of the inputs and the noise).

Collecting all the pieces above, we obtain that for each $i \in \{1, \ldots, p\}$ the corresponding gradient difference is bounded ($|(\nabla \text{risk}_X[\boldsymbol{\gamma}, \Theta] - \nabla \text{risk}[\boldsymbol{\gamma}, \Theta])_i| \leq 2t(\eta + \max\{\|\boldsymbol{\gamma}^*\|_\infty, \|\Theta^*\|_\infty\})(1 + \epsilon)$) with probability at least $1 - 4d^2 \exp(-\kappa_{u,\boldsymbol{x}} \min\{t^2/(\nu_{u,\boldsymbol{x}})^2, t/\nu_{u,\boldsymbol{x}}\}n)$ with $\nu_{u,\boldsymbol{x}} := \max\{\nu, \nu'\}$ and $\kappa_{u,\boldsymbol{x}} := \min\{\kappa, \kappa'\}$ ($\nu_{u,\boldsymbol{x}}$ and $\kappa_{u,\boldsymbol{x}}$ are constants depending only on the distributions of the inputs and noise).

Now we use 1. the definition of sup-norm and 2. our results above together with our earlier argument about implying max operator (note that the gradient vector is of dimension $p$) to obtain for each $t \in [0, \infty)$ that

$$
\sup_{\boldsymbol{\beta}=\text{vec}(\boldsymbol{\gamma}, \Theta) \in \mathcal{C}_{\eta, \epsilon}} \left\| \nabla \text{risk}_X[\boldsymbol{\gamma}, \Theta] - \nabla \text{risk}[\boldsymbol{\gamma}, \Theta] \right\|_\infty
$$
$$
= \sup_{\boldsymbol{\beta}=\text{vec}(\boldsymbol{\gamma}, \Theta) \in \mathcal{C}_{\eta, \epsilon}} \max_{i \in \{1, \ldots, p\}} \left| \left( \nabla \text{risk}_X[\boldsymbol{\gamma}, \Theta] - \nabla \text{risk}[\boldsymbol{\gamma}, \Theta] \right)_i \right|
$$
$$
\leq 2t \left( \eta + \max\{\|\boldsymbol{\gamma}^*\|_\infty, \|\Theta^*\|_\infty\} \right) \left( 1 + \epsilon \right)
$$

with probability at least $1 - 4d^2 p \exp(-\kappa_{u,\boldsymbol{x}} \min\{t^2/(\nu_{u,\boldsymbol{x}})^2, t/\nu_{u,\boldsymbol{x}}\}n)$.

Collecting all pieces of the proof, we obtain for each $t \in [0, \infty)$ that

$$
\sup_{\boldsymbol{\beta}=\text{vec}(\boldsymbol{\gamma}, \Theta) \in \mathcal{C}_{\eta, \epsilon}} \left| \left( \nabla \text{risk}_X[\boldsymbol{\gamma}, \Theta] - \nabla \text{risk}[\boldsymbol{\gamma}, \Theta] \right)^\top (\boldsymbol{\beta}^* - \boldsymbol{\beta}) \right|
$$
$$
\leq \eta \sup_{\boldsymbol{\beta}=\text{vec}(\boldsymbol{\gamma}, \Theta) \in \mathcal{C}_{\eta, \epsilon}} \left\| \nabla \text{risk}_X[\boldsymbol{\gamma}, \Theta] - \nabla \text{risk}[\boldsymbol{\gamma}, \Theta] \right\|_\infty
$$
$$
\leq 2t\eta \left( \eta + \max\{\|\boldsymbol{\gamma}^*\|_\infty, \|\Theta^*\|_\infty\} \right) \left( 1 + \epsilon \right)
$$

with probability at least $1 - 4d^2 p \exp(-\kappa_{u,\boldsymbol{x}} \min\{t^2/(\nu_{u,\boldsymbol{x}})^2, t/\nu_{u,\boldsymbol{x}}\}n)$, where for the ease of notations we replace $\kappa_{u,\boldsymbol{x}}$ and $\nu_{u,\boldsymbol{x}}$ with $\nu$ and $\kappa$ (constants depending only on the distributions of the inputs and noise) in the statement of the lemma. $\qquad \square$

## B.6   Proof of Lemma 3

*Proof.* The main ingredients of the proof are symmetrization of probabilities (van de Geer, 2016, Lemma 16.1) and Bernstein's inequality (Vershynin, 2018, Corollary 2.8.3).

We note that for simplifying the notations, we use $\mathbb{E}[\cdot]$ as a shorthand notation of $\mathbb{E}_{(\boldsymbol{x}_1, y_1), \ldots, (\boldsymbol{x}_n, y_n)}[\cdot]$ throughout this proof.

Let's start the proof and use 1. the definition of $\text{risk}_X[\boldsymbol{\gamma}, \Theta]$ and $\text{risk}[\boldsymbol{\gamma}, \Theta]$, 2. the i.i.d. assumption on the data and that $y_i = \boldsymbol{\gamma}^{*\top} \Theta^* \boldsymbol{x}_i + u_i$, 3. expanding the squared-terms and rearranging, and 4. the triangle inequality to obtain

$$
\sup_{(\boldsymbol{\gamma}, \Theta) \in \mathcal{B}} \left| \text{risk}_X[\boldsymbol{\gamma}, \Theta] - \text{risk}[\boldsymbol{\gamma}, \Theta] \right|
$$
$$
= \sup_{(\boldsymbol{\gamma}, \Theta) \in \mathcal{B}} \left| \frac{1}{n} \sum_{i=1}^n \left( (y_i - \boldsymbol{\gamma}^\top \Theta \boldsymbol{x}_i)^2 \right) - \mathbb{E}_{(\boldsymbol{x}, y)} \left[ (y - \boldsymbol{\gamma}^\top \Theta \boldsymbol{x})^2 \right] \right|
$$
$$
= \sup_{(\boldsymbol{\gamma}, \Theta) \in \mathcal{B}} \left| \frac{1}{n} \sum_{i=1}^n \left( (\boldsymbol{\gamma}^{*\top} \Theta^* \boldsymbol{x}_i + u_i - \boldsymbol{\gamma}^\top \Theta \boldsymbol{x}_i)^2 - \mathbb{E}\left[ (\boldsymbol{\gamma}^{*\top} \Theta^* \boldsymbol{x}_i + u_i - \boldsymbol{\gamma}^\top \Theta \boldsymbol{x}_i)^2 \right] \right) \right|
$$
$$
= \sup_{(\boldsymbol{\gamma}, \Theta) \in \mathcal{B}} \left| \frac{1}{n} \sum_{i=1}^n \left( (\boldsymbol{\gamma}^{*\top} \Theta^* \boldsymbol{x}_i - \boldsymbol{\gamma}^\top \Theta \boldsymbol{x}_i)^2 - \mathbb{E}\left[ (\boldsymbol{\gamma}^{*\top} \Theta^* \boldsymbol{x}_i - \boldsymbol{\gamma}^\top \Theta \boldsymbol{x}_i)^2 \right] \right) \right.
$$
$$
\left. + 2 \left( (\boldsymbol{\gamma}^{*\top} \Theta^* \boldsymbol{x}_i - \boldsymbol{\gamma}^\top \Theta \boldsymbol{x}_i) u_i - \mathbb{E}\left[ (\boldsymbol{\gamma}^{*\top} \Theta^* \boldsymbol{x}_i - \boldsymbol{\gamma}^\top \Theta \boldsymbol{x}_i) u_i \right] \right) + (u_i{}^2 - \mathbb{E}[u_i{}^2]) \right|
$$
$$
\leq \sup_{(\boldsymbol{\gamma}, \Theta) \in \mathcal{B}} \left| \frac{1}{n} \sum_{i=1}^n \left( (\boldsymbol{\gamma}^{*\top} \Theta^* \boldsymbol{x}_i - \boldsymbol{\gamma}^\top \Theta \boldsymbol{x}_i)^2 - \mathbb{E}\left[ (\boldsymbol{\gamma}^{*\top} \Theta^* \boldsymbol{x}_i - \boldsymbol{\gamma}^\top \Theta \boldsymbol{x}_i)^2 \right] \right) \right|
$$

$$+ 2 \sup_{(\boldsymbol{\gamma},\Theta)\in\mathcal{B}} \left| \frac{1}{n}\sum_{i=1}^{n}\left(\left(\boldsymbol{\gamma}^{*\top}\Theta^*\boldsymbol{x}_i - \boldsymbol{\gamma}^\top\Theta\boldsymbol{x}_i\right)u_i - \mathbb{E}\left[\left(\boldsymbol{\gamma}^{*\top}\Theta^*\boldsymbol{x}_i - \boldsymbol{\gamma}^\top\Theta\boldsymbol{x}_i\right)u_i\right]\right)\right|$$

$$+ \left| \frac{1}{n}\sum_{i=1}^{n}\left({u_i}^2 - \mathbb{E}[{u_i}^2]\right)\right|.$$

Now, we continue to work on each term in the last inequality above separately in steps:

*Step 1:* Using Vershynin (2018, Corollary 2.8.3) together with our assumption on noise, which implies the squared of Gaussian noise is sub-exponential, we obtain for each $\bar{t}\in[0,\infty)$ that

$$\mathbb{P}\left(\left|\frac{1}{n}\sum_{i=1}^{n}({u_i}^2 - \mathbb{E}[{u_i}^2])\right| \geq \bar{t}\right) \leq 2\exp(-\kappa\min\{\bar{t}^2/\nu^2, \bar{t}/\nu\}n),$$

where $\kappa,\nu\in(0,\infty)$ are constants depending only on the distribution of the noise (our constants $\kappa$ and $\nu$ may change from line to line in this proof, but they constantly depend just on the distribution of the inputs or noise or both).

*Step 2:* We now prepare the application of van de Geer (2016, Lemma 16.1). Let's 1. define $\mathcal{R}^2$ and 2. use Hölder's inequality and factorizing to obtain

$$\mathcal{R}^2 := \sup_{(\boldsymbol{\gamma},\Theta)\in\mathcal{B}} \frac{1}{n}\sum_{i=1}^{n}\mathbb{E}\left[\left(\boldsymbol{\gamma}^{*\top}\Theta^*\boldsymbol{x}_i - \boldsymbol{\gamma}^\top\Theta\boldsymbol{x}_i\right)^4\right]$$

$$\leq \sup_{(\boldsymbol{\gamma},\Theta)\in\mathcal{B}} \left\|\boldsymbol{\gamma}^{*\top}\Theta^* - \boldsymbol{\gamma}^\top\Theta\right\|_1^4 \frac{1}{n}\sum_{i=1}^{n}\mathbb{E}\left[\|\boldsymbol{x}_i\|_\infty^4\right].$$

We also employ some linear algebra together with compatible norms (for a matrix $A\in\mathbb{R}^{d\times d}$, we define $\|A\|_{\infty,1} := \max_{k\in\{1,\ldots,d\}}\sum_{k'=1}^{d}|A_{k',k}|)$ to obtain

$$\left|\frac{1}{n}\sum_{i=1}^{n}\zeta_i\left(\boldsymbol{\gamma}^{*\top}\Theta^*\boldsymbol{x}_i - \boldsymbol{\gamma}^\top\Theta\boldsymbol{x}_i\right)^2\right| = \left|\frac{1}{n}\sum_{i=1}^{n}\left(\boldsymbol{\gamma}^{*\top}\Theta^*\boldsymbol{x}_i - \boldsymbol{\gamma}^\top\Theta\boldsymbol{x}_i\right)\zeta_i\left(\boldsymbol{\gamma}^{*\top}\Theta^*\boldsymbol{x}_i - \boldsymbol{\gamma}^\top\Theta\boldsymbol{x}_i\right)^\top\right|$$

$$= \left|\frac{1}{n}\sum_{i=1}^{n}\left(\boldsymbol{\gamma}^{*\top}\Theta^* - \boldsymbol{\gamma}^\top\Theta\right)\boldsymbol{x}_i\zeta_i\boldsymbol{x}_i^\top\left(\boldsymbol{\gamma}^{*\top}\Theta^* - \boldsymbol{\gamma}^\top\Theta\right)^\top\right|$$

$$\leq \left\|\left(\boldsymbol{\gamma}^{*\top}\Theta^* - \boldsymbol{\gamma}^\top\Theta\right)^2\right\|_\infty \left\|\frac{1}{n}\sum_{i=1}^{n}\zeta_i\boldsymbol{x}_i\boldsymbol{x}_i^\top\right\|_{\infty,1}.$$

Then, we use 1. symmetrization of probabilities (van de Geer, 2016, Lemma 16.1) with $\mathcal{R}$ as defined earlier, 2. the display above, 3. our assumption that $\sup_{(\boldsymbol{\gamma},\Theta)\in\mathcal{B}}\|(\boldsymbol{\gamma}^{*\top}\Theta^* - \boldsymbol{\gamma}^\top\Theta)^2\|_\infty \leq \epsilon'$ and rearranging, 4. the definition of $\ell_{\infty,1}$-norm for a matrix above, 5. the fact that if $\mathbb{P}(|b_i| \geq t) \leq a$ holds for all $i\in\{1,\ldots d\}$, then we also have $\mathbb{P}(\max_{i\in\{1,\ldots d\}}|b_i| \geq t) \leq da$ (for $k\in\{1,\ldots,d\}$), 6. the fact that for a vector $\boldsymbol{a}\in\mathbb{R}^d$, $\mathbb{P}(\sum_{i=1}^{d}|\boldsymbol{a}_i| \geq t) \leq d\max_{k\in\{1,\ldots,d\}}\mathbb{P}(|\boldsymbol{a}_k| \geq t)$, and 7. our assumption on $\boldsymbol{x}$ (to get rid of max term) together with Vershynin (2018, Corollary 2.8.3) to obtain for each $t\in[0,\infty)$ that

$$\mathbb{P}\left(\sup_{(\boldsymbol{\gamma},\Theta)\in\mathcal{B}} \left|\frac{1}{n}\sum_{i=1}^{n}\left(\left(\boldsymbol{\gamma}^{*\top}\Theta^*\boldsymbol{x}_i - \boldsymbol{\gamma}^\top\Theta\boldsymbol{x}_i\right)^2 - \mathbb{E}\left[\left(\boldsymbol{\gamma}^{*\top}\Theta^*\boldsymbol{x}_i - \boldsymbol{\gamma}^\top\Theta\boldsymbol{x}_i\right)^2\right]\right)\right| \geq 4\mathcal{R}\sqrt{\frac{2t}{n}}\right)$$

$$\leq 4\mathbb{P}\left(\sup_{(\boldsymbol{\gamma},\Theta)\in\mathcal{B}} \left|\frac{1}{n}\sum_{i=1}^{n}\zeta_i\left(\boldsymbol{\gamma}^{*\top}\Theta^*\boldsymbol{x}_i - \boldsymbol{\gamma}^\top\Theta\boldsymbol{x}_i\right)^2\right| \geq \mathcal{R}\sqrt{\frac{2t}{n}}\right)$$

$$\leq 4\mathbb{P}\left(\sup_{(\boldsymbol{\gamma},\Theta)\in\mathcal{B}} \left\|\left(\boldsymbol{\gamma}^{*\top}\Theta^* - \boldsymbol{\gamma}^\top\Theta\right)^2\right\|_\infty \left\|\frac{1}{n}\sum_{i=1}^{n}\zeta_i\boldsymbol{x}_i\boldsymbol{x}_i^\top\right\|_{\infty,1} \geq \mathcal{R}\sqrt{\frac{2t}{n}}\right)$$

$$\leq 4\mathbb{P}\left(\left\|\frac{1}{n}\sum_{i=1}^{n}\zeta_i\boldsymbol{x}_i\boldsymbol{x}_i^\top\right\|_{\infty,1} \geq \frac{\mathcal{R}}{\epsilon'}\sqrt{\frac{2t}{n}}\right)$$

$$\leq\ 4\mathbb{P}\left(\max_{k\in\{1,\dots,d\}}\sum_{k'=1}^{d}\left|\frac{1}{n}\sum_{i=1}^{n}\zeta_i(\boldsymbol{x}_i)_{k'}(\boldsymbol{x}_i)_k\right|\ \geq\ \frac{\mathcal{R}}{\epsilon'}\sqrt{\frac{2t}{n}}\right)$$

$$\leq\ 4d\mathbb{P}\left(\sum_{k'=1}^{d}\left|\frac{1}{n}\sum_{i=1}^{n}\zeta_i(\boldsymbol{x}_i)_{k'}(\boldsymbol{x}_i)_k\right|\ \geq\ \frac{\mathcal{R}}{\epsilon'}\sqrt{\frac{2t}{n}}\right)$$

$$\leq\ 4d^2\max_{k'\in\{1,\dots,d\}}\mathbb{P}\left(\left|\frac{1}{n}\sum_{i=1}^{n}\zeta_i(\boldsymbol{x}_i)_{k'}(\boldsymbol{x}_i)_k\right|\ \geq\ \frac{\mathcal{R}}{\epsilon'}\sqrt{\frac{2t}{n}}=:t''\right)$$

$$\leq\ 8d^2\exp(-\kappa\min\{t''^2/\nu^2, t''/\nu\}n)\,,$$

where $\kappa,\nu\in(0,\infty)$ are constants depending only on the distribution of the inputs.

Collecting results above, we obtain for each $t''\in[0,\infty)$ that

$$\mathbb{P}\left(\left|\frac{1}{n}\sum_{i=1}^{n}\left((\boldsymbol{\gamma}^{*\top}\Theta^*\boldsymbol{x}_i-\boldsymbol{\gamma}^\top\Theta\boldsymbol{x}_i)^2-\mathbb{E}\left[(\boldsymbol{\gamma}^{*\top}\Theta^*\boldsymbol{x}_i-\boldsymbol{\gamma}^\top\Theta\boldsymbol{x}_i)^2\right]\right)\right|\ \geq\ 4\epsilon' t''\right)$$
$$\leq\ 8d^2\exp(-\kappa\min\{t''^2/\nu^2, t''/\nu\}n)\,.$$

*Step 3:* Let's define $(\mathcal{R}')^2$ and use Hölder's inequality to obtain

$$(\mathcal{R}')^2\ :=\ \sup_{(\boldsymbol{\gamma},\Theta)\in\mathcal{B}}\frac{1}{n}\sum_{i=1}^{n}\mathbb{E}\left[\left((\boldsymbol{\gamma}^{*\top}\Theta^*\boldsymbol{x}_i-\boldsymbol{\gamma}^\top\Theta\boldsymbol{x}_i)u_i\right)^2\right]$$

$$\leq\ \sup_{(\boldsymbol{\gamma},\Theta)\in\mathcal{B}}\left\|\boldsymbol{\gamma}^{*\top}\Theta^*-\boldsymbol{\gamma}^\top\Theta\right\|_1^2\frac{1}{n}\sum_{i=1}^{n}\mathbb{E}\left[\|\boldsymbol{x}_i u_i\|_\infty^2\right]\,.$$

Then, we use 1. symmetrization of probabilities (van de Geer, 2016, Lemma 16.1) with $\mathcal{R}'$ defined as above, 2. Hölder's inequality, 3. our assumption that $\sup_{(\boldsymbol{\gamma},\Theta)\in\mathcal{B}}\|(\boldsymbol{\gamma}^{*\top}\Theta^*-\boldsymbol{\gamma}^\top\Theta)^2\|_\infty\ \leq\ \epsilon'$, the fact that for a vector $\boldsymbol{a}\in\mathbb{R}^d$, $\mathbb{P}(\|\boldsymbol{a}\|_1\ \geq\ t)\ \leq\ d\max_{i\in\{1,\dots,d\}}\mathbb{P}(|\boldsymbol{a}_i|\ \geq\ t)\ \leq\ d^2\mathbb{P}(|\boldsymbol{a}_i|\ \geq\ t)$, and the assumption on inputs (for $k\in\{1,\dots,d\}$), and 4. Vershynin (2018, Corollary 2.8.3) together with our assumptions on the input and noise to obtain for each $t'\in[0,\infty)$ that

$$\mathbb{P}\left(\sup_{(\boldsymbol{\gamma},\Theta)\in\mathcal{B}}\left|\frac{1}{n}\sum_{i=1}^{n}\left((\boldsymbol{\gamma}^{*\top}\Theta^*\boldsymbol{x}_i-\boldsymbol{\gamma}^\top\Theta\boldsymbol{x}_i)u_i-\mathbb{E}\left[(\boldsymbol{\gamma}^{*\top}\Theta^*\boldsymbol{x}_i-\boldsymbol{\gamma}^\top\Theta\boldsymbol{x}_i)u_i\right]\right)\right|\ \geq\ 4\mathcal{R}'\sqrt{\frac{2t'}{n}}\right)$$

$$\leq\ 4\mathbb{P}\left(\sup_{(\boldsymbol{\gamma},\Theta)\in\mathcal{B}}\left|\frac{1}{n}\sum_{i=1}^{n}\zeta_i(\boldsymbol{\gamma}^{*\top}\Theta^*\boldsymbol{x}_i-\boldsymbol{\gamma}^\top\Theta\boldsymbol{x}_i)u_i\right|\ \geq\ \mathcal{R}'\sqrt{\frac{2t'}{n}}\right)$$

$$\leq\ 4\mathbb{P}\left(\sup_{(\boldsymbol{\gamma},\Theta)\in\mathcal{B}}\left\|\boldsymbol{\gamma}^{*\top}\Theta^*-\boldsymbol{\gamma}^\top\Theta\right\|_\infty\left\|\frac{1}{n}\sum_{i=1}^{n}\zeta_i\boldsymbol{x}_i u_i\right\|_1\ \geq\ \mathcal{R}'\sqrt{\frac{2t'}{n}}\right)$$

$$\leq\ 4d^2\ \mathbb{P}\left(\left|\frac{1}{n}\sum_{i=1}^{n}\zeta_i(\boldsymbol{x}_i)_k u_i\right|\ \geq\ \mathcal{R}'\sqrt{\frac{2t'}{\epsilon'n}}=:t'''\right)$$

$$\leq\ 8d^2\exp(-\kappa\min\{t'''^2/\nu^2, t'''/\nu\}n)\,,$$

where $\kappa,\nu\in(0,\infty)$ are constants depending only on the distributions of the inputs and noise.

Collecting results above we obtain that

$$\mathbb{P}\left(\left|\frac{1}{n}\sum_{i=1}^{n}\left((\boldsymbol{\gamma}^{*\top}\Theta^*\boldsymbol{x}_i-\boldsymbol{\gamma}^\top\Theta\boldsymbol{x}_i)u_i-\mathbb{E}\left[(\boldsymbol{\gamma}^{*\top}\Theta^*\boldsymbol{x}_i-\boldsymbol{\gamma}^\top\Theta\boldsymbol{x}_i)u_i\right]\right)\right|\ \geq\ 4\sqrt{\epsilon'}t'''\right)$$
$$\leq\ 8d^2\exp(-\kappa\min\{t'''^2/\nu^2, t'''/\nu\}n)\,,$$

where $\kappa,\nu\in(0,\infty)$ are constants depending only on the distributions of the inputs and noise.

Collecting all the pieces of the proof in steps 1:3, we obtain for each $t \in [0, \infty)$ that

$$\sup_{(\boldsymbol{\gamma}, \Theta) \in \mathcal{B}} \left| \text{risk}_X[\boldsymbol{\gamma}, \Theta] - \text{risk}[\boldsymbol{\gamma}, \Theta] \right| \le t\left(1 + 4\epsilon' + 4\sqrt{\epsilon'}\right)$$

with probability at least $1 - (2 + 8d^2 + 8d^2) \exp(-\kappa \min\{t^2/\nu^2, t/\nu\}n)$ or by rewriting as $1 - 18d^2$ $\exp(-\kappa \min\{t^2/\nu^2, t/\nu\}n)$ (using the assumption that $d \ge 1$), where we consider $t = \bar{t} = t'' = t'''$ and $\kappa, \nu \in (0, \infty)$ are constants depending only on the distributions of the inputs and noise. $\qquad\square$

## B.7 Proof of Lemma 4

*Proof.* The proof follows just basic linear algebra.

Since $H(t)$ is invertible exactly when $(A + tC)^\top$ has full (column) rank, we are left to study the rank of $(A + tC)^\top = A^\top + tC^\top$. To do so, we employ the Singular Value Decomposition (SVD) of $A^T \in \mathbb{R}^{d' \times w'}$, that is, $A^\top = UDV^\top$ with $U \in \mathbb{R}^{d' \times w'}$, $V \in \mathbb{R}^{w' \times w'}$, and $D \in \mathbb{R}^{w' \times w'}$ that $U, V$ are semi-orthogonal matrices and $D$ has the same rank as $A$, in this case, full rank. Now, we are motivated to make a squared matrix as

$$U^\top(A^\top + tC^\top)V = U^\top(UDV^\top + tC^\top)V = D + tU^\top C^\top V = tD(t^{-1}I_{w'} + D^{-1}U^\top C^\top V),$$

where we used the SVD form of matrix $A$, orthogonal property of $U, V$, and some rewriting. Since matrices $U$ and $V$ have rank $w$, for studying the rank of $A^\top + tC^\top$ it is enough to study determinant of $U^\top(A^\top + tC^\top)V$. We then use our display above, properties of determinants for squared matrices, and characteristic polynomials to obtain

$$\begin{aligned}
\det\left(U^\top(A^\top + tC^\top)V\right) &= \det\left(tD(t^{-1}I_{w'} + D^{-1}U^\top C^\top V)\right) \\
&= \det(tD)\det\left(t^{-1}I_{w'} + D^{-1}U^\top C^\top V\right) \\
&= t^{w'}\det(D)p_{Z:=D^{-1}U^\top C^\top V}(-t^{-1}).
\end{aligned}$$

Since, $\det(D) \ne 0$ and $t \ne 0$, then the $t$ which $H(t)$ is singular are the roots of $p_Z(-t^{-1})$, where $Z = D^{-1}U^\top C^\top V$. Since the roots of $p_Z$ are the eigenvalues of $Z$, we have found that the only $t$ for which $H(t)$ fails to be invertible are the negative reciprocals of the (nonzero) eigenvalues of $Z$. Since, any $w' \times w'$ matrix has at most $w'$ distinct eigenvalues, there are just finitely many $t$ such that $H(t)$ is not invertible, as desired. $\qquad\square$

## B.8 Proof of Lemma 5

*Proof.* The proof consists of basic algebra.

*Claim 1:* We use 1. the definition of $\text{risk}_X[\boldsymbol{\gamma}, \Theta]$, 2. the chain rule, and 3. taking the derivatives to obtain

$$\begin{aligned}
\frac{\partial}{\partial \gamma_j}\text{risk}_X[\boldsymbol{\gamma}, \Theta] &= \frac{\partial}{\partial \gamma_j}\left(\frac{1}{n}\sum_{i=1}^{n}(y_i - \boldsymbol{\gamma}^\top \Theta \boldsymbol{x}_i)^2\right) \\
&= -\frac{2}{n}\sum_{i=1}^{n}\left((y_i - \boldsymbol{\gamma}^\top \Theta \boldsymbol{x}_i)\frac{\partial}{\partial \gamma_j}(\boldsymbol{\gamma}^\top \Theta \boldsymbol{x}_i)\right) \\
&= -\frac{2}{n}\sum_{i=1}^{n}\left((y_i - \boldsymbol{\gamma}^\top \Theta \boldsymbol{x}_i)(\Theta \boldsymbol{x}_i)_j\right),
\end{aligned}$$

as desired.

*Claim 2:* We use 1. the definition of $\mathrm{risk}_X[\boldsymbol{\gamma}, \Theta]$, 2. the chain rule, and 3. taking the derivatives to obtain

$$
\begin{aligned}
\frac{\partial}{\partial \theta_{jk}} \mathrm{risk}_X[\boldsymbol{\gamma}, \Theta] &= \frac{\partial}{\partial \theta_{jk}} \left( \frac{1}{n} \sum_{i=1}^{n} (y_i - \boldsymbol{\gamma}^\top \Theta \boldsymbol{x}_i)^2 \right) \\
&= -\frac{2}{n} \sum_{i=1}^{n} \left( (y_i - \boldsymbol{\gamma}^\top \Theta \boldsymbol{x}_i) \frac{\partial}{\partial \theta_{jk}} (\boldsymbol{\gamma}^\top \Theta \boldsymbol{x}_i) \right) \\
&= -\frac{2}{n} \sum_{i=1}^{n} \left( (y_i - \boldsymbol{\gamma}^\top \Theta \boldsymbol{x}_i) \gamma_j (\boldsymbol{x}_i)_k \right),
\end{aligned}
$$

as desired.

*Claim 3:* We 1. use Claim 1 and 2. remove the term with zero derivatives and use the chain rule to obtain

$$
\begin{aligned}
\frac{\partial^2}{\partial \gamma_{j'} \partial \gamma_j} \mathrm{risk}_X[\boldsymbol{\gamma}, \Theta] &= \frac{\partial}{\partial \gamma_{j'}} \left( -\frac{2}{n} \sum_{i=1}^{n} \left( (y_i - \boldsymbol{\gamma}^\top \Theta \boldsymbol{x}_i)(\Theta \boldsymbol{x}_i)_j \right) \right) \\
&= \frac{2}{n} \sum_{i=1}^{n} \left( (\Theta \boldsymbol{x}_i)_{j'} (\Theta \boldsymbol{x}_i)_j \right),
\end{aligned}
$$

as desired.

*Claim 4:* We 1. use Claim 2, 2. remove the term with zero derivatives, and 3. compute the derivative of the bracket, and 4. rearranging to obtain

$$
\begin{aligned}
\frac{\partial^2}{\partial \theta_{j'k'} \partial \theta_{jk}} \mathrm{risk}_X[\boldsymbol{\gamma}, \Theta] &= \frac{\partial}{\partial \theta_{j'k'}} \left( -\frac{2}{n} \sum_{i=1}^{n} \left( (y_i - \boldsymbol{\gamma}^\top \Theta \boldsymbol{x}_i) \gamma_j (\boldsymbol{x}_i)_k \right) \right) \\
&= \frac{\partial}{\partial \theta_{j'k'}} \left( \frac{2}{n} \gamma_j \sum_{i=1}^{n} \left( (\boldsymbol{\gamma}^\top \Theta \boldsymbol{x}_i)(\boldsymbol{x}_i)_k \right) \right) \\
&= \frac{2}{n} \gamma_j \sum_{i=1}^{n} \left( \gamma_{j'} (\boldsymbol{x}_i)_{k'} (\boldsymbol{x}_i)_k \right) \\
&= \frac{2}{n} \gamma_{j'} \gamma_j \sum_{i=1}^{n} \left( (\boldsymbol{x}_i)_{k'} (\boldsymbol{x}_i)_k \right),
\end{aligned}
$$

as desired.

*Claims 5 and 6:* We only show the results for $\frac{\partial^2}{\partial \theta_{j'k'} \partial \gamma_j} \mathrm{risk}_X[\boldsymbol{\gamma}, \Theta]$. The results for $\frac{\partial^2}{\partial \gamma_{j'} \partial \theta_{jk}} \mathrm{risk}_X[\boldsymbol{\gamma}, \Theta]$ can be obtained using the same arguments.

We consider two cases:

*Case 1:* if $j' = j$, we use 1. Claim 1, 2. the chain rule, and 3. taking the derivatives and simplifying to obtain

$$
\begin{aligned}
\frac{\partial^2}{\partial \theta_{jk'} \partial \gamma_j} \mathrm{risk}_X[\boldsymbol{\gamma}, \Theta] &= \frac{\partial}{\partial \theta_{jk'}} \left( -\frac{2}{n} \sum_{i=1}^{n} \left( (y_i - \boldsymbol{\gamma}^\top \Theta \boldsymbol{x}_i)(\Theta \boldsymbol{x}_i)_j \right) \right) \\
&= -\frac{2}{n} \sum_{i=1}^{n} \left( (\Theta \boldsymbol{x}_i)_j \frac{\partial}{\partial \theta_{jk'}} (y_i - \boldsymbol{\gamma}^\top \Theta \boldsymbol{x}_i) + (y_i - \boldsymbol{\gamma}^\top \Theta \boldsymbol{x}_i) \frac{\partial}{\partial \theta_{jk'}} (\Theta \boldsymbol{x}_i)_j \right) \\
&= \frac{2}{n} \sum_{i=1}^{n} \left( \gamma_j (\boldsymbol{x}_i)_{k'} (\Theta \boldsymbol{x}_i)_j - (y_i - \boldsymbol{\gamma}^\top \Theta \boldsymbol{x}_i)(\boldsymbol{x}_i)_{k'} \right).
\end{aligned}
$$

*Case 2:* if $j' \neq j$, we use 1. Claim 1, 2. the chain rule, and 3. taking the derivatives and rearranging to obtain

$$
\begin{aligned}
\frac{\partial^2}{\partial \theta_{j'k'} \partial \gamma_j} \mathrm{risk}_X[\boldsymbol{\gamma}, \Theta] &= \frac{\partial}{\partial \theta_{j'k'}} \left( -\frac{2}{n} \sum_{i=1}^n \Big( (y_i - \boldsymbol{\gamma}^\top \Theta \boldsymbol{x}_i)(\Theta \boldsymbol{x}_i)_j \Big) \right) \\
&= -\frac{2}{n} \sum_{i=1}^n \left( (\Theta \boldsymbol{x}_i)_j \frac{\partial}{\partial \theta_{j'k'}} (y_i - \boldsymbol{\gamma}^\top \Theta \boldsymbol{x}_i) + (y_i - \boldsymbol{\gamma}^\top \Theta \boldsymbol{x}_i) \frac{\partial}{\partial \theta_{j'k'}} (\Theta \boldsymbol{x}_i)_j \right) \\
&= \frac{2}{n} \gamma_{j'} \sum_{i=1}^n (\boldsymbol{x}_i)_{k'} (\Theta \boldsymbol{x}_i)_j \,,
\end{aligned}
$$

as desired. $\qquad\square$

### B.9 Proof of Lemma 6

*Proof.* The proof for this lemma follows the same steps as in Lemma 5, just sums are replaced by expectations and so we omit the proof. $\qquad\square$

## C Appendix: Proofs for shallow ReLU networks

### C.1 Proof of Theorem 3

*Proof.* The proof approach follows almost the same line as in Theorem 1.

We use the notation $\boldsymbol{\gamma}_{\boldsymbol{\alpha}}^\top \boldsymbol{\sigma}(\Theta_{\boldsymbol{\alpha}} \boldsymbol{x})$ to make a rescaled networks using a suitable $\boldsymbol{\alpha}$ (see more details about rescaled networks in Section 6.) Using the above definitions, it is easy to see that $\boldsymbol{\gamma}_{\boldsymbol{\alpha}}^\top \boldsymbol{\sigma}(\Theta_{\boldsymbol{\alpha}} \boldsymbol{x}) = \boldsymbol{\gamma}^\top \boldsymbol{\sigma}(\Theta \boldsymbol{x})$, that means, the output of the rescaled network is the same as the original network (using the definition of rescaled weights and Lipschitz property of ReLU networks with Lipschitz constant one).

Now, let's start the proof by writing a second-order Taylor expansion of $\mathrm{risk}[\boldsymbol{\gamma}^*{}_{\boldsymbol{\alpha}}, \Theta^*{}_{\boldsymbol{\alpha}}]$ (the risk in a rescaled version of the target with $\boldsymbol{\beta}^*{}_{\boldsymbol{\alpha}} = \mathrm{vec}(\boldsymbol{\gamma}^*{}_{\boldsymbol{\alpha}}, \Theta^*{}_{\boldsymbol{\alpha}}) \in \mathbb{R}^p$) around a rescaled version of a reasonable stationary $\widetilde{\boldsymbol{\beta}}_{\boldsymbol{\alpha}} = \mathrm{vec}(\widetilde{\boldsymbol{\gamma}}_{\boldsymbol{\alpha}}, \widetilde{\Theta}_{\boldsymbol{\alpha}}) \in \mathbb{R}^p$ with suitable $\boldsymbol{\alpha} \in \mathbb{R}^w$ to get

$$
\begin{aligned}
\mathrm{risk}[\boldsymbol{\gamma}^*{}_{\boldsymbol{\alpha}}, \Theta^*{}_{\boldsymbol{\alpha}}] = {}& \mathrm{risk}[\widetilde{\boldsymbol{\gamma}}_{\boldsymbol{\alpha}}, \widetilde{\Theta}_{\boldsymbol{\alpha}}] + \nabla \mathrm{risk}[\widetilde{\boldsymbol{\gamma}}_{\boldsymbol{\alpha}}, \widetilde{\Theta}_{\boldsymbol{\alpha}}]^\top (\boldsymbol{\beta}^*{}_{\boldsymbol{\alpha}} - \widetilde{\boldsymbol{\beta}}_{\boldsymbol{\alpha}}) \\
&+ \frac{1}{2} (\boldsymbol{\beta}^*{}_{\boldsymbol{\alpha}} - \widetilde{\boldsymbol{\beta}}_{\boldsymbol{\alpha}})^\top \nabla^2 \mathrm{risk}[\widetilde{\boldsymbol{\gamma}}_{\boldsymbol{\alpha}} + t(\boldsymbol{\gamma}^*{}_{\boldsymbol{\alpha}} - \widetilde{\boldsymbol{\gamma}}_{\boldsymbol{\alpha}}), \widetilde{\Theta}_{\boldsymbol{\alpha}} + t(\Theta^*{}_{\boldsymbol{\alpha}} - \widetilde{\Theta}_{\boldsymbol{\alpha}})](\boldsymbol{\beta}^*{}_{\boldsymbol{\alpha}} - \widetilde{\boldsymbol{\beta}}_{\boldsymbol{\alpha}})
\end{aligned}
$$

for some $t \in (0, 1)$ (Bertsekas et al., 2003, Proposition 1.1.13.a).

Then, we employ the property of rescaled networks that is $\mathrm{risk}[\widetilde{\boldsymbol{\gamma}}_{\boldsymbol{\alpha}}, \widetilde{\Theta}_{\boldsymbol{\alpha}}] = \mathrm{risk}[\widetilde{\boldsymbol{\gamma}}, \widetilde{\Theta}]$ and $\mathrm{risk}[\boldsymbol{\gamma}^*{}_{\boldsymbol{\alpha}}, \Theta^*{}_{\boldsymbol{\alpha}}] = \mathrm{risk}[\boldsymbol{\gamma}^*, \Theta^*]$, and use the shorthand notation

$$
m := (\boldsymbol{\beta}^*{}_{\boldsymbol{\alpha}} - \widetilde{\boldsymbol{\beta}}_{\boldsymbol{\alpha}})^\top \nabla^2 \mathrm{risk}[\widetilde{\boldsymbol{\gamma}}_{\boldsymbol{\alpha}} + t(\boldsymbol{\gamma}^*{}_{\boldsymbol{\alpha}} - \widetilde{\boldsymbol{\gamma}}_{\boldsymbol{\alpha}}), \widetilde{\Theta}_{\boldsymbol{\alpha}} + t(\Theta^*{}_{\boldsymbol{\alpha}} - \widetilde{\Theta}_{\boldsymbol{\alpha}})](\boldsymbol{\beta}^*{}_{\boldsymbol{\alpha}} - \widetilde{\boldsymbol{\beta}}_{\boldsymbol{\alpha}})
$$

to obtain

$$
\mathrm{risk}[\boldsymbol{\gamma}^*, \Theta^*] = \mathrm{risk}[\widetilde{\boldsymbol{\gamma}}, \widetilde{\Theta}] + \nabla \mathrm{risk}[\widetilde{\boldsymbol{\gamma}}_{\boldsymbol{\alpha}}, \widetilde{\Theta}_{\boldsymbol{\alpha}}]^\top (\boldsymbol{\beta}^*{}_{\boldsymbol{\alpha}} - \widetilde{\boldsymbol{\beta}}_{\boldsymbol{\alpha}}) + \frac{1}{2} m \,.
$$

It is also straightforward to show that $\nabla \mathrm{risk}[\widetilde{\boldsymbol{\gamma}}_{\boldsymbol{\alpha}}, \widetilde{\Theta}_{\boldsymbol{\alpha}}]^\top (\boldsymbol{\beta}^*{}_{\boldsymbol{\alpha}} - \widetilde{\boldsymbol{\beta}}_{\boldsymbol{\alpha}}) = \nabla \mathrm{risk}[\widetilde{\boldsymbol{\gamma}}, \widetilde{\Theta}]^\top (\boldsymbol{\beta}^* - \widetilde{\boldsymbol{\beta}})$ (we omit the detailed proof). Tabulating this observation in the earlier display we obtain

$$
\mathrm{risk}[\boldsymbol{\gamma}^*, \Theta^*] = \mathrm{risk}[\widetilde{\boldsymbol{\gamma}}, \widetilde{\Theta}] + \nabla \mathrm{risk}[\widetilde{\boldsymbol{\gamma}}, \widetilde{\Theta}]^\top (\boldsymbol{\beta}^* - \widetilde{\boldsymbol{\beta}}) + \frac{1}{2} m \,.
$$

Rearranging the display above we obtain

$$
-\nabla \mathrm{risk}[\widetilde{\boldsymbol{\gamma}}, \widetilde{\Theta}]^\top (\boldsymbol{\beta}^* - \widetilde{\boldsymbol{\beta}}) = \mathrm{risk}[\widetilde{\boldsymbol{\gamma}}, \widetilde{\Theta}] - \mathrm{risk}[\boldsymbol{\gamma}^*, \Theta^*] + \frac{1}{2} m \,.
$$

Now, let's recall the definition of stationary points in equation 3 that implies

$$\nabla \text{risk}_X[\widetilde{\gamma}, \widetilde{\Theta}]^\top (\beta^* - \widetilde{\beta}) + r\tilde{z}^\top (\beta^* - \widetilde{\beta}) \geq 0 \,.$$

We 1. rearrange the above inequality and expand the bracket, 2. use Hölder's inequality and the fact that $\tilde{z}^\top \widetilde{\beta} = \|\widetilde{\beta}\|_1$ (recall that $\tilde{z} \in \partial \|\widetilde{\beta}\|_1$), and 3. use $\|\tilde{z}\|_\infty \leq 1$ to obtain

$$
\begin{aligned}
-\nabla \text{risk}_X[\widetilde{\gamma}, \widetilde{\Theta}]^\top (\beta^* - \widetilde{\beta}) &\leq r\tilde{z}^\top \beta^* - r\tilde{z}^\top \widetilde{\beta} \\
&\leq r\|\tilde{z}\|_\infty \|\beta^*\|_1 - r\|\widetilde{\beta}\|_1 \\
&\leq r\|\beta^*\|_1 - r\|\widetilde{\beta}\|_1 \,,
\end{aligned}
$$

which rearranging implies

$$\nabla \text{risk}_X[\widetilde{\gamma}, \widetilde{\Theta}]^\top (\beta^* - \widetilde{\beta}) + r\|\beta^*\|_1 - r\|\widetilde{\beta}\|_1 \geq 0 \,.$$

Display above reveals the positiveness of the terms on its left-hand side and we can obtain

$$-\nabla \text{risk}[\widetilde{\gamma}, \widetilde{\Theta}]^\top (\beta^* - \widetilde{\beta}) \leq -\nabla \text{risk}[\widetilde{\gamma}, \widetilde{\Theta}]^\top (\beta^* - \widetilde{\beta}) + \nabla \text{risk}_X[\widetilde{\gamma}, \widetilde{\Theta}]^\top (\beta^* - \widetilde{\beta}) + r\|\beta^*\|_1 - r\|\widetilde{\beta}\|_1 \,,$$

that is,

$$-\nabla \text{risk}[\widetilde{\gamma}, \widetilde{\Theta}]^\top (\beta^* - \widetilde{\beta}) \leq \left( \nabla \text{risk}_X[\widetilde{\gamma}, \widetilde{\Theta}] - \nabla \text{risk}[\widetilde{\gamma}, \widetilde{\Theta}] \right)^\top (\beta^* - \widetilde{\beta}) + r\|\beta^*\|_1 - r\|\widetilde{\beta}\|_1 \,.$$

Now, let's use our display earlier (obtained by Taylor expansion) to rewrite the left-hand side of the display above as

$$\text{risk}[\widetilde{\gamma}, \widetilde{\Theta}] - \text{risk}[\gamma^*, \Theta^*] + \frac{1}{2}m \leq \left( \nabla \text{risk}_X[\widetilde{\gamma}, \widetilde{\Theta}] - \nabla \text{risk}[\widetilde{\gamma}, \widetilde{\Theta}] \right)^\top (\beta^* - \widetilde{\beta}) + r\|\beta^*\|_1 - r\|\widetilde{\beta}\|_1 \,.$$

Rearranging the display above we obtain

$$\text{risk}[\widetilde{\gamma}, \widetilde{\Theta}] \leq \text{risk}[\gamma^*, \Theta^*] + r\|\beta^*\|_1 + \left( \nabla \text{risk}_X[\widetilde{\gamma}, \widetilde{\Theta}] - \nabla \text{risk}[\widetilde{\gamma}, \widetilde{\Theta}] \right)^\top (\beta^* - \widetilde{\beta}) - r\|\widetilde{\beta}\|_1 - \frac{1}{2}m \,.$$

For the right-hand side of the inequality above we 1. get an absolute value of the third term, 2. add a zero-valued factor, 3. use triangle inequality, and 4. Remark 1 to obtain

$$
\begin{aligned}
\text{risk}[\widetilde{\gamma}, \widetilde{\Theta}] &\leq \text{risk}[\gamma^*, \Theta^*] + r\|\beta^*\|_1 + \left| \left( \nabla \text{risk}_X[\widetilde{\gamma}, \widetilde{\Theta}] - \nabla \text{risk}[\widetilde{\gamma}, \widetilde{\Theta}] \right)^\top (\beta^* - \widetilde{\beta}) \right| - r\|\widetilde{\beta}\|_1 - \frac{1}{2}m \\
&= \text{risk}[\gamma^*, \Theta^*] + 2r\|\beta^*\|_1 + \left| \left( \nabla \text{risk}_X[\widetilde{\gamma}, \widetilde{\Theta}] - \nabla \text{risk}[\widetilde{\gamma}, \widetilde{\Theta}] \right)^\top (\beta^* - \widetilde{\beta}) \right| - r\left( \|\widetilde{\beta}\|_1 + \|\beta^*\|_1 \right) \\
&\quad - \frac{1}{2}m \\
&\leq \text{risk}[\gamma^*, \Theta^*] + 2r\|\beta^*\|_1 + \left| \left( \nabla \text{risk}_X[\widetilde{\gamma}, \widetilde{\Theta}] - \nabla \text{risk}[\widetilde{\gamma}, \widetilde{\Theta}] \right)^\top (\beta^* - \widetilde{\beta}) \right| - r\|\beta^* - \widetilde{\beta}\|_1 - \frac{1}{2}m \\
&\leq \text{risk}[\gamma^*, \Theta^*] + 2r\|\beta^*\|_1 + r_{\text{orc}}\|\beta^* - \widetilde{\beta}\|_1 + \frac{r_{\text{orc}}}{2n} - r\|\beta^* - \widetilde{\beta}\|_1 - \frac{1}{2}m
\end{aligned}
$$

with probability at least $1 - 1/2n$.

The third and fifth terms in the last inequality above can be canceled if we choose the tuning parameter large enough. Hence, we obtain

$$\text{risk}[\widetilde{\gamma}, \widetilde{\Theta}] \leq \text{risk}[\gamma^*, \Theta^*] + 2r\|\beta^*\|_1 + \frac{r_{\text{orc}}}{2n} - \frac{1}{2}m$$

for $r \geq r_{\text{orc}}$ (see Remark 1).

The rest of the proof is analyzing the behavior of $m$. Let's rewrite $m = \|\beta^*_\alpha - \widetilde{\beta}_\alpha\|_2^2 \, m'$ with

$$m' := \frac{(\beta^*_\alpha - \widetilde{\beta}_\alpha)^\top}{\|\beta^*_\alpha - \widetilde{\beta}_\alpha\|_2} \nabla^2 \text{risk}[\widetilde{\gamma}_\alpha + t(\gamma^*_\alpha - \widetilde{\gamma}_\alpha), \widetilde{\Theta}_\alpha + t(\Theta^*_\alpha - \widetilde{\Theta}_\alpha)] \frac{(\beta^*_\alpha - \widetilde{\beta}_\alpha)}{\|\beta^*_\alpha - \widetilde{\beta}_\alpha\|_2} \,.$$

Now, we are motivated to employ our results in Proposition 2. To do so, we need to make sure about matrix $(\widetilde{\Theta} + t(\Theta^* - \widetilde{\Theta}))$ to verify our required condition, namely, active rows being approximately perpendicular. Employing our assumption that the stationary point $\widetilde{\Theta}$ and $\Theta^*$ have approximately perpendicular (active) rows and they have negligible cross-alignment (off-diagonal elements of $\widetilde{\Theta}\Theta^{*\top}$ and $\Theta^*\widetilde{\Theta}^\top$ are approximately zero), we can show that the line-segment between the two endpoints also verifies the assumption of Proposition 2 (active rows are approximately perpendicular) and so ensures the Hessian exhibits well behavior. To be more precise, $(\widetilde{\Theta} + t(\Theta^* - \widetilde{\Theta}))(\widetilde{\Theta} + t(\Theta^* - \widetilde{\Theta}))^\top = (1-t)^2\widetilde{\Theta}\widetilde{\Theta}^\top + t^2\Theta^*\Theta^{*\top} + t(1-t)(\widetilde{\Theta}\Theta^{*\top} + \Theta^*\widetilde{\Theta}^\top)$ will be approximately diagonal, assuming two end-points having approximately perpendicular rows and that off-diagonal elements of $\widetilde{\Theta}\Theta^{*\top}$ and $\Theta^*\widetilde{\Theta}^\top$ are approximately zero.

Implying Proposition 2 (with $\boldsymbol{a} = (\boldsymbol{\beta}^*_{\boldsymbol{\alpha}} - \widetilde{\boldsymbol{\beta}}_{\boldsymbol{\alpha}})/\|\boldsymbol{\beta}^*_{\boldsymbol{\alpha}} - \widetilde{\boldsymbol{\beta}}_{\boldsymbol{\alpha}}\|_2$) we obtain that $m' \in [0, \infty)$ for appropriate $\boldsymbol{\alpha}$, that is, $\boldsymbol{\alpha}$ with large enough $c$. The observation that $m' \in [0, \infty)$ together with the definition of $m$ implies that $m \in [0, \infty)$ as well.

Tabulating this observation to the display earlier together with our assumption on $\boldsymbol{\beta}^*$ ($\|\boldsymbol{\beta}^*\|_1 = \|\boldsymbol{\gamma}^*\|_1 + \|\Theta^*\|_1 \leq 2\sqrt{\log n}$) and the fact that $1/2n \leq \sqrt{\log n}$, we obtain for all $r \geq r_{\text{orc}}$ that

$$\text{risk}[\widetilde{\boldsymbol{\gamma}}, \widetilde{\Theta}] \leq \text{risk}[\boldsymbol{\gamma}^*, \Theta^*] + 2r\|\boldsymbol{\beta}^*\|_1 + \frac{r_{\text{orc}}}{2n} - \frac{1}{2}m$$

$$\lesssim \text{risk}[\boldsymbol{\gamma}^*, \Theta^*] + 2r\|\boldsymbol{\beta}^*\|_1 + \frac{r_{\text{orc}}}{2n}$$

$$\leq \text{risk}[\boldsymbol{\gamma}^*, \Theta^*] + 5r\sqrt{\log n}$$

with probability at least $1 - 1/2n$, which completes the proof. □

## C.2 Proof of Proposition 2

*Proof.* The proof is based on basic algebra and the property of scaling weights across the layers in neural networks. Without loss of generality, we assume that $\boldsymbol{x}_i \in \mathcal{N}(\boldsymbol{0}, I_{d\times d})$ (the proof for independent and centered sub-Gaussian random vectors $\boldsymbol{x}$ with independent coordinates is the same, just some constants may change, which doesn't affect the main results).

Let's consider all the network parameters as a vector of length $p$ (recall that $p = w + w \cdot d$). Then, we can tabulate the second order subdifferentials of $\text{risk}[\boldsymbol{\gamma}, \Theta]$ in a matrix called $\nabla^2\text{risk}[\boldsymbol{\gamma}, \Theta] \in \mathbb{R}^{p\times p}$ (for notational simplicity, we focus on $\nabla^2\text{risk}[\boldsymbol{\gamma}, \Theta]$ for the moment and then we move to $\nabla^2\text{risk}[\boldsymbol{\gamma}_{\boldsymbol{\alpha}}, \Theta_{\boldsymbol{\alpha}}]$ at the end of the proof) of the form

$$\nabla^2\text{risk}[\boldsymbol{\gamma}, \Theta] = \begin{bmatrix} A & C \\ B & D \end{bmatrix}$$

with $A \in \mathbb{R}^{w\times w}$, $B \in \mathbb{R}^{(w\cdot d)\times w}$, $C \in \mathbb{R}^{w\times(w\cdot d)}$, and $D \in \mathbb{R}^{(w\cdot d)\times(w\cdot d)}$, where

$$A_{j',j} := \frac{\partial^2}{\partial\gamma_{j'}\partial\gamma_j}\text{risk}[\boldsymbol{\gamma}, \Theta],$$

$$B_{(j'-1)d+k',j} := \frac{\partial^2}{\partial\theta_{j'k'}\partial\gamma_j}\text{risk}[\boldsymbol{\gamma}, \Theta],$$

$$C_{j',(j-1)d+k} := \frac{\partial^2}{\partial\gamma_{j'}\partial\theta_{jk}}\text{risk}[\boldsymbol{\gamma}, \Theta],$$

$$D_{(j'-1)d+k',(j-1)d+k} := \frac{\partial^2}{\partial\theta_{j'k'}\partial\theta_{jk}}\text{risk}[\boldsymbol{\gamma}, \Theta]$$

for $j, j' \in \{1, \ldots, w\}$ and $k, k' \in \{1, \ldots, d\}$.

Applying the block-wise structure of $\nabla^2\text{risk}[\boldsymbol{\gamma}, \Theta]$, we are motivated to analyze the behavior of

$$\boldsymbol{a}^\top\nabla^2\text{risk}[\boldsymbol{\gamma}, \Theta]\boldsymbol{a} = (\boldsymbol{a}^1)^\top A\boldsymbol{a}^1 + (\boldsymbol{a}^1)^\top C\boldsymbol{a}^2 + (\boldsymbol{a}^2)^\top B\boldsymbol{a}^1 + (\boldsymbol{a}^2)^\top D\boldsymbol{a}^2.$$

Note that $C = B^\top$ (by symmetry), so, we are left to analyze the behavior of

$$\boldsymbol{a}^\top\nabla^2\text{risk}[\boldsymbol{\gamma}, \Theta]\boldsymbol{a} = (\boldsymbol{a}^1)^\top A\boldsymbol{a}^1 + 2(\boldsymbol{a}^1)^\top C\boldsymbol{a}^2 + (\boldsymbol{a}^2)^\top D\boldsymbol{a}^2$$

for all $\boldsymbol{a} \in \mathbb{R}^p$ with $\|\boldsymbol{a}\|_2 = 1$.

We do the proof in steps: We start by going through the three terms on the right-hand side of display above separately, to write them in a mathematically nice formulation (Steps 1:3). In Step 4, we sum up the results computed in Steps 1:3 to prove the main claims of the proposition.

*Step 1:* On a high level, we prove that the entries of the matrix $D$ are a function of $\boldsymbol{\gamma}$.

Employing our results in Lemma 8, the symmetry over the input, and our assumption over $\Theta$ for $k = k'$ and $j \neq j'$, we obtain $\frac{\partial^2}{\partial \theta_{j'k'}\partial \theta_{jk}}\mathrm{risk}[\boldsymbol{\gamma}, \Theta] = \gamma_j \gamma_{j'}/2$, and for $k = k'$ and $j = j'$ we obtain $\frac{\partial^2}{\partial \theta_{jk}\partial \theta_{jk}}\mathrm{risk}[\boldsymbol{\gamma}, \Theta] = \gamma_j{}^2$. For other cases ($k \neq k'$) we use 1. our results in Lemma 8, 2. cauchy-schwarz inequality, and 3. our assumption on the input (symmetry) to obtain

$$
\frac{\partial^2}{\partial \theta_{j'k'}\partial \theta_{jk}}\mathrm{risk}[\boldsymbol{\gamma}, \Theta] = 2\gamma_j \gamma_{j'}\mathbb{E}_{\boldsymbol{x}}\big[(\boldsymbol{x})_{k'}(\boldsymbol{x})_k \mathbf{1}\{(\Theta\boldsymbol{x})_j > 0, (\Theta\boldsymbol{x})_{j'} > 0\}\big]
$$
$$
\leq 2|\gamma_j||\gamma_{j'}|\sqrt{\mathbb{E}_{\boldsymbol{x}}\big[((\boldsymbol{x})_k \mathbf{1}\{(\Theta\boldsymbol{x})_j > 0\})^2\big]\mathbb{E}_{\boldsymbol{x}}\big[((\boldsymbol{x})_{k'}\mathbf{1}\{(\Theta\boldsymbol{x})_{j'} > 0\})^2\big]}
$$
$$
\leq |\gamma_j||\gamma_{j'}|.
$$

*Step 2:* We prove that for $\boldsymbol{a}^1 \in \mathbb{R}^w$ and $A \in \mathbb{R}^{w \times w}$,

$$
(\boldsymbol{a}^1)^\top A \boldsymbol{a}^1 \approx \Big(1 - \frac{1}{\pi}\Big)\|\boldsymbol{a}^1\|_2^2 + \Big(\sum_{j=1}^w \frac{1}{\sqrt{\pi}}(\boldsymbol{a}^1{}_j)\Big)^2.
$$

For ReLU networks and according to Lemma 8, we have

$$
(\boldsymbol{a}^1)^\top A \boldsymbol{a}^1 = \sum_{j=1}^w \sum_{j'=1}^w \boldsymbol{a}^1{}_j A_{j'j} \boldsymbol{a}^1{}_{j'},
$$

in which $A_{jj'} = 2\mathbb{E}_{\boldsymbol{x}}[(\Theta\boldsymbol{x})_{j'}(\Theta\boldsymbol{x})_j \mathbf{1}\{(\Theta\boldsymbol{x})_{j'} > 0, (\Theta\boldsymbol{x})_j > 0\}]$. Employing some basic linear algebra implies

$$
(\boldsymbol{a}^1)^\top A \boldsymbol{a}^1 = \sum_{j=1}^w (\boldsymbol{a}^1{}_j)^2 A_{jj} + \sum_{j=1}^w \sum_{j'=1, j'\neq j}^w \boldsymbol{a}^1{}_j A_{j'j} \boldsymbol{a}^1{}_{j'}
$$
$$
= 2\sum_{j=1}^w (\boldsymbol{a}^1{}_j)^2 \mathbb{E}_{\boldsymbol{x}}\big[(\Theta\boldsymbol{x})_j(\Theta\boldsymbol{x})_j \mathbf{1}\{(\Theta\boldsymbol{x})_j > 0\}\big]
$$
$$
+ 2\sum_{j=1}^w \sum_{j'=1, j'\neq j}^w \boldsymbol{a}^1{}_j \mathbb{E}_{\boldsymbol{x}}\big[(\Theta\boldsymbol{x})_{j'}(\Theta\boldsymbol{x})_j \mathbf{1}\{(\Theta\boldsymbol{x})_{j'} > 0, (\Theta\boldsymbol{x})_j > 0\}\big]\boldsymbol{a}^1{}_{j'}
$$
$$
= 2\sum_{j=1}^w (\boldsymbol{a}^1{}_j)^2 \mathbb{E}_{\boldsymbol{x}}\big[\big((\Theta\boldsymbol{x})_j - \mathbb{E}_{\boldsymbol{x}}[(\Theta\boldsymbol{x})_j]\big)^2 \mathbf{1}\{(\Theta\boldsymbol{x})_j > 0\}\big]
$$
$$
+ 2\sum_{j=1}^w \sum_{j'=1, j'\neq j}^w \boldsymbol{a}^1{}_j \mathbb{E}_{\boldsymbol{x}}\big[(\Theta\boldsymbol{x})_{j'}(\Theta\boldsymbol{x})_j \mathbf{1}\{(\Theta\boldsymbol{x})_{j'} > 0, (\Theta\boldsymbol{x})_j > 0\}\big]\boldsymbol{a}^1{}_{j'}
$$
$$
= \sum_{j=1}^w (\boldsymbol{a}^1{}_j)^2 \mathbb{E}_{\boldsymbol{x}}\big[\big((\Theta\boldsymbol{x})_j - \mathbb{E}_{\boldsymbol{x}}[(\Theta\boldsymbol{x})_j]\big)^2\big]
$$
$$
+ 2\sum_{j=1}^w \sum_{j'=1, j'\neq j}^w \boldsymbol{a}^1{}_j \mathbb{E}_{\boldsymbol{x}}\big[(\Theta\boldsymbol{x})_{j'}(\Theta\boldsymbol{x})_j \mathbf{1}\{(\Theta\boldsymbol{x})_{j'} > 0, (\Theta\boldsymbol{x})_j > 0\}\big]\boldsymbol{a}^1{}_{j'}
$$
$$
= \sum_{j=1}^w (\boldsymbol{a}^1{}_j)^2 (\Theta\Theta^\top)_{jj} + 2\sum_{j=1}^w \sum_{j'=1, j'\neq j}^w \boldsymbol{a}^1{}_j \mathbb{E}_{\boldsymbol{x}}\big[(\Theta\boldsymbol{x})_{j'}(\Theta\boldsymbol{x})_j \mathbf{1}\{(\Theta\boldsymbol{x})_{j'} > 0, (\Theta\boldsymbol{x})_j > 0\}\big]\boldsymbol{a}^1{}_{j'}.
$$

We can prove that for cases with small $|\rho_{jj'}|$ (roughly about $|\rho_{jj'}| \le 0.2$), where $\rho_{jj'}$ is the correlation between the $(\Theta x)_j$ and $(\Theta x)_{j'}$ with Gaussian $x$, we can approximate

$$\mathbb{E}_{\boldsymbol{x}}\big[(\Theta \boldsymbol{x})_{j'}(\Theta \boldsymbol{x})_j \mathbf{1}\{(\Theta \boldsymbol{x})_{j'} > 0, (\Theta \boldsymbol{x})_j > 0\}\big] \approx \Big(\frac{1}{2\pi} + \frac{\rho_{jj'}}{4} - \frac{3\rho_{jj'}^2}{4\pi}\Big)\|\Theta_j\|\|\Theta_{j'}\|.$$

To be more specific, we can reach above result from scaling properties of Gaussian distributions and the homogeneity of the ReLU function together with Lemma 9

$$\mathbb{E}_{\boldsymbol{x}}[\boldsymbol{\sigma}(\Theta_j \boldsymbol{x})\boldsymbol{\sigma}(\Theta_{j'} \boldsymbol{x})] = \mathbb{E}_{\boldsymbol{x}}\Big[\|\Theta_j\|\boldsymbol{\sigma}\Big(\frac{\Theta_j}{\|\Theta_j\|}\boldsymbol{x}\Big)\|\Theta_{j'}\|\boldsymbol{\sigma}\Big(\frac{\Theta_{j'}}{\|\Theta_{j'}\|}\boldsymbol{x}\Big)\Big]$$
$$= \|\Theta_j\|\|\Theta_{j'}\|\mathbb{E}_{\boldsymbol{x}}\Big[\boldsymbol{\sigma}\Big(\frac{\Theta_j}{\|\Theta_j\|}\boldsymbol{x}\Big)\boldsymbol{\sigma}\Big(\frac{\Theta_{j'}}{\|\Theta_{j'}\|}\boldsymbol{x}\Big)\Big].$$

Then, we have

$$\sum_{j=1}^{w}(\boldsymbol{a}^1{}_j)^2\Big(\sum_{k=1}^{d}\theta_{jk}{}^2\Big) + 2\sum_{j=1}^{w}\sum_{j'=1,j'\neq j}^{w}\boldsymbol{a}^1{}_j\mathbb{E}_{\boldsymbol{x}}\big[(\Theta \boldsymbol{x})_{j'}(\Theta \boldsymbol{x})_j \mathbf{1}\{(\Theta \boldsymbol{x})_{j'} > 0, (\Theta \boldsymbol{x})_j > 0\}\big]\boldsymbol{a}^1{}_{j'}$$

$$\approx \sum_{j=1}^{w}(\boldsymbol{a}^1{}_j)^2\|\Theta_j\|^2 + 2\sum_{j=1}^{w}\sum_{j'=1,j'\neq j}^{w}\boldsymbol{a}^1{}_j\boldsymbol{a}^1{}_{j'}\|\Theta_j\|\|\Theta_{j'}\|\Big(\frac{1}{2\pi} + \frac{\rho_{jj'}}{4} - \frac{3\rho_{jj'}^2}{4\pi}\Big)$$

$$= \sum_{j=1}^{w}(\boldsymbol{a}^1{}_j)^2\|\Theta_j\|^2 - \frac{1}{\pi}\sum_{j=1}^{w}(\boldsymbol{a}^1{}_j)^2\|\Theta_j\|^2 + \frac{1}{\pi}\sum_{j=1}^{w}(\boldsymbol{a}^1{}_j)^2\|\Theta_j\|^2$$

$$+ \sum_{j=1}^{w}\sum_{j'=1,j'\neq j}^{w}\boldsymbol{a}^1{}_j\boldsymbol{a}^1{}_{j'}\|\Theta_j\|\|\Theta_{j'}\|\Big(\frac{1}{\pi} + \frac{\rho_{jj'}}{2} - \frac{3\rho_{jj'}^2}{2\pi}\Big)$$

$$= \sum_{j=1}^{w}(\boldsymbol{a}^1{}_j)^2\|\Theta_j\|^2\Big(1 - \frac{1}{\pi}\Big) + \Big(\sum_{j=1}^{w}\frac{1}{\sqrt{\pi}}(\boldsymbol{a}^1{}_j)\|\Theta_j\|\Big)^2$$

$$+ \sum_{j=1}^{w}\sum_{j'=1,j'\neq j}^{w}\boldsymbol{a}^1{}_j\boldsymbol{a}^1{}_{j'}\|\Theta_j\|\|\Theta_{j'}\|\Big(\frac{\rho_{jj'}}{2} - \frac{3\rho_{jj'}^2}{2\pi}\Big).$$

In the last equality above, the first two terms are our desired terms, while the last term still needs care. But we can argue that for small correlation values, we can ignore this term as it is a function of $\rho_{jj'}$ employing our assumption (rows are approximately perpendicular). Also note that for inactive rows, we are already good, since related factors will disappear from bounds.

*Step 3:* On a high level, we prove that the entries of the matrix $C$ are a function of the product over $\Theta$ and $\boldsymbol{\gamma}$.

Expanding $(\boldsymbol{a}^1)^\top C \boldsymbol{a}^2$ yields

$$(\boldsymbol{a}^1)^\top C \boldsymbol{a}^2 = \sum_{j=1}^{w}\sum_{k=1}^{d}\Big(\sum_{j'=1}^{w}\Big((\boldsymbol{a}^1)_{j'}\frac{\partial^2}{\partial\theta_{j'k'}\partial\gamma_j}\mathrm{risk}[\boldsymbol{\gamma},\Theta]\Big)(\boldsymbol{a}^2)_{(j-1)d+k}\Big).$$

Now, we need to consider two different cases:

*Case 1:* $(j \neq j')$

We use 1. Lemma 8, 2. rewriting the ReLU function, 3. rewriting the product in the form of sum, 4. linearity of expectations, 5. again linearity of expectation and rewriting, 6. using the assumption over the input, and 7. the same argument as above to obtain,

$$\frac{\partial^2}{\partial\theta_{j'k'}\partial\gamma_j}\mathrm{risk}[\boldsymbol{\gamma},\Theta] = 2\gamma_{j'}\mathbb{E}_{\boldsymbol{x}}[(\boldsymbol{x})_{k'}\boldsymbol{\sigma}(\Theta \boldsymbol{x})_j\kappa(\boldsymbol{x},j')]$$

$$= 2\gamma_{j'} \mathbb{E}_{\boldsymbol{x}}[(\boldsymbol{x})_{k'}(\Theta\boldsymbol{x})_j \mathbf{1}\{(\Theta\boldsymbol{x})_{j'} > 0\}\mathbf{1}\{(\Theta\boldsymbol{x})_j > 0\}]$$

$$= 2\gamma_{j'} \mathbb{E}_{\boldsymbol{x}}\left[(\boldsymbol{x})_{k'}\Big(\sum_{k=1}^{d}(\theta_{jk}\boldsymbol{x}_k)\Big)\mathbf{1}\{(\Theta\boldsymbol{x})_{j'} > 0\}\mathbf{1}\{(\Theta\boldsymbol{x})_j > 0\}\right]$$

$$= 2\gamma_{j'} \sum_{k=1}^{d} \mathbb{E}_{\boldsymbol{x}}\left[(\boldsymbol{x})_{k'}(\theta_{jk}\boldsymbol{x}_k)\mathbf{1}\{(\Theta\boldsymbol{x})_{j'} > 0\}\mathbf{1}\{(\Theta\boldsymbol{x})_j > 0\}\right]$$

$$= 2\gamma_{j'}\theta_{jk'} \mathbb{E}_{\boldsymbol{x}}\left[(\boldsymbol{x}_{k'})^2\mathbf{1}\{(\Theta\boldsymbol{x})_{j'} > 0\}\mathbf{1}\{(\Theta\boldsymbol{x})_j > 0\}\right]$$

$$+ 2\gamma_{j'} \sum_{k=1,k\neq k'}^{d} \theta_{jk}\mathbb{E}_{\boldsymbol{x}}\left[\boldsymbol{x}_{k'}\boldsymbol{x}_k\mathbf{1}\{(\Theta\boldsymbol{x})_{j'} > 0\}\mathbf{1}\{(\Theta\boldsymbol{x})_j > 0\}\right]$$

$$= \frac{1}{2}\gamma_{j'}\theta_{jk'} + 2\gamma_{j'}\Big(\sum_{k=1,k\neq k'}^{d} \theta_{jk}\mathbb{E}_{\boldsymbol{x}}\left[\boldsymbol{x}_{k'}\mathbf{1}\{(\Theta\boldsymbol{x})_{j'} > 0\}\mathbf{1}\{(\Theta\boldsymbol{x})_j > 0\}\right]$$

$$\mathbb{E}_{\boldsymbol{x}}\left[\boldsymbol{x}_k\mathbf{1}\{(\Theta\boldsymbol{x})_{j'} > 0\}\mathbf{1}\{(\Theta\boldsymbol{x})_j > 0\}\right]\Big)$$

$$= \frac{1}{2}\gamma_{j'}\theta_{jk'} + \frac{1}{4\pi}\gamma_{j'}\sum_{k=1,k\neq k'}^{d} \theta_{jk}\,.$$

*Case 2:* $(j = j')$

We use 1. the result of Lemma 8, 2. linearity of expectations, almost the same proof as above for simplifying the first term, replacing $\boldsymbol{y}$ with its definition, and the assumption over noise to obtain

$$\frac{\partial^2}{\partial\theta_{jk'}\partial\gamma_j}\text{risk}[\boldsymbol{\gamma}, \Theta] = 2\mathbb{E}_{\boldsymbol{x},y}\Big[\gamma_j(\boldsymbol{x})_{k'}\boldsymbol{\sigma}(\Theta\boldsymbol{x})_j\kappa(\boldsymbol{x},j) - (y - \boldsymbol{\gamma}^\top\boldsymbol{\sigma}(\Theta\boldsymbol{x}))(\boldsymbol{x})_{k'}\kappa(\boldsymbol{x},j)\Big]$$

$$= \gamma_{j'}\theta_{jk'} + \frac{1}{2\pi}\gamma_{j'}\sum_{k=1,k\neq k'}^{d}\theta_{jk}$$

$$+ 2\mathbb{E}_{\boldsymbol{x},y}\Big[\big(\boldsymbol{\gamma}^\top\boldsymbol{\sigma}(\Theta\boldsymbol{x}) - \boldsymbol{\gamma}^{*\top}\boldsymbol{\sigma}(\Theta^*\boldsymbol{x})\big)(\boldsymbol{x})_{k'}\mathbf{1}\{(\Theta\boldsymbol{x})_j) > 0\}\Big]\,.$$

Then, we use the linearity of expectations to obtain

$$\mathbb{E}_{\boldsymbol{x}}\Big[\big(\boldsymbol{\gamma}^\top\boldsymbol{\sigma}(\Theta\boldsymbol{x}) - \boldsymbol{\gamma}^{*\top}\boldsymbol{\sigma}(\Theta^*\boldsymbol{x})\big)(\boldsymbol{x})_{k'}\mathbf{1}\{(\Theta\boldsymbol{x})_j) > 0\}\Big]$$

$$= \mathbb{E}_{\boldsymbol{x}}\Big[\big(\boldsymbol{\gamma}^\top\boldsymbol{\sigma}(\Theta\boldsymbol{x})\big)(\boldsymbol{x})_{k'}\mathbf{1}\{(\Theta\boldsymbol{x})_j) > 0\}\Big] - \mathbb{E}_{\boldsymbol{x}}\Big[\big(\boldsymbol{\gamma}^{*\top}\boldsymbol{\sigma}(\Theta^*\boldsymbol{x})\big)(\boldsymbol{x})_{k'}\mathbf{1}\{(\Theta\boldsymbol{x})_j) > 0\}\Big]$$

and

$$\mathbb{E}_{\boldsymbol{x}}\Big[\big(\boldsymbol{\gamma}^\top\boldsymbol{\sigma}(\Theta\boldsymbol{x})\big)(\boldsymbol{x})_{k'}\mathbf{1}\{(\Theta\boldsymbol{x})_j) > 0\}\Big] = \sum_{j'=1}^{w} \mathbb{E}_{\boldsymbol{x}}\Big[\big(\gamma_{j'}\big(\boldsymbol{\sigma}(\Theta\boldsymbol{x})\big)_{j'}(\boldsymbol{x})_{k'}\mathbf{1}\{(\Theta\boldsymbol{x})_j) > 0\}\Big]$$

$$= \sum_{j'=1}^{w}\Big(\gamma_{j'}\theta_{jk'} + \frac{1}{2\pi}\sum_{k=1,k\neq k'}^{d}\gamma_{j'}\theta_{j'k}\Big).$$

The same argument can also hold for the other term. Looking at the extracted entries of the matrix $C$ above, it is clear that the entries are a function of the product over parameters of the first and second layers.

*Step 4* Collecting the results from Steps 1–3, we can easily approve the first claim. For the second claim, we realize that by employing the same scaling trick as in the linear case, that is considering parameters of the first layer large enough (by selecting $\boldsymbol{\theta}$ large enough) and dividing $\boldsymbol{\gamma}$ by the same value, the result from Step 2 (that the squared of the scaling parameter $\boldsymbol{\theta}$ will appear in the front) can dominate all the other terms. According to Step 1, the entries of the matrix $D$ are a function of $\boldsymbol{\gamma}$ and also according to Step 3, matrix $C$ involves a product of first and last layer parameters, which in this case cancel out the scaling parameter and

so, the result from Step 2 can dominate all other parts, as long as $\boldsymbol{\theta}$ is selected large enough. To be more precise we have

$$
\begin{aligned}
\boldsymbol{a}^\top \nabla^2 \mathrm{risk}[\boldsymbol{\gamma_\alpha}, \Theta_{\boldsymbol{\alpha}}]\boldsymbol{a} &= (\boldsymbol{a}^1)^\top A \boldsymbol{a}^1 + 2(\boldsymbol{a}^1)^\top C \boldsymbol{a}^2 + (\boldsymbol{a}^2)^\top D \boldsymbol{a}^2 \\
&\gtrapprox (\boldsymbol{a}^1)^\top A \boldsymbol{a}^1 + (\boldsymbol{a}^2)^\top D \boldsymbol{a}^2 - 2\|\boldsymbol{a}^1\|_2 \|C\|_2 \|\boldsymbol{a}^2\|_2 \\
&\geq (\boldsymbol{a}^1)^\top A \boldsymbol{a}^1 - \|\boldsymbol{a}^2\|_2^2 \|D\|_2 - 2\|C\|_2 \\
&\geq \left(1 - \frac{1}{\pi}\right)\|\boldsymbol{a}^1\|_2^2 \theta^2 + \left(\sum_{j=1}^w \frac{1}{\sqrt{\pi}}(\boldsymbol{a}^1{}_j)\boldsymbol{\theta}\right)^2 - \|D\|_2 - 2\|C\|_2
\end{aligned}
$$

for all $\boldsymbol{a} \in \mathbb{R}^p$ with $\|\boldsymbol{a}\|_2 = 1$. For large enough $\boldsymbol{\theta}$, the first term in the last inequality above can dominate the last two terms, which involve the product of parameters that cancel out the scaling constant or they are just dependent over $\boldsymbol{\gamma}$. For the special case of $\boldsymbol{a}^1 = \boldsymbol{0}$, if we consider a large enough $\boldsymbol{\theta}$, the entries of the matrix $D$ can go to zero (so implying its norm $\|D\|_2$ going to zero) and so we can reach our desired results. $\qquad\square$

### C.3    Proof of Lemma 7

*Proof.* The proof consists of basic linear algebra.

*Claim 1:* We use 1. the definition of $\mathrm{risk}_X[\boldsymbol{\gamma}, \Theta]$, 2. the chain rule, and 3. differentiating to obtain

$$
\begin{aligned}
\frac{\partial}{\partial \gamma_j}\mathrm{risk}_X[\boldsymbol{\gamma}, \Theta] &= \frac{\partial}{\partial \gamma_j}\left(\frac{1}{n}\sum_{i=1}^n (y_i - \boldsymbol{\gamma}^\top \boldsymbol{\sigma}(\Theta \boldsymbol{x}_i))^2\right) \\
&= -\frac{2}{n}\sum_{i=1}^n \left((y_i - \boldsymbol{\gamma}^\top \boldsymbol{\sigma}(\Theta \boldsymbol{x}_i))\frac{\partial}{\partial \gamma_j}(\boldsymbol{\gamma}^\top \boldsymbol{\sigma}(\Theta \boldsymbol{x}_i))\right) \\
&= -\frac{2}{n}\sum_{i=1}^n \left((y_i - \boldsymbol{\gamma}^\top \boldsymbol{\sigma}(\Theta \boldsymbol{x}_i))\boldsymbol{\sigma}(\Theta \boldsymbol{x}_i)_j\right),
\end{aligned}
$$

as desired.

*Claim 2:* We 1. use Claim 1, and 2. remove the term with zero derivative and use the chain rule to obtain

$$
\begin{aligned}
\frac{\partial^2}{\partial \gamma_{j'}\partial \gamma_j}\mathrm{risk}_X[\boldsymbol{\gamma}, \Theta] &= \frac{\partial}{\partial \gamma_{j'}}\left(-\frac{2}{n}\sum_{i=1}^n \left((y_i - \boldsymbol{\gamma}^\top \boldsymbol{\sigma}(\Theta \boldsymbol{x}_i))\boldsymbol{\sigma}(\Theta \boldsymbol{x}_i)_j\right)\right) \\
&= \frac{2}{n}\sum_{i=1}^n \left((\boldsymbol{\sigma}(\Theta \boldsymbol{x}_i)_{j'}\boldsymbol{\sigma}(\Theta \boldsymbol{x}_i)_j)\right),
\end{aligned}
$$

as desired.

*Claim 3:* We use 1. the definition of $\mathrm{risk}_X[\boldsymbol{\gamma}, \Theta]$, 2. the chain rule, and 3. differentiating to obtain

$$
\begin{aligned}
\frac{\partial}{\partial \theta_{jk}}\mathrm{risk}_X[\boldsymbol{\gamma}, \Theta] &= \frac{\partial}{\partial \theta_{jk}}\left(\frac{1}{n}\sum_{i=1}^n (y_i - \boldsymbol{\gamma}^\top \boldsymbol{\sigma}(\Theta \boldsymbol{x}_i))^2\right) \\
&= -\frac{2}{n}\sum_{i=1}^n \left((y_i - \boldsymbol{\gamma}^\top \boldsymbol{\sigma}(\Theta \boldsymbol{x}_i))\frac{\partial}{\partial \theta_{jk}}(\boldsymbol{\gamma}^\top \boldsymbol{\sigma}(\Theta \boldsymbol{x}_i))\right) \\
&= -\frac{2}{n}\sum_{i=1}^n \left((y_i - \boldsymbol{\gamma}^\top \boldsymbol{\sigma}(\Theta \boldsymbol{x}_i))\gamma_j(\boldsymbol{x}_i)_k \kappa(\boldsymbol{x}_i, j)\right).
\end{aligned}
$$

*Claim 4:* We 1. use Claim 3 and 2. differentiate the bracket to obtain for

$$\frac{\partial^2}{\partial\theta_{j'k'}\partial\theta_{jk}}\mathrm{risk}_X[\boldsymbol{\gamma},\Theta] = \frac{\partial}{\partial\theta_{j'k'}}\left(-\frac{2}{n}\sum_{i=1}^{n}\left((y_i - \boldsymbol{\gamma}^\top\boldsymbol{\sigma}(\Theta\boldsymbol{x}_i))\gamma_j(\boldsymbol{x}_i)_k\kappa(\boldsymbol{x}_i,j)\right)\right)$$

$$= \frac{\partial}{\partial\theta_{j'k'}}\left(\frac{2}{n}\gamma_j\sum_{i=1}^{n}\left((\boldsymbol{\gamma}^\top\boldsymbol{\sigma}(\Theta\boldsymbol{x}_i))(\boldsymbol{x}_i)_k\kappa(\boldsymbol{x}_i,j)\right)\right)$$

$$- \frac{\partial}{\partial\theta_{j'k'}}\left(\frac{2}{n}\gamma_j\sum_{i=1}^{n}\left(y_i(\boldsymbol{x}_i)_k\kappa(\boldsymbol{x}_i,j)\right)\right).$$

We obtain then for $j' \neq j$ that

$$\frac{\partial^2}{\partial\theta_{j'k'}\partial\theta_{jk}}\mathrm{risk}_X[\boldsymbol{\gamma},\Theta] = \frac{\partial}{\partial\theta_{j'k'}}\left(\frac{2}{n}\gamma_j\sum_{i=1}^{n}\left((\boldsymbol{\gamma}^\top\boldsymbol{\sigma}(\Theta\boldsymbol{x}_i))(\boldsymbol{x}_i)_k\kappa(\boldsymbol{x}_i,j)\right)\right)$$

$$= \frac{2}{n}\gamma_j\gamma_{j'}\left(\sum_{i=1}^{n}(\boldsymbol{x}_i)_{k'}(\boldsymbol{x}_i)_k\kappa(\boldsymbol{x}_i,j')\kappa(\boldsymbol{x}_i,j)\right)$$

and for $j' = j$ with $(\Theta\boldsymbol{x}_i)_j \neq 0$ for all $i \in \{1,\dots,n\}$

$$\frac{\partial^2}{\partial\theta_{j'k'}\partial\theta_{jk}}\mathrm{risk}_X[\boldsymbol{\gamma},\Theta] = \frac{\partial}{\partial\theta_{j'k'}}\left(\frac{2}{n}\gamma_j\sum_{i=1}^{n}\left((\boldsymbol{\gamma}^\top\boldsymbol{\sigma}(\Theta\boldsymbol{x}_i))(\boldsymbol{x}_i)_k\kappa(\boldsymbol{x}_i,j)\right)\right)$$

$$- \frac{\partial}{\partial\theta_{j'k'}}\left(\frac{2}{n}\gamma_j\sum_{i=1}^{n}\left(y_i(\boldsymbol{x}_i)_k\kappa(\boldsymbol{x}_i,j)\right)\right)$$

$$= \frac{2}{n}\gamma_j\gamma_{j'}\sum_{i=1}^{n}(\boldsymbol{x}_i)_{k'}(\boldsymbol{x}_i)_k\kappa(\boldsymbol{x}_i,j)\kappa(\boldsymbol{x}_i,j)\,,$$

otherwise, the corresponding subdifferential doesn't exist, as desired.

*Claims 5 and 6:* We only show the results for $\frac{\partial^2}{\partial\theta_{j'k'}\partial\gamma_j}\mathrm{risk}_X[\boldsymbol{\gamma},\Theta]$. The result for $\frac{\partial^2}{\partial\gamma_{j'}\partial\theta_{jk}}\mathrm{risk}_X[\boldsymbol{\gamma},\Theta]$ can be obtained using the same arguments.

We consider two cases:

*Case 1:* for $j' = j$ we use 1. Claim 1, 2. the chain rule, and 3. differentiating and simplifying to obtain

$$\frac{\partial^2}{\partial\theta_{jk'}\partial\gamma_j}\mathrm{risk}_X[\boldsymbol{\gamma},\Theta] = \frac{\partial}{\partial\theta_{jk'}}\left(-\frac{2}{n}\sum_{i=1}^{n}\left((y_i - \boldsymbol{\gamma}^\top\boldsymbol{\sigma}(\Theta\boldsymbol{x}_i))\boldsymbol{\sigma}(\Theta\boldsymbol{x}_i)_j\right)\right)$$

$$= -\frac{2}{n}\sum_{i=1}^{n}\left(\boldsymbol{\sigma}(\Theta\boldsymbol{x}_i)_j\frac{\partial}{\partial\theta_{jk'}}(y_i - \boldsymbol{\gamma}^\top\boldsymbol{\sigma}(\Theta\boldsymbol{x}_i)) + (y_i - \boldsymbol{\gamma}^\top\boldsymbol{\sigma}(\Theta\boldsymbol{x}_i))\frac{\partial}{\partial\theta_{jk'}}\boldsymbol{\sigma}(\Theta\boldsymbol{x}_i)_j\right)$$

$$= \frac{2}{n}\sum_{i=1}^{n}\left(\gamma_j(\boldsymbol{x}_i)_{k'}\boldsymbol{\sigma}(\Theta\boldsymbol{x}_i)_j\kappa(\boldsymbol{x}_i,j) - (y_i - \boldsymbol{\gamma}^\top\boldsymbol{\sigma}(\Theta\boldsymbol{x}_i))(\boldsymbol{x}_i)_{k'}\kappa(\boldsymbol{x}_i,j)\right).$$

*Case 2:* For $j' \neq j$ we use 1. Claim 1, 2. the chain rule, and 3. differentiating to obtain

$$\frac{\partial^2}{\partial\theta_{j'k'}\partial\gamma_j}\mathrm{risk}_X[\boldsymbol{\gamma},\Theta] = \frac{\partial}{\partial\theta_{j'k'}}\left(-\frac{2}{n}\sum_{i=1}^{n}\left((y_i - \boldsymbol{\gamma}^\top\boldsymbol{\sigma}(\Theta\boldsymbol{x}_i))\boldsymbol{\sigma}(\Theta\boldsymbol{x}_i)_j\right)\right)$$

$$= -\frac{2}{n}\sum_{i=1}^{n}\boldsymbol{\sigma}(\Theta\boldsymbol{x}_i)_j\frac{\partial}{\partial\theta_{j'k'}}(y_i - \boldsymbol{\gamma}^\top\boldsymbol{\sigma}(\Theta\boldsymbol{x}_i))$$

$$= \frac{2}{n}\gamma_{j'}\sum_{i=1}^{n}(\boldsymbol{x}_i)_{k'}\boldsymbol{\sigma}(\Theta\boldsymbol{x}_i)_j\kappa(\boldsymbol{x}_i,j').$$

A similar approach can give us

$$\frac{\partial^2}{\partial\gamma_{j'}\partial\theta_{jk}}\mathrm{risk}_X[\boldsymbol{\gamma},\Theta] = \frac{\partial}{\partial\gamma_{j'}}\left(-\frac{2}{n}\sum_{i=1}^{n}\Big((y_i - \boldsymbol{\gamma}^\top\boldsymbol{\sigma}(\Theta\boldsymbol{x}_i))\gamma_j(\boldsymbol{x}_i)_k\kappa(\boldsymbol{x}_i,j)\Big)\right).$$

For $j = j'$ we obtain

$$\frac{\partial^2}{\partial\gamma_{j'}\partial\theta_{jk}}\mathrm{risk}_X[\boldsymbol{\gamma},\Theta] = \left(-\frac{2}{n}\sum_{i=1}^{n}\Big((y_i)(\boldsymbol{x}_i)_k\kappa(\boldsymbol{x}_i,j)\Big)\right)$$
$$+ \left(\frac{2}{n}\sum_{i=1}^{n}\Big((\boldsymbol{\sigma}(\Theta\boldsymbol{x}_i)_j\gamma_j + \boldsymbol{\gamma}^\top\boldsymbol{\sigma}(\Theta\boldsymbol{x}_i))(\boldsymbol{x}_i)_k\kappa(\boldsymbol{x}_i,j)\Big)\right).$$

And for $j \neq j'$ we have

$$\frac{\partial^2}{\partial\gamma_{j'}\partial\theta_{jk}}\mathrm{risk}_X[\boldsymbol{\gamma},\Theta] = \frac{\partial}{\partial\gamma_{j'}}\left(-\frac{2}{n}\sum_{i=1}^{n}\Big((y_i - \boldsymbol{\gamma}^\top\boldsymbol{\sigma}(\Theta\boldsymbol{x}_i))\gamma_j(\boldsymbol{x}_i)_k\kappa(\boldsymbol{x}_i,j)\Big)\right)$$
$$= \frac{2}{n}\sum_{i=1}^{n}\boldsymbol{\sigma}(\Theta\boldsymbol{x}_i)_{j'}\gamma_j(\boldsymbol{x}_i)_k\kappa(\boldsymbol{x}_i,j),$$

as desired.

$\square$

## C.4 Proof of Remark 1

*Proof.* The proof can be followed almost in the same line as in Lemma 2 and Lemma 1; so we just provide a high-level proof here. The only difference with linear case is how to treat the ReLU function in subdifferentials. To do so, we study here the behavior of the absolute difference between the subdifferentials of the in-sample risk and population risk for ReLU networks, showing that they almost behave the same as linear networks despite minor changes in the constants and some log terms. First, we use the definition and employ some linear algebra to obtain

$$\left|\frac{\partial}{\partial\theta_{jk}}\mathrm{risk}_X[\boldsymbol{\gamma},\Theta] - \frac{\partial}{\partial\theta_{jk}}\mathrm{risk}[\boldsymbol{\gamma},\Theta]\right|$$
$$= \left|-\frac{2}{n}\sum_{i=1}^{n}(y_i - \boldsymbol{\gamma}^\top\boldsymbol{\sigma}(\Theta\boldsymbol{x}_i))\gamma_j(\boldsymbol{x}_i)_k\kappa(\boldsymbol{x}_i,j) + \mathbb{E}\left[\frac{2}{n}\sum_{i=1}^{n}(y_i - \boldsymbol{\gamma}^\top\boldsymbol{\sigma}(\Theta\boldsymbol{x}_i))\gamma_j(\boldsymbol{x}_i)_k\kappa(\boldsymbol{x}_i,j)\right]\right|$$
$$\leq 2|\gamma_j|\left|\frac{1}{n}\sum_{i=1}^{n}(u_i + \boldsymbol{\gamma}^{*\top}\boldsymbol{\sigma}(\Theta^*\boldsymbol{x}_i) - \boldsymbol{\gamma}^\top\boldsymbol{\sigma}(\Theta\boldsymbol{x}_i))(\boldsymbol{x}_i)_k\kappa(\boldsymbol{x}_i,j)\right.$$
$$\left. - \mathbb{E}\big[(\boldsymbol{\gamma}^{*\top}\boldsymbol{\sigma}(\Theta^*\boldsymbol{x}_i) - \boldsymbol{\gamma}^\top\boldsymbol{\sigma}(\Theta\boldsymbol{x}_i))(\boldsymbol{x}_i)_k\kappa(\boldsymbol{x}_i,j)\big]\right|$$
$$\leq 2\|\boldsymbol{\gamma}\|_\infty\left(\left|\frac{1}{n}\sum_{i=1}^{n}u_i(\boldsymbol{x}_i)_k\right| + \left|\frac{1}{n}\sum_{i=1}^{n}(\boldsymbol{\gamma}^{*\top}\boldsymbol{\sigma}(\Theta^*\boldsymbol{x}_i))(\boldsymbol{x}_i)_k\kappa(\boldsymbol{x}_i,j) - \mathbb{E}\big[(\boldsymbol{\gamma}^{*\top}\boldsymbol{\sigma}(\Theta^*\boldsymbol{x}_i))(\boldsymbol{x}_i)_k\kappa(\boldsymbol{x}_i,j)\big]\right|\right.$$
$$\left. + \left|\frac{1}{n}\sum_{i=1}^{n}(\boldsymbol{\gamma}^\top\boldsymbol{\sigma}(\Theta\boldsymbol{x}_i))(\boldsymbol{x}_i)_k\kappa(\boldsymbol{x}_i,j) - \mathbb{E}\big[(\boldsymbol{\gamma}^\top\boldsymbol{\sigma}(\Theta\boldsymbol{x}_i))(\boldsymbol{x}_i)_k\kappa(\boldsymbol{x}_i,j)\big]\right|\right).$$

The first term in the last inequality above was already treated in Lemma 2. So, we continue with the second term. We use 1. Hölder's inequality, 2. symmetrization (Bühlmann & Van De Geer, 2011, Theorem 14.3) with $\zeta_i$ as Rademacher random variables, and 3. an extension of contraction principle to obtain

$$\left|\frac{1}{n}\sum_{i=1}^{n}(\boldsymbol{\gamma}^{*\top}\boldsymbol{\sigma}(\Theta^*\boldsymbol{x}_i))(\boldsymbol{x}_i)_k\kappa(\boldsymbol{x}_i,j) - \mathbb{E}\big[(\boldsymbol{\gamma}^{*\top}\boldsymbol{\sigma}(\Theta^*\boldsymbol{x}_i))(\boldsymbol{x}_i)_k\kappa(\boldsymbol{x}_i,j)\big]\right|$$

$$\leq \|\boldsymbol{\gamma}^*\|_1 \Big\|\frac{1}{n}\sum_{i=1}^{n}\big(\boldsymbol{\sigma}(\Theta^*\boldsymbol{x}_i)(\boldsymbol{x}_i)_k\kappa(\boldsymbol{x}_i,j) - \mathbb{E}\big[\boldsymbol{\sigma}(\Theta^*\boldsymbol{x}_i)(\boldsymbol{x}_i)_k\kappa(\boldsymbol{x}_i,j)\big]\big)\Big\|_\infty$$

$$\leq 2\|\boldsymbol{\gamma}^*\|_1 \Big\|\frac{1}{n}\sum_{i=1}^{n}\big(\boldsymbol{\sigma}(\Theta^*\boldsymbol{x}_i)(\boldsymbol{x}_i)_k\kappa(\boldsymbol{x}_i,j)\zeta_i\big)\Big\|_\infty$$

$$\leq 4\|\boldsymbol{\gamma}^*\|_1 \Big\|\frac{1}{n}\sum_{i=1}^{n}\big(\boldsymbol{\sigma}(\Theta^*\boldsymbol{x}_i)(\boldsymbol{x}_i)_k\zeta_i\big)\Big\|_\infty.$$

Then we consider $\boldsymbol{z}_i = \boldsymbol{\sigma}(\Theta^*\boldsymbol{x}_i)(\boldsymbol{x}_i)_k\zeta_i$ as independent and mean-zero sub-exponential random vectors and the proof can be followed same line by the proof of Lemma 2. Also for $|\partial\mathrm{risk}_X[\boldsymbol{\gamma},\Theta]/\partial\gamma_j - \partial\mathrm{risk}[\boldsymbol{\gamma},\Theta]/\partial\gamma_j|$ we obtain

$$\left|\frac{\partial}{\partial\gamma_j}\mathrm{risk}_X[\boldsymbol{\gamma},\Theta] - \frac{\partial}{\partial\gamma_j}\mathrm{risk}[\boldsymbol{\gamma},\Theta]\right|$$

$$= \left|\frac{2}{n}\sum_{i=1}^{n}\Big(\big(y_i - \boldsymbol{\gamma}^\top\boldsymbol{\sigma}(\Theta\boldsymbol{x}_i)\big)\boldsymbol{\sigma}(\Theta\boldsymbol{x}_i)_j - \mathbb{E}\big[\big(y_i - \boldsymbol{\gamma}^\top\boldsymbol{\sigma}(\Theta\boldsymbol{x}_i)\big)\boldsymbol{\sigma}(\Theta\boldsymbol{x}_i)_j\big]\Big)\right|$$

$$= \left|\frac{2}{n}\sum_{i=1}^{n}\Big(\big(u_i + \boldsymbol{\gamma}^{*\top}\boldsymbol{\sigma}(\Theta^*\boldsymbol{x}_i) - \boldsymbol{\gamma}^\top\boldsymbol{\sigma}(\Theta\boldsymbol{x}_i)\big)\boldsymbol{\sigma}(\Theta\boldsymbol{x}_i)_j - \mathbb{E}\big[\big(\boldsymbol{\gamma}^{*\top}\boldsymbol{\sigma}(\Theta^*\boldsymbol{x}_i) - \boldsymbol{\gamma}^\top\boldsymbol{\sigma}(\Theta\boldsymbol{x}_i)\big)\boldsymbol{\sigma}(\Theta\boldsymbol{x}_i)_j\big]\Big)\right|$$

$$\leq \left|\frac{1}{n}\sum_{i=1}^{n}u_i(\Theta\boldsymbol{x}_i)_j\right| + \left|\frac{2}{n}\sum_{i=1}^{n}\Big(\big(\boldsymbol{\gamma}^{*\top}\boldsymbol{\sigma}(\Theta^*\boldsymbol{x}_i) - \boldsymbol{\gamma}^\top\boldsymbol{\sigma}(\Theta\boldsymbol{x}_i)\big)\boldsymbol{\sigma}(\Theta\boldsymbol{x}_i)_j\right.$$
$$\left. - \mathbb{E}\big[\big(\boldsymbol{\gamma}^{*\top}\boldsymbol{\sigma}(\Theta^*\boldsymbol{x}_i) - \boldsymbol{\gamma}^\top\boldsymbol{\sigma}(\Theta\boldsymbol{x}_i)\big)\boldsymbol{\sigma}(\Theta\boldsymbol{x}_i)_j\big]\Big)\right|$$

$$\leq \left|\frac{1}{n}\sum_{i=1}^{n}u_i(\Theta\boldsymbol{x}_i)_j\right| + \left|\frac{4}{n}\sum_{i=1}^{n}\Big(\big(\boldsymbol{\gamma}^{*\top}\boldsymbol{\sigma}(\Theta^*\boldsymbol{x}_i) - \boldsymbol{\gamma}^\top\boldsymbol{\sigma}(\Theta\boldsymbol{x}_i)\big)\boldsymbol{\sigma}(\Theta\boldsymbol{x}_i)_j\zeta_i\right|$$

$$\leq \left|\frac{1}{n}\sum_{i=1}^{n}u_i(\Theta\boldsymbol{x}_i)_j\right| + \left|\frac{4}{n}\sum_{i=1}^{n}\big(\boldsymbol{\gamma}^{*\top}\boldsymbol{\sigma}(\Theta^*\boldsymbol{x}_i)\big)\boldsymbol{\sigma}(\Theta\boldsymbol{x}_i)_j\zeta_i\right| + \left|\frac{4}{n}\sum_{i=1}^{n}\big(\boldsymbol{\gamma}^\top\boldsymbol{\sigma}(\Theta\boldsymbol{x}_i)\big)\boldsymbol{\sigma}(\Theta\boldsymbol{x}_i)_j\zeta_i\right|.$$

Treating the last two terms: we use Hölder's inequality to obtain

$$\left|\frac{4}{n}\sum_{i=1}^{n}\big(\boldsymbol{\gamma}^\top\boldsymbol{\sigma}(\Theta\boldsymbol{x}_i)\big)\boldsymbol{\sigma}(\Theta\boldsymbol{x}_i)_j\zeta_i\right| \leq \|\boldsymbol{\gamma}\|_1\Big\|\frac{4}{n}\sum_{i=1}^{n}\boldsymbol{\sigma}(\Theta\boldsymbol{x}_i)\boldsymbol{\sigma}(\Theta\boldsymbol{x}_i)_j\zeta_i\Big\|_\infty,$$

where $\boldsymbol{z}_i = \boldsymbol{\sigma}(\Theta\boldsymbol{x}_i)\boldsymbol{\sigma}(\Theta\boldsymbol{x}_i)_j\zeta_i$ are, mean-zero and independent sub-exponential random vectors (again can be followed as in Lemma 2).

The same is also true for

$$\left|\frac{4}{n}\sum_{i=1}^{n}\big(\boldsymbol{\gamma}^{*\top}\boldsymbol{\sigma}(\Theta^*\boldsymbol{x}_i)\big)\boldsymbol{\sigma}(\Theta\boldsymbol{x}_i)_j\zeta_i\right| \leq \|\boldsymbol{\gamma}^*\|_1\Big\|\frac{4}{n}\sum_{i=1}^{n}\boldsymbol{\sigma}(\Theta^*\boldsymbol{x}_i)\boldsymbol{\sigma}(\Theta\boldsymbol{x}_i)_j\zeta_i\Big\|_\infty$$

with $\boldsymbol{z}_i = \boldsymbol{\sigma}(\Theta^*\boldsymbol{x}_i)\boldsymbol{\sigma}(\Theta\boldsymbol{x}_i)_j\zeta_i$ again as independent with zero mean sub-exponential random vectors. $\qquad\square$

as desired.

# D   Appendix: Complementary simulations

We show the log-training error for shallow linear and shallow ReLU neural networks in Figure 3. To extend the simulations in Section 4, we show the relative error and test error for a different setting (with $d = 100, w = 20$) in Table 2. Moreover, we run our experiments in the numerical observations section 200 times (each time we run 100 runs to compute the potential global optimum and approximate stationary point) to reach the

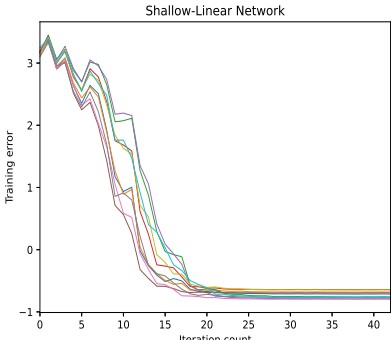 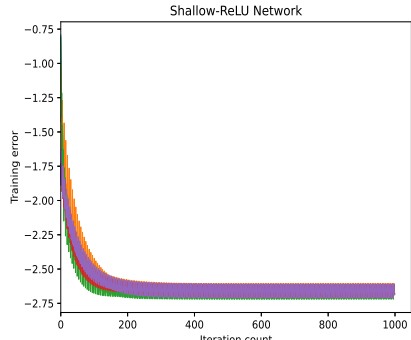

Figure 3: Log-training error for neural networks (with $d = w = 10$) with linear (left panel) and ReLU (right panel) activations in 10 different runs (allocated with different colors). Due to the non-convexity of neural networks, optimization algorithms may end up in different approximate stationary points.

mean and standard deviation of the relative error for the approximate stationary point. For the network with $d = w = 10$ and linear activation function, we reach the relative training error $1.0013 \pm 0.0003$ and relative test error $1.0011 \pm 0.0003$. For the ReLU activation function, we reach the relative training error $1.004 \pm 0.001$ and relative test error $1.005 \pm 0.001$. The same experiment for the larger network ($d = 100, w = 20$), concludes $1.04 \pm 0.01$, $1.03 \pm 0.008$, $1.89 \pm 0.07$, and $1.40 \pm 0.08$ for the relative training and test error of linear and ReLU activations, respectively. These results show that our empirical observations are stable. All the simulations were executed on a local computer (Apple M2, 16GB memory), with an average run time of less than 10 minutes per individual run in Python. For optimization, we employed SGD with the learning rate 0.02.

Table 2: relative training error and test error for trained neural networks (with $d = 100, w = 20$) with linear and ReLU activations in a potential global optimum, an approximate stationary point, and a randomly generated network.

|  | Linear | | ReLU | |
|---|---|---|---|---|
|  | Training Error | Test Error | Training Error | Test Error |
| Potential Global Optimum | 1.00 | 1.00 | 1.00 | 1.00 |
| Approximate Stationary Point | 1.04 | 1.03 | 1.85 | 1.10 |
| Randomly Generated Network | 1146373.94 | 1095543.69 | 5062.83 | 3626.28 |

In Tables 3 and 4 , we repeat the experiment from Section 4, this time employing different initialization strategies—namely the random Gaussian initialization and its scaled variant. For the random Gaussian initialization, weights are drawn independently from a standard normal distribution. In the scaled version, the weights are subsequently rescaled so that the $\ell_1$-norm of each layer individually satisfies $\|W\|_1 \leq \sqrt{\log n}$. Results in Table 4 perfectly match our previous observations in Section 4. Thanks to our initialization technique, we now expect the weight matrices to also satisfy the required assumption for ReLU networks (see further discussion following Theorem 3 that random Gaussian weights yield nearly orthogonal rows with high probability). Our results in Table 3 show that, since the weights are not scaled and their norm bounds are large, the behavior of approximate stationary points does not closely match that of the global minimum. This clearly indicates that initializing weights with small values significantly aids the optimization (supporting the need for our reasonability assumption). We also conducted experiments with a larger tuning rate, namely of the order $\log(np)/n^{1/4}$, as shown in Table 5. Comparing these results with Table 1 clearly demonstrates the optimality of the tuning rate $\log(np)/\sqrt{n}$ (vs $\log(np)/n^{1/4}$) supporting our proposed oracle tuning in equation 4. We also examined the relative error of the regularized estimator (same setting as Section 4) across a range of tuning parameters, obtained by multiplying a base tuning value by different factors, as shown in Figure 4 for both linear and ReLU networks. The results clearly illustrate a bias–variance trade-off when the tuning parameter is either too large or too small.

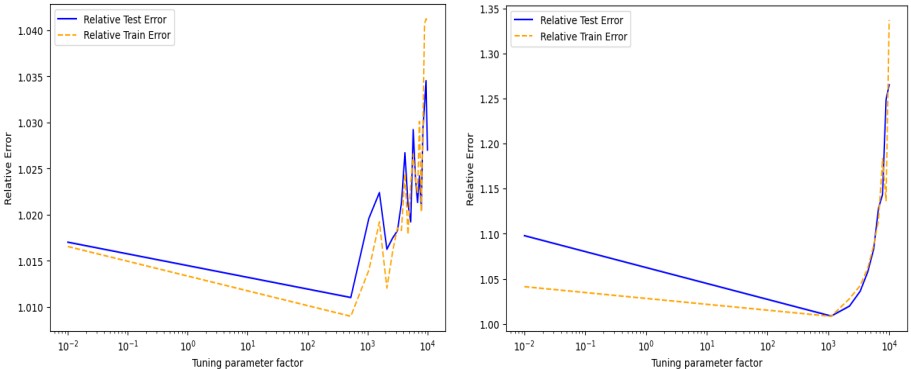

Figure 4: Relative error versus tuning parameter for shallow networks. Left: linear activation; right: ReLU activation. The results clearly illustrate a bias–variance trade-off when the tuning parameter is either too large or too small.

Table 3: Relative training error and test error for trained shallow neural networks (with $d = 10, w = 10$) with linear and ReLU activations in a potential global optimum, an approximate stationary point, and a randomly generated network employing random Gaussian initialization.

|  | Linear | | ReLU | |
|---|---|---|---|---|
|  | Training Error | Test Error | Training Error | Test Error |
| Potential Global Optimum | 1.00 | 1.00 | 1.00 | 1.00 |
| Approximate Stationary Point | 1.13 | 1.15 | 3.69 | 3.61 |
| Randomly Generated Network | 22576.49 | 17970.63 | 4157.68 | 2835.71 |

*Beyond SGD:* For the sake of completeness, we have now included further simulations to assess the impact of changing the optimization method. Specifically, we replaced SGD with Adam, using a learning rate of 0.005, to analyze its effect on the simulation outcomes in Table 1. Our results are reported in Table 6. These results show that the performance of SGD appears to be more aligned with our case (compare results in Table 6 with Table 1) which is high likely due to the verification of our assumptions for the corresponding approximate stationary point, but in general, approximate sub-optimal solutions remain still satisfactory.

*Conjecture for deep neural networks:* We have now extended our simulations in Table 1 employing neural networks with 4 layers. Our numerical observations make this conjecture that our theory can also hold for deep networks (with possibly minor different rates), given we reached the results in Table 7.

*Conjecture beyond regression:* We have now extended our simulations by employing more complex networks and testing beyond our regression simulated data. We applied our method to the MNIST, fashion-MNIST, and K-MNIST dataset using cross-entropy loss, with a neural network consisting of 10-layer weight matrices and ReLU activations, with network width 50. Our results continue to support the same conclusion we aim to demonstrate for approximate sub-optimal in Table 8. This observation can support the conjecture that our results can be extended for classification settings and even for deep neural networks in further studies.

# E    Appendix: Relaxing the $\ell_1$-norm bound

In fact, the bound $\sqrt{\log n}$ is merely for convenience: it can be replaced by any fixed constant or another function that is increasing slowly in the sample size $n$. It basically means that $\ell_1$-norm bound can be replaced by $c\sqrt{\log n}$ (with $c \in (0, \infty)$ an arbitrary constant) or $q(n)$ that the function $q(\cdot)$ is just mildly increasing in the sample size $n$. What we end up by moving to these bounds is that our rates change to $O((\log n)^2 \sqrt{(\log(pn))/n})$ or $O((q(n))^4 \sqrt{(\log(pn))/n})$, respectively that makes sense once $c$ and $q(n)$ are mild.

Table 4: Relative training error and test error for trained shallow neural networks (with $d = 10, w = 10$) with linear and ReLU activations in a potential global optimum, an approximate stationary point, and a randomly generated network employing scaled random Gaussian initialization.

| | Linear | | ReLU | |
|---|---|---|---|---|
| | Training Error | Test Error | Training Error | Test Error |
| Potential Global Optimum | 1.000 | 1.0000 | 1.00 | 1.00 |
| Approximate Stationary Point | 1.001 | 1.0003 | 1.02 | 1.05 |
| Randomly Generated Network | 79618.240 | 58198.240 | 4130.04 | 3650.06 |

Table 5: Relative training error and test error for trained shallow neural networks (with $d = 10, w = 10$) with linear and ReLU activations in a potential global optimum, an approximate stationary point, and a randomly generated network employing larger tuning parameter $(\log(np)/n^{1/4})$.

| | Linear | | ReLU | |
|---|---|---|---|---|
| | Training Error | Test Error | Training Error | Test Error |
| Potential Global Optimum | 1.00 | 1.00 | 1.00 | 1.00 |
| Approximate Stationary Point | 1.01 | 1.009 | 1.10 | 1.06 |
| Randomly Generated Network | 11498.68 | 8271.43 | 4157.68 | 2835.71 |

More explicitly, let's define

$$r_{\mathrm{orc},q} \; := \; c'\big(q(n)\big)^3 \sqrt{\frac{\log(np)}{n}} \tag{15}$$

the oracle tuning parameter, where $c' \in (0, \infty)$ is a constant that depends only on the distributions of the inputs and noise. Then, we get the following result:

**Theorem 5** (Statistical Guarantees for Norm-Bounded Stationary Points of Shallow Linear Networks). *Suppose that the second and the third part of Assumption 1 are satisfied and that $\|\boldsymbol{\gamma}^*\|_1, \|\Theta^*\|_1 \leq q(n)$ for a fixed function $q(n) \in (0, \infty)$. Then, any reasonable stationary point $(\widetilde{\boldsymbol{\gamma}}, \widetilde{\Theta})$ of the objective function in equation 2 with $r \geq r_{\mathrm{orc},q}$ satisfies the risk bound*

$$\mathrm{risk}[\widetilde{\boldsymbol{\gamma}}, \widetilde{\Theta}] \; \leq \; \mathrm{risk}[\boldsymbol{\gamma}^*, \Theta^*] + 5rq(n) \tag{16}$$

*with probability at least $1 - 1/2n$.*

In the theorem above, 1. $(\boldsymbol{\gamma}^*, \Theta^*)$ is a pair that approximates the target function and 2. by reasonable stationary, we mean that $\|\widetilde{\boldsymbol{\gamma}}\|_1, \|\widetilde{\Theta}\|_1 \leq q(n)$. The proof of this theorem follows the same steps as our Theorem 1 and so we omit the proof.

Another interesting and practical point in the training process of deep learning is that neural network weights are usually initialized by near-zero values. For example, PyTorch by default initializes weights as uniform$(-1/\sqrt{p}, 1/\sqrt{p})$ ($p$ refers to the number of parameters in the network), that means the $\ell_1-$norm of the matrix and vector weights are very small. Then, in the training process, the optimization algorithm looks for a stationary point around the initialized network (and not too far from this space). So, it is more likely that the computed (approximate) stationary point has a small norm, while there might also exist other stationeries with larger norms. This argument shows that even from a practical point of view, the reasonability assumption on stationary points and the points nearby makes sense.

## F  Appendix: On the reasonability assumption on the stationary points and the points nearby

It is stated in the text that the reasonability assumption on the stationary points makes sense. Here, we prove that claim by showing that the reasonability assumption on the target also implies reasonability on the stationary points.

Table 6: Relative training error and test error for trained shallow neural networks (with $d = 10, w = 10$) with linear and ReLU activations in an approximate stationary point employing Adam.

| | Linear | | ReLU | |
| --- | --- | --- | --- | --- |
| | Training Error | Test Error | Training Error | Test Error |
| Approximate Stationary Point | 1.0007 | 1.003 | 1.20 | 1.27 |

Table 7: Relative training error and test error for trained neural networks (with $d = 10, w = 10$, and depth 4) with linear and ReLU activations in an approximate stationary point.

| | Linear | | ReLU | |
| --- | --- | --- | --- | --- |
| | Training Error | Test Error | Training Error | Test Error |
| Approximate Stationary Point | 1.002 | 1.004 | 1.16 | 1.21 |

Following the same lines as in the proof of Theorem 1, we have

$$
\begin{aligned}
\operatorname{risk}[\widetilde{\boldsymbol{\gamma}}, \widetilde{\Theta}] \leq{} & \operatorname{risk}[\boldsymbol{\gamma}^*, \Theta^*] + r\|\boldsymbol{\beta}^*\|_1 + \left|\left(\nabla \operatorname{risk}_X[\widetilde{\boldsymbol{\gamma}}, \widetilde{\Theta}] - \nabla \operatorname{risk}[\widetilde{\boldsymbol{\gamma}}, \widetilde{\Theta}]\right)^\top (\boldsymbol{\beta}^* - \widetilde{\boldsymbol{\beta}})\right| - \frac{1}{2}r\|\widetilde{\boldsymbol{\beta}}\|_1 \\
& - \frac{1}{2}r\|\widetilde{\boldsymbol{\beta}}\|_1 - \frac{1}{2}m \\
={} & \operatorname{risk}[\boldsymbol{\gamma}^*, \Theta^*] + \frac{3}{2}r\|\boldsymbol{\beta}^*\|_1 + \left|\left(\nabla \operatorname{risk}_X[\widetilde{\boldsymbol{\gamma}}, \widetilde{\Theta}] - \nabla \operatorname{risk}[\widetilde{\boldsymbol{\gamma}}, \widetilde{\Theta}]\right)^\top (\boldsymbol{\beta}^* - \widetilde{\boldsymbol{\beta}})\right| \\
& - \frac{1}{2}r\left(\|\widetilde{\boldsymbol{\beta}}\|_1 + \|\boldsymbol{\beta}^*\|_1\right) - \frac{1}{2}r\|\widetilde{\boldsymbol{\beta}}\|_1 - \frac{1}{2}m \\
\leq{} & \operatorname{risk}[\boldsymbol{\gamma}^*, \Theta^*] + \frac{3}{2}r\|\boldsymbol{\beta}^*\|_1 + \left|\left(\nabla \operatorname{risk}_X[\widetilde{\boldsymbol{\gamma}}, \widetilde{\Theta}] - \nabla \operatorname{risk}[\widetilde{\boldsymbol{\gamma}}, \widetilde{\Theta}]\right)^\top (\boldsymbol{\beta}^* - \widetilde{\boldsymbol{\beta}})\right| - \frac{1}{2}r\|\boldsymbol{\beta}^* - \widetilde{\boldsymbol{\beta}}\|_1 \\
& - \frac{1}{2}r\|\widetilde{\boldsymbol{\beta}}\|_1 - \frac{1}{2}m \\
\leq{} & \operatorname{risk}[\boldsymbol{\gamma}^*, \Theta^*] + \frac{3}{2}r\|\boldsymbol{\beta}^*\|_1 + r_{\mathrm{orc}}\|\boldsymbol{\beta}^* - \widetilde{\boldsymbol{\beta}}\|_1 + \frac{r_{\mathrm{orc}}}{2n} - \frac{1}{2}r\|\boldsymbol{\beta}^* - \widetilde{\boldsymbol{\beta}}\|_1 - \frac{1}{2}r\|\widetilde{\boldsymbol{\beta}}\|_1 - \frac{1}{2}m \,.
\end{aligned}
$$

Moreover,

$$
\operatorname{risk}[\widetilde{\boldsymbol{\gamma}}, \widetilde{\Theta}] + \frac{1}{2}r\|\widetilde{\boldsymbol{\beta}}\|_1 \leq \operatorname{risk}[\boldsymbol{\gamma}^*, \Theta^*] + \frac{3}{2}r\|\boldsymbol{\beta}^*\|_1 + r_{\mathrm{orc}}\|\boldsymbol{\beta}^* - \widetilde{\boldsymbol{\beta}}\|_1 + \frac{r_{\mathrm{orc}}}{2n} - \frac{1}{2}r\|\boldsymbol{\beta}^* - \widetilde{\boldsymbol{\beta}}\|_1 - \frac{1}{2}m \,.
$$

Then, by considering $r \geq 2r_{\mathrm{orc}}$ we have

$$
\operatorname{risk}[\widetilde{\boldsymbol{\gamma}}, \widetilde{\Theta}] + \frac{1}{2}r\|\widetilde{\boldsymbol{\beta}}\|_1 \leq \operatorname{risk}[\boldsymbol{\gamma}^*, \Theta^*] + \frac{3}{2}r\|\boldsymbol{\beta}^*\|_1 + \frac{r_{\mathrm{orc}}}{2n} - \frac{1}{2}m \,.
$$

Following the same argument for $m$ as in the proof of Theorem 1, we obtain

$$
\operatorname{risk}[\widetilde{\boldsymbol{\gamma}}, \widetilde{\Theta}] + \frac{1}{2}r\|\widetilde{\boldsymbol{\beta}}\|_1 \leq \operatorname{risk}[\boldsymbol{\gamma}^*, \Theta^*] + \frac{3}{2}r\|\boldsymbol{\beta}^*\|_1 + \frac{r_{\mathrm{orc}}}{2n}
$$

and

$$
\frac{1}{2}r_{\mathrm{orc}}\|\widetilde{\boldsymbol{\beta}}\|_1 \leq \operatorname{risk}[\boldsymbol{\gamma}^*, \Theta^*] + \frac{3}{2}r_{\mathrm{orc}}\|\boldsymbol{\beta}^*\|_1 + \frac{r_{\mathrm{orc}}}{2n} \,.
$$

Finally, by assuming a small variance in the noise and reasonability assumptions on the target, we can conclude (for large $n$) that

$$
\|\widetilde{\boldsymbol{\beta}}\|_1 \lessapprox 3\|\boldsymbol{\beta}^*\|_1 + \frac{1}{n} \leq 4\|\boldsymbol{\beta}^*\|_1 \leq 4\sqrt{\log n} \,.
$$

The above display reveals that having a reasonability assumption on the target can also imply reasonability on the stationary points as well, once tuning is selected large enough, which also implies reasonability on the points nearby.

Table 8: Relative training error and test error for trained neural networks (with $w = 50$ and depth 10) with ReLU activations in an approximate stationary point.

|  | ReLU | |
| --- | --- | --- |
|  | Training Error | Test Error |
| Approximate Stationary Point (MNIST) | 1.0004 | 1.39 |
| Approximate Stationary Point (Fashion-MNIST) | 1.00005 | 1.40 |
| Approximate Stationary Point (K-MNIST) | 1.00003 | 1.18 |

## G    Appendix: Dynamical accessibility of approximate stationary points

In this section, we argue that $\tau$-approximate stationary points can be reached in practice (in a reasonable time) once gradient-based algorithms iterate sufficiently.

For non-convex and differentiable objectives $\ell(\boldsymbol{\beta})$ with gradient-based methods, dynamical accessibility of approximate stationaries $\widetilde{\widetilde{\boldsymbol{\beta}}} \in \mathcal{B}$ (points with small gradients $\|\nabla \ell(\widetilde{\widetilde{\boldsymbol{\beta}}})\| \leq \tau'$ that $\tau' \in (0, \infty)$) have widely been studied (Ghadimi & Lan, 2013; Carmon et al., 2018; Wang & Srebro, 2019; Lei et al., 2019; Drori & Shamir, 2020; Arjevani et al., 2022).

Here, we provide some results from Ghadimi & Lan (2013) and Lei et al. (2019). Before going through the main results, we impose some assumptions:

$$\mathbb{E}_z[g(\boldsymbol{\beta}, z)] = \nabla \ell(\boldsymbol{\beta}), \qquad \exists\, \sigma_{\mathrm{g}} \in (0, \infty) : \mathbb{E}_z \|g(\boldsymbol{\beta}, z) - \nabla \ell(\boldsymbol{\beta})\|^2 \leq \sigma_{\mathrm{g}}^2, \tag{17}$$

where $g(\boldsymbol{\beta}, z)$ is an estimator of $\nabla \ell(\boldsymbol{\beta})$ computed using a subsets of samples called $z$. And

$$\exists\, \Delta, L_{\mathrm{g}} \in (0, \infty) : \ell(\boldsymbol{\beta}^{(0)}) - \inf_{\boldsymbol{\beta} \in \mathcal{B}} \ell(\boldsymbol{\beta}) \leq \Delta, \qquad \|\nabla \ell(\boldsymbol{\beta}) - \nabla \ell(\boldsymbol{\beta}')\| \leq L_{\mathrm{g}} \|\boldsymbol{\beta} - \boldsymbol{\beta}'\| \quad \forall \boldsymbol{\beta}, \boldsymbol{\beta}' \in \mathcal{B}, \tag{18}$$

where $\ell(\boldsymbol{\beta}^{(0)})$ is the value of the objective function in the initialized step. Then, Ghadimi & Lan (2013, Theorem 2.1) prove that SGD finds an estimator such that $\mathbb{E}[\|\nabla \ell(\boldsymbol{\beta}^{(R)})\|] \leq \tau'$ for a randomly selected $R \in \{1, \ldots, T\}$ (according to a certain probability distribution, see Ghadimi & Lan (2013, Equation 2.3)), where the expectation is taken over $R$ and the randomness of SGD, using $O(\Delta L_{\mathrm{g}} \sigma_{\mathrm{g}}^2 / (\tau')^4)$ oracle queries. Above result also imply $\min_{t \in \{1, \ldots, T\}} \mathbb{E}[\|\nabla \ell(\boldsymbol{\beta}^{(t)})\|] \leq \tau'$ using $O(\Delta L_{\mathrm{g}} \sigma_{\mathrm{g}}^2 / (\tau')^4)$ oracle queries.

We can argue that Assumptions equation 17 and equation 18 can hold in the setting of our paper: for Assumption equation 17 and the first part of Assumption equation 18 (objective has bounded initial suboptimality), we can use the reasonability assumption over the parameter space. For twice-differentiable objectives, the second part of Assumption equation 18 means that the eigenvalues of the objective's Hessian are bounded above by $L_{\mathrm{g}}$, which is typically a reasonable assumption.

Important here is that $\mathbb{E}[\|\nabla \ell(\boldsymbol{\beta}^{(R)})\|] \leq \tau'$ and our definition of approximate stationary points in equation 7 are in a sense similar. Using 1. the definition of the objective function, 2. a first order Taylor expansion of $\ell(\widetilde{\boldsymbol{\beta}})$ around $\ell(\widetilde{\widetilde{\boldsymbol{\beta}}})$ (with $\widetilde{\widetilde{\boldsymbol{\beta}}} := \boldsymbol{\beta}^R$), 3. Hölder's inequality, 4 our definition of $\widetilde{\widetilde{\boldsymbol{\beta}}}$, result above, and the reasonability of approximate stationary and exact stationary we obtain

$$\begin{aligned} \mathrm{risk}_X[\widetilde{\widetilde{\boldsymbol{\gamma}}}, \widetilde{\widetilde{\Theta}}] + r\|\widetilde{\widetilde{\boldsymbol{\beta}}}\|_1 - \mathrm{risk}_X[\widetilde{\boldsymbol{\gamma}}, \widetilde{\Theta}] - r\|\widetilde{\boldsymbol{\beta}}\|_1 &= \ell(\widetilde{\widetilde{\boldsymbol{\beta}}}) - \ell(\widetilde{\boldsymbol{\beta}}) \\ &\approx \left(\nabla \ell(\widetilde{\widetilde{\boldsymbol{\beta}}})\right)^{\top}(\widetilde{\boldsymbol{\beta}} - \widetilde{\widetilde{\boldsymbol{\beta}}}) \\ &\leq \|\nabla \ell(\widetilde{\widetilde{\boldsymbol{\beta}}})\| \|\widetilde{\boldsymbol{\beta}} - \widetilde{\widetilde{\boldsymbol{\beta}}}\| \\ &\leq c\tau' \sqrt{\log n} \end{aligned}$$

for a constant $c \in (0, \infty)$. It means that having a small norm on the gradients of approximate stationary can also imply a small difference between the objective function of the approximate stationary and exact stationary. The results of Ghadimi & Lan (2013) imply that gradient-based algorithms with sufficiently many steps, let's say $O(n^2)$, can guarantee small $\tau \in O(1/\sqrt{n})$.

Lei et al. (2019, Theorem 3) prove that for differentiable loss functions with $\alpha$-Hölder continuous gradients:

$$\exists\, L_{g,\alpha} \in (0,\infty) : \|\nabla\ell(\boldsymbol{\beta}) - \nabla\ell(\boldsymbol{\beta}')\| \leq L_{g,\alpha}\|\boldsymbol{\beta} - \boldsymbol{\beta}'\|^\alpha \quad \forall \boldsymbol{\beta}, \boldsymbol{\beta}' \in \mathcal{B} \tag{19}$$

where $\alpha \in (0,1]$ and $L_{g,\alpha} \in (0,\infty)$, SGD gets

$$\min_{t\in\{1,\ldots,T\}} \mathbb{E}\big[\|\nabla\ell(\boldsymbol{\beta}^{(t)})\|^2\big] \leq C\left(\sum_{i=1}^T \eta_t\right)^{-1} =: \tau'',$$

where $C$ is a constant independent of $t$, $\eta_t$ are stepsizes satisfying $\sum_{t=1}^\infty \eta_t^{1+\alpha} < \infty$, and the expectation is taken over the randomness of SGD. Lei et al. (2019, Theorem 3) reveal a rate of convergence $1/T$ for the smallest gradient. As a comparison, the convergence rate in Lei et al. (2019, Theorem 3) only holds for the minimum of the first $T$ iterates, while the convergence rate in Ghadimi & Lan (2013, Theorem 2.1) holds for $\mathbb{E}[\|\nabla\ell(\boldsymbol{\beta}^{(R)})\|]$ that is more practical (we also used Ghadimi & Lan (2013, Theorem 2.1)).

## H  Appendix: Heavier-tailed noise

In this section, we are motivated to provide materials proving our Theorem 4.

First, we present an adapted version of the result in Bakhshizadeh et al. (2020, Corollary 2):

**Lemma 10** (Empirical Processes for Heavy-Tailed Data). *Suppose $z_1, \ldots, z_n$ are centered i.i.d. random variables whose tail is captured by $I_\alpha(t) = c_\alpha t^{1/\alpha}$ for some $\alpha \in [1,\infty)$ and $c_\alpha \in (0,\infty)$. Moreover, assume $\mathbb{E}[z^2 \mathbf{1}(z \leq 0)] = (\sigma_\alpha)^2 < \infty$. Then, for all $t \in [0,\infty)$ we have*

$$\mathbb{P}\left(\left|\frac{1}{n}\sum_{i=1}^n z_i\right| > t\right) \leq 6n\exp(-c\min\{nt^2, (nt)^{1/\alpha}\}), \tag{20}$$

*where $c$ is a constant depending on the distribution of $z_i$.*

*Proof of Lemma 10.* The lemma is just an adapted version of Bakhshizadeh et al. (2020, Corollary 2) and reached in three steps:

Step 1: We use the result in Bakhshizadeh et al. (2020, Corollary 2) that gives

$$\mathbb{P}\left(\frac{1}{n}\sum_{i=1}^n z_i > t\right) \leq \exp\left(-\frac{nt^2}{2\bar{v}(nt,\beta)}\right) + \exp(-\beta\max\{c_t, 0.5\}c_\alpha(nt)^{1/\alpha}) + n\exp(-c_\alpha(nt)^{1/\alpha}), \tag{21}$$

where $\beta \in (0,1)$ is arbitrary, $c_t \in (0,1)$ is a constant depending on $n$ and $t$, and

$$\bar{v}(nt,\beta) := (\sigma_\alpha)^2 + \frac{\Gamma(2\alpha+1)}{\big((1-\beta)c_\alpha\big)^{2\alpha}} + (nt)^{(1/\alpha)-1}\frac{\beta c_\alpha \Gamma(3\alpha+1)}{3\big((1-\beta)c_\alpha\big)^{3\alpha}}.$$

Step 2: Since the factors $c_t \in (0,1)$ and $\bar{v}(nt,\beta)$ depend on $n$ and $t$, we need to remove this dependence, otherwise we are in trouble. We can easily remove the constant $c_t$ from equation 21 because there is a max function there. Also, the factor $\bar{v}(nt,\beta)$ in the rate above is basically bounded from above. For example, for large enough $n$ ($t > 1/n$) and specific $\beta = 1/2$ we have

$$\bar{v}(nt,\beta) \leq v_\alpha := \sigma_\alpha{}^2 + \frac{\Gamma(2\alpha+1)}{c_1^{2\alpha}} + \frac{c_\alpha\Gamma(3\alpha+1)}{3c_1^{3\alpha}},$$

where $c_1 \in (0,\infty)$ is a constant. Then, we reach

$$\mathbb{P}\left(\frac{1}{n}\sum_{i=1}^n z_i > t\right) \leq 3n\exp(-c\min\{nt^2, (nt)^{1/\alpha}\}),$$

where $c := \min\{1/2v_\alpha, c_\alpha/4, c_\alpha\}$.

Step 3: We use the symmetry of random variables $z_i$ moving to a two-sided tail by paying a factor of two as desired.

$\square$

Using the above lemma, we derive a uniform bound on the absolute difference between $\text{risk}_X[\boldsymbol{\gamma}, \Theta]$ and $\text{risk}[\boldsymbol{\gamma}, \Theta]$ for heavier-tailed noise.

**Lemma 11** (Difference Between $\nabla\text{risk}_X[\boldsymbol{\gamma}, \Theta]$ and $\nabla\text{risk}[\boldsymbol{\gamma}, \Theta]$ for Heavier-tailed Noise)**.** *Under the first two parts of Assumption 1, it holds for each $t, \eta, \epsilon \in (0, \infty)$ and $\boldsymbol{\beta} \in \mathcal{C}_{\eta,\epsilon} := \{\boldsymbol{\beta} = \text{vec}(\boldsymbol{\gamma}, \Theta) \in \mathbb{R}^p : \|\boldsymbol{\beta}^* - \boldsymbol{\beta}\|_1 \le \eta \text{ and } \|\boldsymbol{\gamma}^\top \Theta - \boldsymbol{\gamma}^{*\top}\Theta^*\|_1 \le \epsilon\}$ that*

$$\sup_{\boldsymbol{\beta} \in \mathcal{C}_{\eta,\epsilon}} \left|\left(\nabla\text{risk}_X[\boldsymbol{\gamma}, \Theta] - \nabla\text{risk}[\boldsymbol{\gamma}, \Theta]\right)^\top (\boldsymbol{\beta}^* - \boldsymbol{\beta})\right| \le 2t\eta\big(\eta + \max\{\|\boldsymbol{\gamma}^*\|_\infty, \|\Theta^*\|_\infty\}\big)\big(1 + \epsilon\big)$$

*with probability at least $1 - 12d^2pn \exp(-c\min\{nt^2, (nt)^{1/\alpha}\})$ with constants $c \in (0, \infty)$ and $\alpha \in [2, \infty)$ depending only on the distributions of the inputs and noise.*

*Proof of Lemma 11.* The proof follows almost the same steps as in the proof of Lemma 2. The only difference is handling the empirical processes parts.

We start the proof with Hölder's inequality and the definition of $\mathcal{C}_{\eta,\epsilon}$, which implies $\|\boldsymbol{\beta}^* - \boldsymbol{\beta}\|_1 \le \eta$ for all $\boldsymbol{\beta} \in \mathcal{C}_{\eta,\epsilon}$ to obtain

$$\sup_{\boldsymbol{\beta}=\text{vec}(\boldsymbol{\gamma},\Theta)\in\mathcal{C}_{\eta,\epsilon}} \left|\left(\nabla\text{risk}_X[\boldsymbol{\gamma}, \Theta] - \nabla\text{risk}[\boldsymbol{\gamma}, \Theta]\right)^\top (\boldsymbol{\beta}^* - \boldsymbol{\beta})\right|$$
$$\le \sup_{\boldsymbol{\beta}=\text{vec}(\boldsymbol{\gamma},\Theta)\in\mathcal{C}_{\eta,\epsilon}} \left(\left\|\nabla\text{risk}_X[\boldsymbol{\gamma}, \Theta] - \nabla\text{risk}[\boldsymbol{\gamma}, \Theta]\right\|_\infty \|\boldsymbol{\beta}^* - \boldsymbol{\beta}\|_1\right)$$
$$\le \eta \sup_{\boldsymbol{\beta}=\text{vec}(\boldsymbol{\gamma},\Theta)\in\mathcal{C}_{\eta,\epsilon}} \left\|\nabla\text{risk}_X[\boldsymbol{\gamma}, \Theta] - \nabla\text{risk}[\boldsymbol{\gamma}, \Theta]\right\|_\infty.$$

The rest of the proof employs our Lemma 5 and Lemma 10 to find an upper bound for $\sup_{\boldsymbol{\beta}=\text{vec}(\boldsymbol{\gamma},\Theta)\in\mathcal{C}_{\eta,\epsilon}} \|\nabla\text{risk}_X[\boldsymbol{\gamma}, \Theta] - \nabla\text{risk}[\boldsymbol{\gamma}, \Theta]\|_\infty$. Note that for simplifying the notation, we use $\mathbb{E}[\cdot]$ as a shorthand notation of $\mathbb{E}_{(\boldsymbol{x}_1,y_1),\dots,(\boldsymbol{x}_n,y_n)}[\cdot]$ throughout this proof.

We use 1. our result in Lemma 5 and i.i.d. assumption on the data, 2. equation 1 and our assumption that $f[\boldsymbol{x}] = \boldsymbol{\gamma}^{*\top}\Theta^*\boldsymbol{x}$, zero-mean noise, linearity of expectations, and factorizing, 3. the definition of sup-norm, triangle inequality, and Hölder's inequality, 4. the definition of $\mathcal{C}_{\eta,\epsilon}$, which implies $\|\boldsymbol{\gamma}^{*\top}\Theta^* - \boldsymbol{\gamma}^\top\Theta\|_1 \le \epsilon$, 5. adding a zero-valued term and rewriting, and 6. the triangle inequality and the definition of $\mathcal{C}_{\eta,\epsilon}$, which implies $\|\boldsymbol{\gamma} - \boldsymbol{\gamma}^*\|_1 \le \|\boldsymbol{\beta} - \boldsymbol{\beta}^*\|_1 \le \eta$, to obtain for each $j \in \{1, \dots, w\}$ and $k \in \{1, \dots, d\}$ that

$$\left|\frac{\partial}{\partial\theta_{jk}}\text{risk}_X[\boldsymbol{\gamma}, \Theta] - \frac{\partial}{\partial\theta_{jk}}\text{risk}[\boldsymbol{\gamma}, \Theta]\right|$$
$$= \left|-\frac{2}{n}\sum_{i=1}^n (y_i - \boldsymbol{\gamma}^\top\Theta\boldsymbol{x}_i)\gamma_j(\boldsymbol{x}_i)_k + \mathbb{E}\left[\frac{2}{n}\sum_{i=1}^n (y_i - \boldsymbol{\gamma}^\top\Theta\boldsymbol{x}_i)\gamma_j(\boldsymbol{x}_i)_k\right]\right|$$
$$= 2|\gamma_j|\left|\frac{1}{n}\sum_{i=1}^n \Big(u_i(\boldsymbol{x}_i)_k + (\boldsymbol{\gamma}^{*\top}\Theta^* - \boldsymbol{\gamma}^\top\Theta)\big(\boldsymbol{x}_i(\boldsymbol{x}_i)_k - \mathbb{E}[\boldsymbol{x}_i(\boldsymbol{x}_i)_k]\big)\Big)\right|$$
$$\le 2\|\boldsymbol{\gamma}\|_\infty\left(\left|\frac{1}{n}\sum_{i=1}^n u_i(\boldsymbol{x}_i)_k\right| + \|\boldsymbol{\gamma}^\top\Theta - \boldsymbol{\gamma}^{*\top}\Theta^*\|_1\left\|\frac{1}{n}\sum_{i=1}^n \big(\mathbb{E}[\boldsymbol{x}_i(\boldsymbol{x}_i)_k] - \boldsymbol{x}_i(\boldsymbol{x}_i)_k\big)\right\|_\infty\right)$$
$$\le 2\|\boldsymbol{\gamma}\|_\infty\left(\left|\frac{1}{n}\sum_{i=1}^n u_i(\boldsymbol{x}_i)_k\right| + \epsilon\left\|\frac{1}{n}\sum_{i=1}^n \big(\mathbb{E}[\boldsymbol{x}_i(\boldsymbol{x}_i)_k] - \boldsymbol{x}_i(\boldsymbol{x}_i)_k\big)\right\|_\infty\right)$$
$$= 2\|\boldsymbol{\gamma} - \boldsymbol{\gamma}^* + \boldsymbol{\gamma}^*\|_\infty\left(\left|\frac{1}{n}\sum_{i=1}^n u_i(\boldsymbol{x}_i)_k\right| + \epsilon\left\|\frac{1}{n}\sum_{i=1}^n \big(\boldsymbol{x}_i(\boldsymbol{x}_i)_k - \mathbb{E}[\boldsymbol{x}_i(\boldsymbol{x}_i)_k]\big)\right\|_\infty\right)$$

$$\leq 2(\eta + \|\boldsymbol{\gamma}^*\|_\infty) \left( \left| \frac{1}{n} \sum_{i=1}^n u_i(\boldsymbol{x}_i)_k \right| + \epsilon \left\| \frac{1}{n} \sum_{i=1}^n (\boldsymbol{x}_i(\boldsymbol{x}_i)_k - \mathbb{E}[\boldsymbol{x}_i(\boldsymbol{x}_i)_k]) \right\|_\infty \right).$$

We continue to work on the absolute value and sup-norm term in the last inequality above separately. For each $i \in \{1, \ldots, n\}$ and $k \in \{1, \ldots, d\}$, we use our assumptions on $\boldsymbol{x}_i$ and $u_i$ to obtain that $z_i = u_i(\boldsymbol{x}_i)_k$ are i.i.d. random variables with zero-mean and their tail is captured by $c_\alpha(t)^{1/\alpha}$ for some $\alpha \in [2, \infty)$ and $c_\alpha \in (0, \infty)$, depending on the noise and input distributions. We are using the fact that the product of two random variables with tail parameters $\alpha_1$ and $\alpha_2$ has the tail parameter $\alpha_1 + \alpha_2$ (Vladimirova et al., 2020, Proposition 2.3). And since we are assuming heavier-tailed noise it implies $z_i$ be at least sub-exponential with $\alpha = 2$ (recall that we assumed $\boldsymbol{x}_i$ are sub-gaussian). Employing Lemma 10, we obtain for each $t \in [0, \infty)$ that

$$\mathbb{P}\left( \left| \frac{1}{n} \sum_{i=1}^n u_i(\boldsymbol{x}_i)_k \right| \geq t \right) \leq 6n \exp(-c \min\{nt^2, (nt)^{1/\alpha}\}).$$

Now, we study the behavior of the sup-norm term in the last inequality of the earlier display. Let's rewrite the sup-norm in the form of max as

$$\left\| \frac{1}{n} \sum_{i=1}^n (\boldsymbol{x}_i(\boldsymbol{x}_i)_k - \mathbb{E}[\boldsymbol{x}_i(\boldsymbol{x}_i)_k]) \right\|_\infty = \max_{k' \in \{1, \ldots, d\}} \left| \frac{1}{n} \sum_{i=1}^n ((\boldsymbol{x}_i)_{k'}(\boldsymbol{x}_i)_k - \mathbb{E}[(\boldsymbol{x}_i)_{k'}(\boldsymbol{x}_i)_k]) \right|.$$

Following the same argument as earlier and for each $i \in \{1, \ldots, n\}$ and $k, k' \in \{1, \ldots, d\}$, we can employ Lemma 10 with $z_i = (\boldsymbol{x}_i)_{k'}(\boldsymbol{x}_i)_k$ to obtain for each $t' \in [0, \infty)$ that

$$\mathbb{P}\left( \left| \frac{1}{n} \sum_{i=1}^n ((\boldsymbol{x}_i)_{k'}(\boldsymbol{x}_i)_k - \mathbb{E}[(\boldsymbol{x}_i)_{k'}(\boldsymbol{x}_i)_k]) \right| \geq t' \right) \leq 6n \exp(-c' \min\{nt'^2, (nt')^{1/\alpha'}\}),$$

for some $\alpha' \in [1, \infty)$ and $c' \in (0, \infty)$, depending on the input distribution. Then, we use our result above together with the fact that if $\mathbb{P}(|b_i| \geq t) \leq a$ holds for all $i \in \{1, \ldots p\}$, then we also have $\mathbb{P}(\max_{i \in \{1, \ldots p\}} |b_i| \geq t) \leq pa$ to obtain

$$\mathbb{P}\left( \max_{k' \in \{1, \ldots, d\}} \left| \frac{1}{n} \sum_{i=1}^n ((\boldsymbol{x}_i)_{k'}(\boldsymbol{x}_i)_k - \mathbb{E}[(\boldsymbol{x}_i)_{k'}(\boldsymbol{x}_i)_k]) \right| \geq t' \right) \leq 6dn \exp(-c' \min\{nt'^2, (nt')^{1/\alpha'}\}).$$

Collecting all pieces above together with considering $t = t'$, we obtain for each $j \in \{1, \ldots, w\}$ and $k \in \{1, \ldots, d\}$ that

$$\left| \frac{\partial}{\partial \theta_{jk}} \text{risk}_X[\boldsymbol{\gamma}, \Theta] - \frac{\partial}{\partial \theta_{jk}} \text{risk}[\boldsymbol{\gamma}, \Theta] \right| \leq 2t(\eta + \|\boldsymbol{\gamma}^*\|_\infty)(1 + \epsilon)$$

with probability at least $1 - 6n \exp(-c \min\{nt^2, (nt)^{1/\alpha}\}) - 6dn \exp(-c' \min\{nt'^2, (nt')^{1/\alpha'}\})$, which is obtained using the fact that

$$P(A + bD \leq t + bt) = 1 - P(A + bD > t + bt) \geq 1 - P(A > t) - P(D > t)$$

for any $b \in (0, \infty)$ and $t \in \mathbb{R}$.

Then, we follow the same argument as earlier and use 1. our result in Lemma 5 and i.i.d. assumption on the data, 2. the properties of absolute values and linearity of expectations, 3. some rewriting, 4. Hölder's inequality, 5. equation 1 and our assumptions that $f[\boldsymbol{x}] = \boldsymbol{\gamma}^{*\top} \Theta^* \boldsymbol{x}$, zero-mean noise, and definition of sup-norm, 6. triangle inequality, compatible norms (for a matrix $A \in \mathbb{R}^{d \times d}$, we define $\|A\|_{\infty,1} := \max_{k \in \{1, \ldots, d\}} \sum_{k'=1}^d |A_{k',k}|$), and the definition of $\mathcal{C}_{\eta, \epsilon}$, which implies $\|\boldsymbol{\gamma}^{*\top} \Theta^* - \boldsymbol{\gamma}^\top \Theta\|_1 \leq \epsilon$, 7. adding a zero-valued term, 8. the triangle inequality and the definition of $\mathcal{C}_{\eta, \epsilon}$, which implies $\|\Theta - \Theta^*\|_1 \leq \|\boldsymbol{\beta} - \boldsymbol{\beta}^*\|_1 \leq \eta$ to obtain for each $j \in \{1, \ldots, w\}$ that

$$\left| \frac{\partial}{\partial \gamma_j} \text{risk}_X[\boldsymbol{\gamma}, \Theta] - \frac{\partial}{\partial \gamma_j} \text{risk}[\boldsymbol{\gamma}, \Theta] \right|$$

$$= \left| -\frac{2}{n}\sum_{i=1}^{n}\left((y_i - \boldsymbol{\gamma}^\top\Theta\boldsymbol{x}_i)(\Theta\boldsymbol{x}_i)_j\right) + \mathbb{E}\left[\frac{2}{n}\sum_{i=1}^{n}\left((y_i - \boldsymbol{\gamma}^\top\Theta\boldsymbol{x}_i)(\Theta\boldsymbol{x}_i)_j\right)\right]\right|$$

$$= \left| \frac{2}{n}\sum_{i=1}^{n}\left((y_i - \boldsymbol{\gamma}^\top\Theta\boldsymbol{x}_i)(\Theta\boldsymbol{x}_i)_j - \mathbb{E}[(y_i - \boldsymbol{\gamma}^\top\Theta\boldsymbol{x}_i)(\Theta\boldsymbol{x}_i)_j]\right)\right|$$

$$= \left| \frac{2}{n}\sum_{i=1}^{n}\left((y_i - \boldsymbol{\gamma}^\top\Theta\boldsymbol{x}_i)\boldsymbol{x}_i^\top\Theta_{j,\cdot} - \mathbb{E}[(y_i - \boldsymbol{\gamma}^\top\Theta\boldsymbol{x}_i)\boldsymbol{x}_i^\top\Theta_{j,\cdot}]\right)\right|$$

$$\leq \left\| \frac{2}{n}\sum_{i=1}^{n}\left((y_i - \boldsymbol{\gamma}^\top\Theta\boldsymbol{x}_i)\boldsymbol{x}_i^\top - \mathbb{E}[(y_i - \boldsymbol{\gamma}^\top\Theta\boldsymbol{x}_i)\boldsymbol{x}_i^\top]\right)\right\|_\infty \|\Theta_{j,\cdot}\|_1$$

$$\leq 2\|\Theta\|_\infty\left(\left\| \frac{1}{n}\sum_{i=1}^{n}\left(u_i\boldsymbol{x}_i^\top + (\boldsymbol{\gamma}^{*\top}\Theta^* - \boldsymbol{\gamma}^\top\Theta)(\boldsymbol{x}_i\boldsymbol{x}_i^\top - \mathbb{E}[\boldsymbol{x}_i\boldsymbol{x}_i^\top])\right)\right\|_\infty\right)$$

$$\leq 2\|\Theta\|_\infty\left(\left\| \frac{1}{n}\sum_{i=1}^{n}u_i\boldsymbol{x}_i^\top\right\|_\infty + \epsilon\left\| \frac{1}{n}\sum_{i=1}^{n}(\boldsymbol{x}_i\boldsymbol{x}_i^\top - \mathbb{E}[\boldsymbol{x}_i\boldsymbol{x}_i^\top])\right\|_{\infty,1}\right)$$

$$\leq 2\|\Theta - \Theta^* + \Theta^*\|_\infty\left(\left\| \frac{1}{n}\sum_{i=1}^{n}u_i\boldsymbol{x}_i^\top\right\|_\infty + \epsilon\left\| \frac{1}{n}\sum_{i=1}^{n}(\boldsymbol{x}_i\boldsymbol{x}_i^\top - \mathbb{E}[\boldsymbol{x}_i\boldsymbol{x}_i^\top])\right\|_{\infty,1}\right)$$

$$\leq 2(\eta + \|\Theta^*\|_\infty)\left(\left\| \frac{1}{n}\sum_{i=1}^{n}u_i\boldsymbol{x}_i^\top\right\|_\infty + \epsilon\left\| \frac{1}{n}\sum_{i=1}^{n}(\boldsymbol{x}_i\boldsymbol{x}_i^\top - \mathbb{E}[\boldsymbol{x}_i\boldsymbol{x}_i^\top])\right\|_{\infty,1}\right).$$

Then, we use the same argument as earlier to treat the sup-norm terms above (we use our assumptions on $\boldsymbol{x}_i$ and $u_i$ and application of Lemma 10) to obtain that

$$\left| \frac{\partial}{\partial\gamma_j}\text{risk}_X[\boldsymbol{\gamma},\Theta] - \frac{\partial}{\partial\gamma_j}\text{risk}[\boldsymbol{\gamma},\Theta]\right| \leq 2t(\eta + \|\Theta^*\|_\infty)(1 + \epsilon)$$

with probability at least $1 - 6dn\exp(-c\min\{nt^2, (nt)^{1/\alpha}\}) - 6d^2n\exp(-c'\min\{nt'^2, (nt')^{1/\alpha'}\})$.

Collecting all the pieces above, we obtain that for each $i \in \{1, \ldots, p\}$ the corresponding gradient difference is bounded $(|(\nabla\text{risk}_X[\boldsymbol{\gamma},\Theta] - \nabla\text{risk}[\boldsymbol{\gamma},\Theta])_i| \leq 2t(\eta + \max\{\|\boldsymbol{\gamma}^*\|_\infty, \|\Theta^*\|_\infty\})(1 + \epsilon))$ with probability at least $1 - 12d^2n\exp(-c'\min\{nt^2, (nt)^{1/\alpha'}\})$ for some $\alpha' \in [2, \infty)$ and $c' \in (0, \infty)$, depending on the distributions of inputs and noise.

Now we use 1. the definition of sup-norm and 2. our results above together with our earlier argument about implying max operator (note that the gradient vector is of dimension $p$) to obtain for each $t \in [0, \infty)$ that

$$\sup_{\boldsymbol{\beta} = \text{vec}(\boldsymbol{\gamma},\Theta)\in\mathcal{C}_{\eta,\epsilon}}\left\|\nabla\text{risk}_X[\boldsymbol{\gamma},\Theta] - \nabla\text{risk}[\boldsymbol{\gamma},\Theta]\right\|_\infty$$

$$= \sup_{\boldsymbol{\beta} = \text{vec}(\boldsymbol{\gamma},\Theta)\in\mathcal{C}_{\eta,\epsilon}}\max_{i\in\{1,\ldots,p\}}\left|\left(\nabla\text{risk}_X[\boldsymbol{\gamma},\Theta] - \nabla\text{risk}[\boldsymbol{\gamma},\Theta]\right)_i\right|$$

$$\leq 2t\left(\eta + \max\{\|\boldsymbol{\gamma}^*\|_\infty, \|\Theta^*\|_\infty\}\right)\left(1 + \epsilon\right)$$

with probability at least $1 - 12d^2pn\exp(-c\min\{nt^2, (nt)^{1/\alpha}\})$, where for the ease of notations we replace $c'$ and $\alpha'$ with $c$ and $\alpha$ (constants depending only on the distributions of the inputs and noise).

Collecting all pieces of the proof, we obtain for each $t \in [0, \infty)$ that

$$\sup_{\boldsymbol{\beta} = \text{vec}(\boldsymbol{\gamma},\Theta)\in\mathcal{C}_{\eta,\epsilon}}\left|\left(\nabla\text{risk}_X[\boldsymbol{\gamma},\Theta] - \nabla\text{risk}[\boldsymbol{\gamma},\Theta]\right)^\top(\boldsymbol{\beta}^* - \boldsymbol{\beta})\right|$$

$$\leq \eta\sup_{\boldsymbol{\beta} = \text{vec}(\boldsymbol{\gamma},\Theta)\in\mathcal{C}_{\eta,\epsilon}}\left\|\nabla\text{risk}_X[\boldsymbol{\gamma},\Theta] - \nabla\text{risk}[\boldsymbol{\gamma},\Theta]\right\|_\infty$$

$$\leq 2t\eta\left(\eta + \max\{\|\boldsymbol{\gamma}^*\|_\infty, \|\Theta^*\|_\infty\}\right)\left(1 + \epsilon\right)$$

with probability at least $1 - 12d^2pn\exp(-c\min\{nt^2, (nt)^{1/\alpha}\})$ for some $\alpha \in [2, \infty)$ and $c \in (0, \infty)$, depending on the distributions of inputs and noise. $\square$

Now, we are ready to use our Lemma 11 for extending Lemma 2 for heavier-tailed noise. First, recall

$$r_{\mathrm{orc},\alpha} \;=\; \nu(\log n)^{3/2}\frac{\big(\log(np)\big)^{\alpha}}{\sqrt{n}} \tag{22}$$

where $\alpha \in [2,\infty)$ and $\nu, c \in (0,\infty)$ are constants depending on the distributions of inputs and noise. Then, we obtain

**Lemma 12** (Empirical Processes for Heavier-tailed Noise). *Under the first two parts of Assumption 1, it holds for each reasonable stationary point $\widetilde{\boldsymbol{\beta}} = \mathrm{vec}(\widetilde{\boldsymbol{\gamma}}, \widetilde{\Theta})$ of the objective function in equation 2 that*

$$\left| \left( \nabla \mathrm{risk}_X[\widetilde{\boldsymbol{\gamma}}, \widetilde{\Theta}] - \nabla \mathrm{risk}[\widetilde{\boldsymbol{\gamma}}, \widetilde{\Theta}] \right)^{\top} (\boldsymbol{\beta}^* - \widetilde{\boldsymbol{\beta}}) \right| \le r_{\mathrm{orc},\alpha} \|\boldsymbol{\beta}^* - \widetilde{\boldsymbol{\beta}}\|_1 + \frac{r_{\mathrm{orc},\alpha}}{2n}$$

*with probability at least $1 - 1/2n$.*

*Proof of Lemma 12.* The proof follows almost the same steps as in the proof of Lemma 1. The only difference is employing Lemma 11 and the assignment of $t = (\log{(8n^2 d^2 p\lceil \log_2{(n\eta)}\rceil)})^{\alpha}/c^{\alpha}\sqrt{n}$ with different constants. $\square$

