# OpenReview forum: "Statistical Guarantees for Approximate Stationary Points of Shallow Neural Networks"
_TMLR — Accepted by TMLR_

### Review · Reviewer_ZckR · 2025-10-01

**Summary Of Contributions:**

The paper develops statistical guarantees for stationary points (and their neighborhoods) in shallow linear neural networks on regression problems, showing that they generalize essentially as well as global optima (up to logarithmic factors). The work then extends these guarantees to shallow ReLU networks under near-identity initialization assumptions, and further characterizes optimal rates for tuning parameters across different network and noise settings. The contributions are thus both conceptual—demonstrating that global optimization is not necessary for strong generalization—and technical, in providing precise theorems (Theorems 1–4) with rate guarantees.

**Audience:**

Yes

**Audience Explanation:**

Yes. While the results are restricted to shallow networks and regression problems, the community has a strong interest in theoretical insights that bring statistical learning theory closer to empirical successes. The fact that the guarantees hold for stationary points rather than just global optima makes the results more practically relevant, and the extension to ReLU networks with specific initialization conditions touches on a widely used architecture. This work is likely to appeal particularly to the subset of researchers working at the intersection of optimization, learning theory, and deep learning practice.

**Broader Impact Concerns:**

The paper is primarily theoretical and does not present immediate risks of harmful applications. Its contributions lie in improving the mathematical understanding of neural network training and generalization for shallow architectures.

**Claims And Evidence:**

Yes

**Claims Explanation:**

1. In the paragraph after Assumption 1 on page 3, there are comments and rationales for the assumptions. It would be helpful to provide more intuition on why assumptions 1 and 2 make sense. I hope to clarify with the following:
(1) In Assumption 1, the authors assume $\||\gamma^{\star}\||_1 \leq\sqrt{\log n}$ and  $\|||\Theta^{\star}\|||_1 \leq\sqrt{\log n}$. It is mentioned in Appendix E that L1 norms should increase mildly in the sample size $n$. Could the author elaborate on "mildly"?
(2) The authors mention that Assumption 2 is not necessarily true in practice but is a common one in the literature. Could you add citations here? It would also help if some intuition were provided here. For example, we can interpret each coordinate of $x$ as a feature extracted from the raw data and thus they are independent; $x$'s are centered sub-Gaussian random vectors due to the construction and training of the feature extractor, with some references added after the intuition.

2. In equation 2, the authors use L1 norm for regularization. Is L2 norm more popular and easier to analyze? If so, why is L1 norm preferred over L2 norm?

3. Theorem 3 limits the discussed reasonable stationary points to those with ly perpendicu rows. It would be helpful to elaborate on why this assumption (approximately) holds in practice and how this assumption simplifies the theoretical analyses.

**Requested Changes:**

1. Please refer to the answer to "Are the claims made in the submission supported by accurate, convincing and clear evidence" which calls for elaboration on the rationales for the assumptions.
2. Please leverage grammar tools to improve preposition usage. For example, the paragraph name "further discussion over our assumptions" on page 7 is not standard.
3. In the introduction, a more concrete and clear summary of prior work + their limitations + this work's contributions would be helpful. The authors write that "But it is currently unclear how to extend these insights to deep learning—if at all possible" as the limitation of prior work, but readers might not understand why prior results are hard to extend to deep learning. For example, they might wonder if this work does not cover deep neural networks with a large number of layers as well, then how is this work fundamentally more deep-learning-amenable than others?

---

### Review · Reviewer_8T5S · 2025-10-08

**Summary Of Contributions:**

The paper provides statistical guarantees for any (reasonable) stationary point of the shallow neural networks. The performance of these stationary points and the points within their vicinity is proved to coincide with the global optimal in excess population risk. This directly addresses the gap between optimization practice and theory. The guarantee is also proved for shallow ReLU nets under some assumptions. Small synthetic experiments show test error at an approximate stationary point is nearly identical to those at a potential global optimum.

**Audience:**

Yes

**Audience Explanation:**

The paper studies the gap between theory and practice for neural network statistical guarantees. The results in this paper show compatible performance of the stationary points compared with the global optimum in shallow neural networks. This is of certain interest to the audience of TMLR.

**Broader Impact Concerns:**

no concern

**Claims And Evidence:**

Yes

**Claims Explanation:**

The proofs appear correct.

**Requested Changes:**

(1) The assumptions on ReLU are not clear. Quantify "approximately perpendicular..", "appxomately orthogonal", etc.. in the section after Assumption 2.

(2) Does the $\tau$-appxmate guarantee work on ReLU? The use and connections of this are not very clear to me from the writing of this section.

(3) The experiments are very small. Though it shows the expected performance of the approximated stationary points and optimum. The assumptions and settings of the theoretical results are not reflected, for example, the "reasonability" assumption and how tuning oracle r can affect the result, since all the bounds involve r.

---

> ### Author Response · Authors · 2025-10-21
>
> We thank the reviewer for carefully reading our paper and for their positive and thoughtful evaluation. We will address the remaining questions carefully below, and the proposed changes, highlighted in blue, will be incorporated in the updated version, which will be available soon.
>
> **Does the $\tau$-approximate guarantee work on ReLU?**  We thank the reviewer for raising this point; in principle, the answer is yes. As stated on Page 7, the $\tau$-approximate guarantee can also hold for ReLU networks; however, we omitted it originally to avoid repetition. To be more specific, one would need to establish a uniform bound similar to Lemma 3, but for ReLU networks. From there, bounds for $\tau$-approximate solutions for ReLU networks can be readily obtained (following the same line of reasoning as in the proof of Theorem 2).   We have now clarified this point more clearly on Page 7:  ``Also, an extension of Theorem 2 to shallow ReLU networks can be obtained by combining Theorem 3 with additional machinery from empirical process theory, following the same line of reasoning as in the proof of Theorem 2. However, we omit this extension here to avoid redundancy.''
>
> **Experiments:** We thank the reviewer for asking about the experiments. We'd like to address this by pointing to our complementary experiments, Section D in the Appendix, as well as the new experiments that we have now added in the updated version.
>
> 1-We have complementary experiments in our Appendix, namely: Table 2, which extends simulations to larger data and network sizes; Table 6, which explores another optimization method, namely Adam; and results in the paragraphs **Conjecture for Deep Neural Networks** and **Conjecture Beyond Regression**, where we perform simulations on the MNIST family and classification tasks.
>
> 2-We have now conducted new experiments to study the impact of initialization on our results. In Table 3, we use random Gaussian initialization, representing a setting where the reasonability assumption does not hold. In Table 4, we use a scaled version of random Gaussian initialization (weights scaled to satisfy $||W||_1 \le \sqrt{\log n}$), representing a setting where the reasonability assumption holds. These results clearly indicate that initializing weights with small values significantly aids optimization, supporting the necessity of our reasonability assumption. We also chose Gaussian initialization to approximately satisfy the orthogonality requirement for stationaries in ReLU networks, following our discussion in **Further Discussion of Our Assumption 2**, which states that random Gaussian weights yield nearly orthogonal rows with high probability.
> We also  have pointed this out in Section D as well.
>
> 3-In Table 5, we provide results using a different rate for the tuning parameter, namely of the order $\log(np)/n^{1/4}$, compared to the optimal rate $\log(np)/n^{1/2}$ used in Section 4. Results clearly show the optimality of $\log(np)/n^{1/2}$.
>
> 4-We also provide Figure 4, which shows the relative error across a range of tuning parameters. The results clearly illustrate the bias-variance tradeoff for both linear and ReLU networks.
>
> 5-We have also added new comments in Section 4 to clarify the employed tuning rate ($\log(np)/n^{1/2}$) and initialization techniques (default setting by Python).
>
> Details of all experiments are provided in Section D, with new simulations highlighted in blue.
>
> **The assumptions on ReLU:**  Following the comment ``Quantify approximately perpendicular..", we would refer to our discussion following Assumption 2, that sates: ''The term active rows in a matrix $\Theta$ are approximately perpendicular to each other (orthogonal)  in our assumption above means, for any two distinct active rows $\Theta_{j,\cdot}$ and $\Theta_{j',\cdot}$ (where $\Theta_{j,\cdot},\Theta_{j',\cdot} \neq \mathbf{0}$), their inner product is negligible, that is $\langle \Theta_{j,\cdot}, \Theta_{j',\cdot} \rangle \approx 0.0$''.
>
> We hope we have addressed your questions thoroughly; if not, please let us know.

---

### Review · Reviewer_cSSm · 2025-10-13

**Summary Of Contributions:**

In terms of novelty: prior to this work, theoretical guarantees in statistical learning for neural networks almost always concerned either (i) global minima of the loss or (ii) specific solutions found by algorithms under idealized conditions (often in over-parameterized regimes where global minima are reachable). In contrast, this paper is the first to provide generalization guarantees that hold for any stationary point of the training objective in a neural network setting. This is a noteworthy advancemnet in our theoretical understanding of non-convex learning problems. It is an important step in bridgings a gap between optimization and generalization. I have gone through most of the theorem statements, the statements are clearly presented with relevant discussions of the assumptions and the impact of the theorem result. the paper was an easy read because of this. i did not find any particular problems in the proofs I went through, though i did not go through all the proofs.


THe paper does a good job of placing the results in context of exisiting works. e.g. It acknowledges Loh, Elsener for stationary points in simpler settings; Kawaguchi, Zhou & Liang for linear networks; Hardt & Ma, Bartlett et al., Li & Yuan for assumptions similar to theirs.etc. It covers classic results (Bartlett 1998 for neural network generalization bounds, etc.) as well as very recent ones (JMLR 2024 article by Achour et al., and even a 2025 reference on layer sparsity). The authors compare their rates with prior work: for example, they note that Taheri et al. (2021) and Lederer (2022a) had derived similar risk bounds for global optima of shallow networks. Those works did not consider stationary points, so this paper extends them. The authors properly credit those sources for the rates and highlight their own novelty that they apply to all stationary points, not just the minimizer. They also mention a recent work (Taheri et al. ) that provided theory for “approximate solutions in a convex setting” – essentially an analog of Theorem 2 but for convex problems – indicating that their notion of analyzing approximate optimization error is in line with current trends.
There are some additional references I would suggest adding to further improve the disucssion:

Haeffele & Vidal (2017) "Global Optimality in Neural Network Training" -- This work provided sufficient conditions under which every local minimum of a neural network is global

Nguyen & Hein (2018) –“Adding one neuron can eliminate all bad local minima” by Nguyen, Hein, etc. and related works by the same authors studied conditions to avoid bad local minima by slightly modifying network architectures.

Choromanska et al -- "The Loss Surfaces of Multilayer Networks"

There are some typos, please proof read. e.g.:

emprical -> empirical
"considerably progress"
"objection function"

**Audience:**

Yes

**Audience Explanation:**

Significant result in theoretical ML.

**Claims And Evidence:**

Yes

**Claims Explanation:**

the assumptions, theorem statemetns, proofs and discussions are clearly stated.

**Requested Changes:**

addition of some disucssions on some relevant papers. minor typo fixes

---

> ### Author Response · Authors · 2025-10-21
> **Response to the Reviewer cSSm**
>
> We thank the reviewer for carefully reading our paper and for their positive and thoughtful evaluation. We greatly appreciate the time taken to assess our work and to recognize its contribution to bridging optimization and generalization in neural network theory. We are also grateful for the helpful suggestions, particularly the additional references and the noted typos. We have incorporated these recommendations and improvements in the updated version of the paper, which will be made available soon. In particular, the suggested references have been added to our Related Literature section and are highlighted in blue.

---

### Decision · Action_Editor_ui6r · 2025-12-02

**Recommendation:** Accept as is

**Audience:**

Yes

**Audience Explanation:**

The paper addresses an important gap in the literature by providing the first statistical guarantees for the approximate stationary points of shallow neural nets. I believe that TMLR's audience will be very interested in such results.

**Claims And Evidence:**

Yes

**Claims Explanation:**

All reviewers provided a positive answer to this question. The claims are supported by rigorous proofs.